# Structural basis of gating modulation of Kv4 channel complexes

Yoshiaki Kise[1,5 ✉], Go Kasuya[2,5 ✉], Hiroyuki H. Okamoto[1], Daichi Yamanouchi[1], Kan Kobayashi[1,3], Tsukasa Kusakizako[1], Tomohiro Nishizawa[1,4], Koichi Nakajo[2] & Osamu Nureki[1 ✉]

Modulation of voltage-gated potassium (Kv) channels by auxiliary subunits is central to the physiological function of channels in the brain and heart[1,2]. Native Kv4 tetrameric channels form macromolecular ternary complexes with two auxiliary β-subunits—intracellular Kv channel-interacting proteins (KChIPs) and transmembrane dipeptidyl peptidase-related proteins (DPPs)—to evoke rapidly activating and inactivating A-type currents, which prevent the backpropagation of action potentials[1–5]. However, the modulatory mechanisms of Kv4 channel complexes remain largely unknown. Here we report cryo-electron microscopy structures of the Kv4.2–DPP6S–KChIP1 dodecamer complex, the Kv4.2–KChIP1 and Kv4.2–DPP6S octamer complexes, and Kv4.2 alone. The structure of the Kv4.2–KChIP1 complex reveals that the intracellular N terminus of Kv4.2 interacts with its C terminus that extends from the S6 gating helix of the neighbouring Kv4.2 subunit. KChIP1 captures both the N and the C terminus of Kv4.2. In consequence, KChIP1 would prevent N-type inactivation and stabilize the S6 conformation to modulate gating of the S6 helices within the tetramer. By contrast, unlike the reported auxiliary subunits of voltage-gated channel complexes, DPP6S interacts with the S1 and S2 helices of the Kv4.2 voltage-sensing domain, which suggests that DPP6S stabilizes the conformation of the S1–S2 helices. DPP6S may therefore accelerate the voltage-dependent movement of the S4 helices. KChIP1 and DPP6S do not directly interact with each other in the Kv4.2–KChIP1–DPP6S ternary complex. Thus, our data suggest that two distinct modes of modulation contribute in an additive manner to evoke A-type currents from the native Kv4 macromolecular complex.

Voltage-gated ion channels often form macromolecular complexes that consist of a pore-forming α-subunit and auxiliary subunits[1,6,7]. Auxiliary subunits not only regulate subcellular localization, but also modulate the gating properties of the α-subunit for the physiological functions of channels in neurons and muscle cells. However, the mechanisms of modulation by auxiliary subunits remain mostly unknown, whereas the ion selectivity and voltage-dependent activation and inactivation mechanisms have been extensively studied[8,9].

Among 12 subfamilies of Kv channels, Kv4 (Kv4.1–Kv4.3) channels mediate the transient outward A-type current, which is characterized by fast activation at subthreshold membrane potentials, fast inactivation and fast recovery from the inactivated state[3,4]. In neurons, Kv4 is localized at the soma and dendrites, where it controls the frequency of slow repetitive spike firing and attenuates the backpropagation of action potentials[2–4]. In cardiomyocytes, Kv4 controls the early repolarization phase of the action potential[10]. Kv4s exhibit a unique inactivation process called closed-state inactivation (CSI), which is mechanistically distinct from open-state inactivation (OSI) as characterized by the 'N-type inactivation' observed in Shaker-related Kv1 channels[11–18]

(Extended Data Fig. 1). After depolarization and S6 gate opening, Kv1 enters the N-type inactivation state in which the N-terminal 'inactivation ball' of the α- or β-subunit occludes the pore[13,19] (Extended Data Fig. 1). Although the N terminus of Kv4 reportedly serves as the inactivation ball when Kv4 is expressed alone[20], Kv4s close the gate immediately with unknown mechanisms and end up in a closed inactivated state irrespective of the magnitude of depolarization (that is, CSI), from which they recover with fast kinetics[12,21,22] (Extended Data Fig. 1).

Kv4s require both of two auxiliary β-subunits—cytoplasmic KChIPs and single-pass transmembrane DPPs—to achieve the native A-type current, particularly with the unique voltage dependence of inactivation kinetics characteristic of CSI and fast recovery from inactivation[5,23]. KChIPs reportedly inhibit N-type inactivation, but accelerate CSI and recovery[11,24,25]. Previous crystal structures of the Kv4.3 N-terminal domain (tetramerization 1 (T1) domain) in complex with KChIP1 revealed a cross-shaped octamer, in which four KChIP1 molecules are attached on the lateral side of the Kv4.3 T1 tetramer and interact with the N-terminal inactivation ball[26,27]. These studies support the model that KChIP prevents N-type inactivation through sequestering

[1]Department of Biological Sciences, Graduate School of Science, The University of Tokyo, Tokyo, Japan. [2]Division of Integrative Physiology, Department of Physiology, Jichi Medical University, Shimotsuke, Japan. [3]Present address: Peptidream, Kawasaki, Japan. [4]Present address: Graduate School of Medical Life Science, Yokohama City University, Yokohama, Japan. [5]These authors contributed equally: Yoshiaki Kise, Go Kasuya. ✉e-mail: yoshiaki.kise@bs.s.u-toyko.ac.jp; gokasuya@jichi.ac.jp; nureki@bs.s.u-tokyo.ac.jp

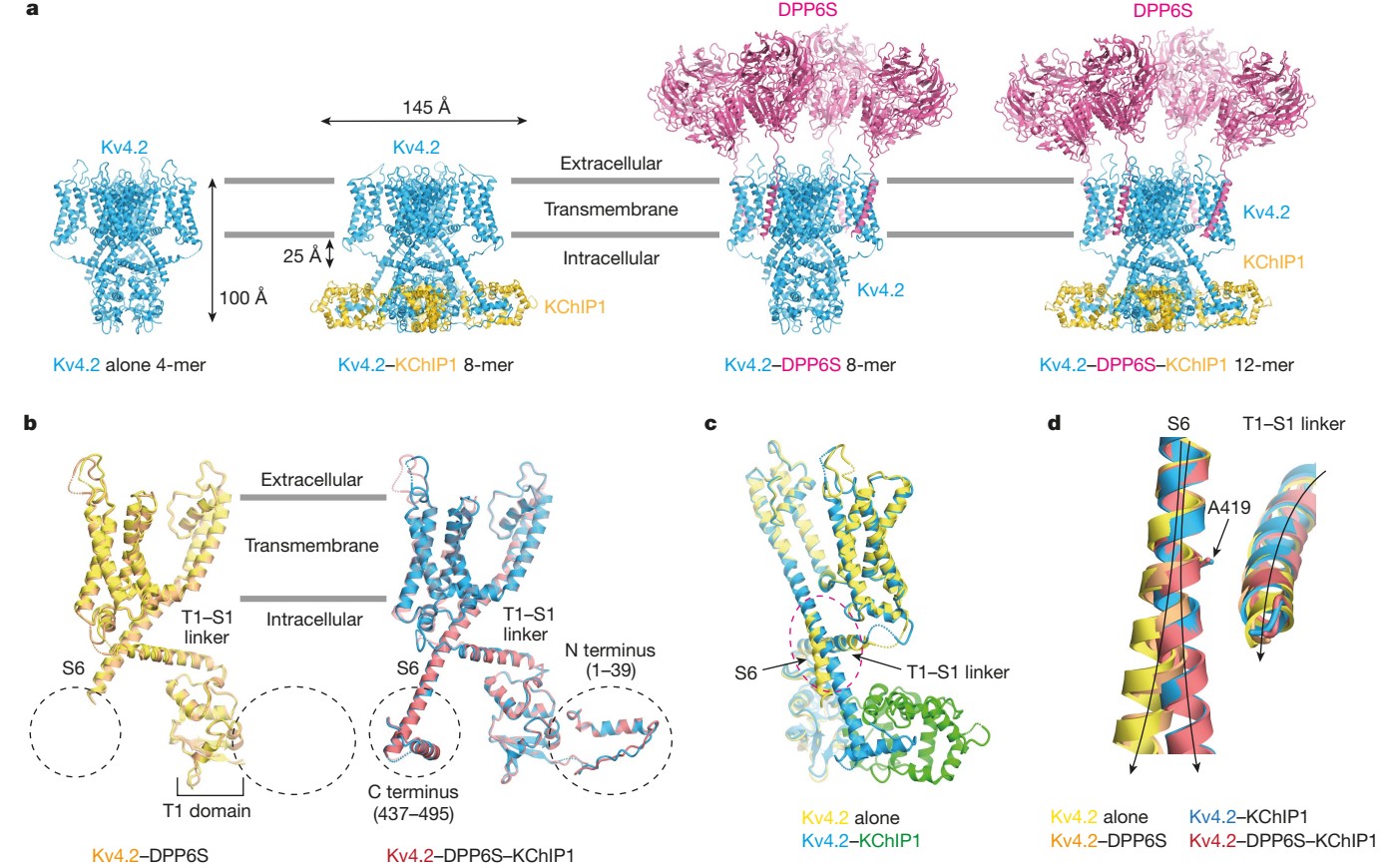

**a**

Kv4.2 alone 4-mer

Kv4.2–KChIP1 8-mer

Kv4.2–DPP6S 8-mer

Kv4.2–DPP6S–KChIP1 12-mer

145 Å

100 Å

25 Å

Extracellular

Transmembrane

Intracellular

DPP6S

DPP6S

Kv4.2

Kv4.2

Kv4.2

Kv4.2

Kv4.2

KChIP1

KChIP1

**b**

Extracellular

Transmembrane

Intracellular

S6

T1–S1 linker

T1 domain

S6

T1–S1 linker

N terminus (1–39)

C terminus (437–495)

Kv4.2–DPP6S
Kv4.2 alone

Kv4.2–DPP6S–KChIP1
Kv4.2–KChIP1

**c**

S6

T1–S1 linker

Kv4.2 alone
Kv4.2–KChIP1

**d**

S6

T1–S1 linker

A419

Kv4.2 alone
Kv4.2–DPP6S

Kv4.2–KChIP1
Kv4.2–DPP6S–KChIP1

**Fig. 1 | Structures of Kv4.2 alone and the Kv4.2–KChIP1, Kv4.2–DPP6S and Kv4.2–DPP6S–KChIP1 complexes. a**, Overall structures of the Kv4.2-alone tetramer, Kv4.2–KChIP1 octamer, Kv4.2–DPP6S octamer and Kv4.2–DPP6S–KChIP1 dodecamer (left to right). Four Kv4.2 subunits are coloured blue, four KChIP1 subunits are coloured yellow and four DPP6S subunits are coloured magenta. **b**, Structural comparison of the Kv4.2 N and C termini in the presence (right) and absence (left) of KChIP1. Protomers of Kv4.2 alone and three complexes are shown. Although both N and C termini are disordered in Kv4.2 alone and in the Kv4.2–DPP6S complex (left), both termini are resolved in the

Kv4.2–KChIP1 and the Kv4.2–DPP6S–KChIP1 complexes (right). **c**, The intracellular S6 helix of Kv4.2 alone bends at the interface on the T1–S1 linker (dashed ellipse) and is subsequently disordered. By contrast, the S6 helix of the Kv4.2–KChIP1 complex extends straight toward KChIP1. **d**, Close-up view of the superimposed image in the dashed ellipse in **c**. The intracellular S6 of Kv4.2 starts bending from A419 and extend away from the T1–S1 linker in Kv4.2 alone and in the Kv4.2–DPP6S complex. However, it keeps a close distance to the T1–S1 linker without bending in the Kv4.2–KChIP1 and the Kv4.2–DPP6S–KChIP1 complexes.

the N terminus of Kv4s[20] (Extended Data Fig. 1b). However, it remains unknown how KChIP modulates other gating properties of CSI and recovery. DPP6 has been shown to accelerate the 'gating charge' movement of Kv4.2, suggesting that DPP6 expedites the movement of the S4 voltage-sensing helix directly or indirectly[28]. However, the structure of the Kv4–DPP complex has not been reported, which hinders our understanding of the modulatory mechanisms. To gain insight into the mechanisms of gating modulation of Kv4s by KChIPs and DPPs, we determined the structures of full-length Kv4.2 alone, the Kv4.2–KChIP1 and Kv4.2–DPP6S binary complexes, and the Kv4.2–DPP6S–KChIP1 macromolecular ternary complex by single-particle cryo-electron microscopy (cryo-EM) (Fig. 1).

### Structures of Kv4.2 alone and Kv4.2–KChIP1

We first determined the cryo-EM structures of human Kv4.2 alone and the Kv4.2–KChIP1 complex (Fig. 1a, Extended Data Table 1). The fourfold symmetrical structures of both Kv4.2 alone and the Kv4.2–KChIP1 complex were determined to an overall resolution of 2.9 Å by single-particle cryo-EM analysis with *C*4 symmetry imposed (Fig. 1a, Extended Data Figs. 2–4, Supplementary Figs. 1–4). The structure of Kv4.2 alone is compact with dimensions of around 75 Å × 75 Å × 100 Å and both N- and C-terminal regions (amino acids 1–39 and 437–630)

are disordered (Fig. 1a, b, Supplementary Fig. 2a). The structure of the Kv4.2–KChIP1 complex has dimensions of around 105 Å × 105 Å × 100 Å (Fig. 1a, Supplementary Fig. 2b), which are consistent with the previous negative-stain electron microscopy structure of Kv4.2–KChIP2 at 21 Å resolution[29]. As observed in the previous crystal structures of the Kv4.3 T1 domain–KChIP1 complex[26,27], the full-length Kv4.2–KChIP1 complex forms an octamer that consists of four Kv4.2s and four KChIP1s (Fig. 1a, Supplementary Fig. 2b). As compared to the structure of Kv4.2 alone, the N terminus (amino acids 2–39) and part of the C terminus (437–450 and 473–495) of Kv4.2 are resolved and captured by KChIP1 (Figs. 1a–c, 2a, b, e, f).

The structures of Kv4.2–KChIP1 show that the Kv4.2 protomer comprises an N-terminal cytoplasmic domain with an N-terminal hydrophobic segment of approximately 40 residues in length (referred to as the inactivation ball), the T1 domain, a transmembrane domain with six transmembrane helices S1–S6, and the C-terminal cytoplasmic domain (Fig. 1b, Extended Data Fig. 5a). The transmembrane domain of Kv4.2 adopts the Shaker-type topology, with the S1–S4 voltage-sensing domain and the S5–S6 channel pore forming helices composing a homo-tetramer in a domain-swapped manner, whereby the S1–S4 voltage sensor interacts with S5 of the pore domain from the neighbouring subunit[30,31] (Supplementary Fig. 2b, Extended Data Fig. 5a, b). It adopts

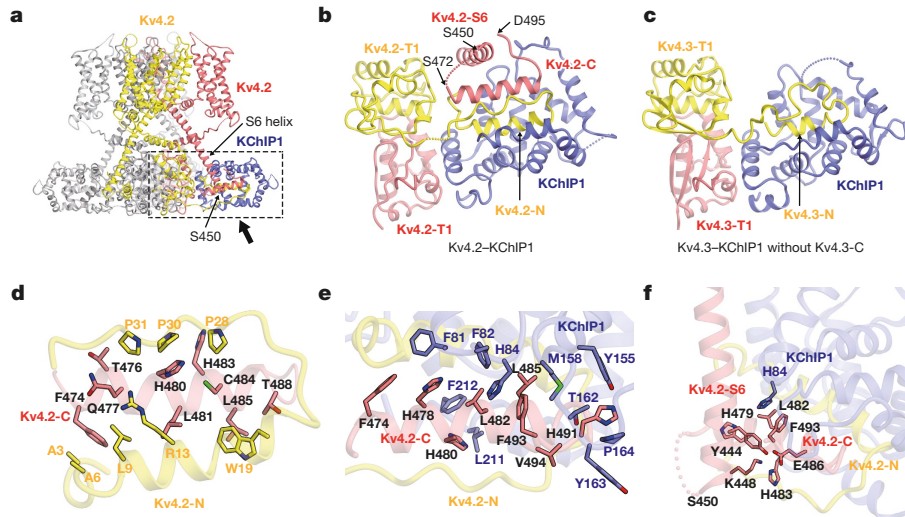

**Fig. 2 | Tripartite interactions of the Kv4.2 N terminus, Kv4.2 C terminus and KChIP1. a**, Overall structure of the Kv4.2–KChIP1 complex. The two neighbouring Kv4.2 subunits and one KChIP1 subunit are coloured yellow, red and blue, respectively. The C-terminal cytoplasmic S6 helix stops at S450 when it reaches the bottom of the complex. The interaction site of Kv4.2 and KChIP1 is highlighted by a dotted box. A magnified view from the direction of the arrow is presented in **b**. **b**, **c**, Comparison of the Kv4–KChIP1 complex with (**b**) or without (**c**) the Kv4 C terminus. Bottom views of the Kv4.2 (full-length)–KChIP1 complex (**b**) and the Kv4.3(T1)–KChIP1 complex (**c**; Protein Data Bank (PDB) code: 2NZ0) are shown. The neighbouring Kv4 subunits are coloured red and yellow. The Kv4.2 S6 helix (Kv4.2-S6) extends downward to the bottom of the complex (S450) and is further followed by the C-terminal segment (Kv4.2-C) consisting of a short helix and a loop (S472–D495), which occupies the hydrophobic space generated by the Kv4.2 N terminus (Kv4.2-N) and KChIP1 (**b**). **d**, Inter-subunit interaction of the Kv4.2 N and C termini. Residues involved in the interaction are shown. Two neighbouring Kv4.2 subunits are coloured red and yellow. **e**, Interaction of the Kv4.2 C terminus (red) and KChIP1 (blue). Residues involved in the interaction are shown. **f**, The Kv4.2 intracellular S6 helix is captured by KChIP1 and the Kv4.2 C terminus. Residues involved in the interaction are shown.

a depolarized S4 up and S6 open conformation in both Kv4.2 alone and Kv4.2–KChIP1[30–32] (Extended Data Figs. 5b, 6). The C-terminal intracellular S6 helix continuously extends from the transmembrane S6 helix toward KChIP1 (Figs. 1a, c, 2a, Extended Data Fig. 5a), which was not observed in previous studies. In addition, the intracellular S6 helix interacts with the T1–S1 linker in the structure of the Kv4.2–KChIP1 complex (Fig. 1b–d, Extended Data Fig. 5c–e). By contrast, the intracellular S6 helix bends at A419 in the structure of Kv4.2 alone, which results in a partial loss of interaction between the intracellular S6 helix and the T1–S1 linker (Fig. 1d, Extended Data Fig. 5d, e), suggesting a key mechanism of Kv4 gating modulation by KChIPs. The last 130 or so C-terminal amino acid residues of Kv4.2 (residues 496–630) are not resolved and are thus predicted to lack secondary structure (Fig. 1b, Extended Data Fig. 5a, f), suggesting their flexibility. As in the Kv1.2 structure[30,31], the tetrameric T1 domain of Kv4.2 is located under the tetrameric channel pore domains at a distance of 25 Å—provided by the long T1–S1 linker and the long intracellular S6 helix—thus creating sufficient space for K+ ions to laterally enter the channel pore (Fig. 1a, Extended Data Fig. 5a). However, it should be noted that, within the protomer of both Kv4.2 alone and the Kv4.2–KChIP1 complex, the topological relationship between the T1 and transmembrane domains is different from that in Kv1.2, owing to the distinct orientation of the T1–S1 linker following the T1 domain (Extended Data Fig. 5b, g–i). The Kv4-specific topology of the T1 domain would facilitate the proper interaction between the intracellular S6 helix and KChIP1 (Extended Data Fig. 5j).

## Kv4.2–KChIP1 interaction

KChIP1s are laterally anchored next to the T1 domains of Kv4.2, consistent with the previous crystal structures of the Kv4.3 T1 domain–KChIP1 complex[26,27] (Fig. 2a–c). The N-terminal hydrophobic segment (A2–R35) of Kv4.2, referred to as the inactivation ball, was captured by KChIP1 (Fig. 2b), which may explain why Kv4.2 exhibits a closed inactivated (CSI) mechanism, rather than an open inactivated (OSI) mechanism like Kv1.2, as previously discussed for Kv4.3[26,27]. The present structure of the full-length Kv4.2–KChIP1 complex reveals that the C terminus of Kv4.2 tightly interacts with both KChIP1 and the N terminus of Kv4.2 (Figs. 1a, c, 2b, d–f). The C-terminal cytoplasmic S6 helix continuously extends from the transmembrane S6 helix and terminates at S450, which is localized at the bottom of the complex (Fig. 2a, b, Extended Data Fig. 5a). Although the residues from G451 to G471 are disordered, the following second cytoplasmic helix with a short loop (C-terminal segment: S472–D495) fits into the hydrophobic crevice formed by KChIP1 and the Kv4.2 N-terminal segment (A2–R35) from the neighbouring Kv4.2 subunit (Fig. 2b). In addition, the cytoplasmic S6 helix (around S450) is captured by KChIP1 directly and indirectly, through the hydrophobic interactions between Kv4.2 (Y444), Kv4.2 (H479-L482-F493) and KChIP1 (H84) and the electrostatic interactions of Kv4.2 (Y444–K448) with Kv4.2 (H483–E486), respectively (Fig. 2f). Together, these interactions suggest that KChIPs modulate the inactivation and recovery of the Kv4 channel by directly regulating S6 gating, and are consistent with a previous study that suggested that the Kv4 C-terminal region is involved in modulation by KChIPs[33]. The amino acid sequence of the C-terminal helix segment (S473–T489) perfectly matches the dendritic targeting motif that is conserved in the Shal family of potassium channels—including Kv4—suggesting that this motif has a dual function as a KChIP-binding site and a dendrite localization signal[34].

To examine how the interaction of KChIP1 with the C terminus of Kv4 (S472–D495) affects Kv4 modulation, four alanine-substituted mutant versions of Kv4 were generated (F474A/H478A, H480A, L482A/L485A and H491A/F493A/V494A) on the basis of the hydrophobic interactions with KChIP1 (Fig. 2e). Using two-electrode voltage clamp (TEVC) recording in *Xenopus* oocytes, we assessed the effects of these mutations on activation, inactivation and recovery (Fig. 3, Extended Data Figs. 7, 8, Supplementary Fig. 5, Supplementary Table 1). KChIP1 decelerates the inactivation of wild type Kv4.2 at the early phase of depolarization (OSI), but accelerates inactivation during the late phase (CSI)[11,24,35] (Fig. 3a, Extended Data Fig. 7a). When expressed alone, all of the Kv4.2 mutants exhibited similar current-time traces to those of the wild type, and the H480A and H491A/F493A/V494A mutants exhibited slightly faster inactivation than the wild type (Fig. 3a, Extended Data Fig. 8,

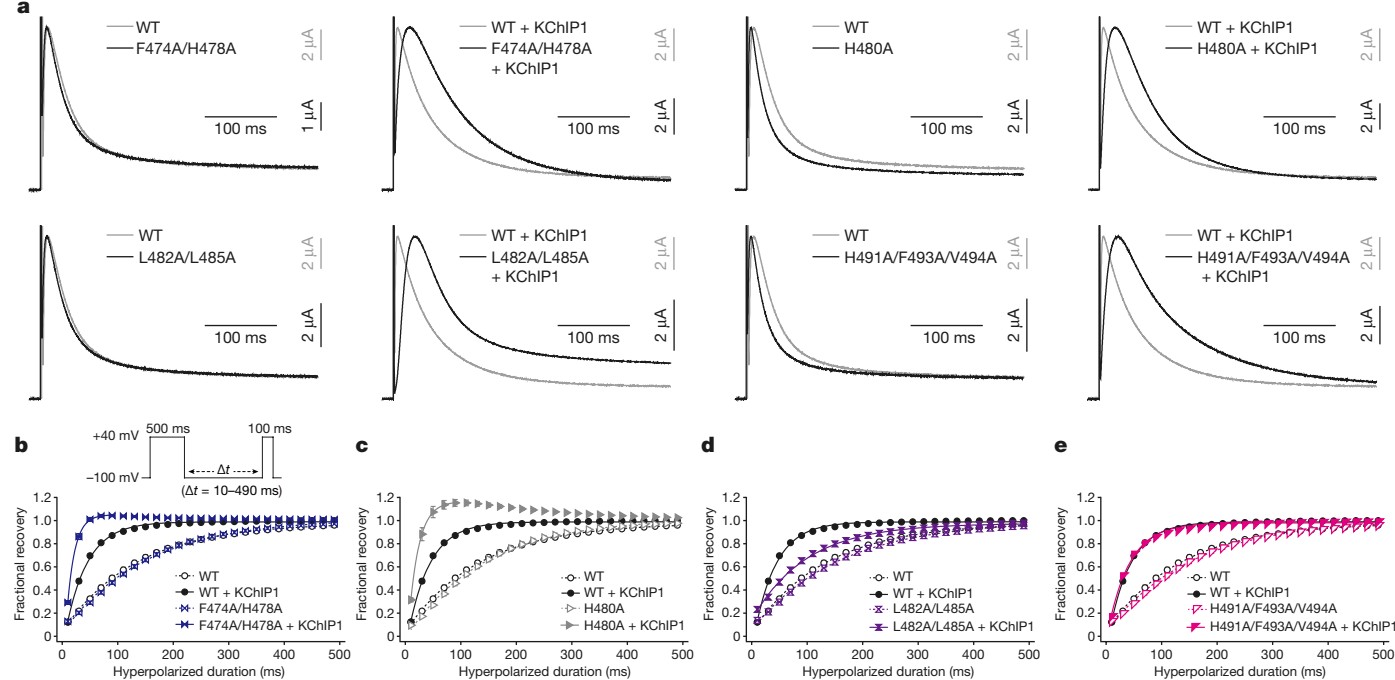

**Fig. 3 | Influence of Kv4.2–KChIP1 interface mutations on KChIP1 modulation. a**, Normalized and superposed current traces of wild-type Kv4.2 (WT) (grey) and each mutant (black) with (right) or without (left) KChIP1 elicited by test pulses of 40 mV for the qualitative comparisons of inactivation kinetics ($n = 8$ independent experiments). **b**–**e**, Comparisons of the recovery rate from inactivation in wild-type Kv4.2 with (black) or without (black and dashed) KChIP1, and in each mutant Kv4.2 (F474A/H478A (**b**), H480A (**c**), L482A/L485A (**d**) and H491A/F493A/V494A (**e**)) with (coloured) or without (coloured and dashed) KChIP1. The currents were elicited by a two-pulse protocol (inset)

using prepulses (500 ms) and test pulses (100 ms) at 40 mV with an interpulse interval ($\Delta t$) of the duration from 10 to 490 ms at −100 mV. The fractional recovery at each point was determined by normalizing the peak current amplitude of the test pulse by the amplitude of the prepulse. Symbols and bars represent mean ± s.e.m. ($n = 8$). Lines represent single-exponential fits. For the Kv4.2(F474A/H478A) with KChIP1 and Kv4.2(H480A) with KChIP1 conditions, only data obtained using prepulses from 10 ms to 90 ms were used for single-exponential fits, owing to reduced fractional recovery at longer prepulses.

Supplementary Fig. 5a, b, Supplementary Table 1). However, all of the Kv4.2 C-terminal mutants were inactivated more slowly than the wild type in the presence of KChIP1 (Fig. 3a, Extended Data Fig. 8, Supplementary Fig. 5a, b, Supplementary Table 1). In addition, whereas KChIP1 produced a negative voltage shift to activate wild type Kv4.2 in the conductance–voltage relation, as described in previous reports[24], it produced a positive voltage shift to activate all of the Kv4.2 C-terminal mutants (Extended Data Figs. 7b–e, 8, Supplementary Table 1).

Next, we assessed the effects of these mutations on voltage-dependent inactivation. KChIP1 shifted the inactivation curve of the wild type to the positive direction, indicating a relative destabilization of the inactivated state[35] (Extended Data Figs. 7f–i, 8). Although KChIP1 shifted the inactivation curve of F474A/H478A, H480A and H491A/F493A/V494A mutants to the positive direction, as for the wild type, it shifted that of the L482A/L485A mutant to the negative direction, suggesting that the Kv4.2 C terminus is important for the modulation of steady-state inactivation by KChIP1 (Extended Data Figs. 7f–i, 8, Supplementary Fig. 5c).

Finally, we assessed the effects of the mutations on recovery from inactivation, as KChIP1 reportedly accelerates the recovery from inactivation of Kv4s[24]. In the absence of KChIP1, all of the Kv4.2 C-terminal mutants exhibited quite similar recovery rates to that of the wild type (Fig. 3b–e, Supplementary Fig. 5d, Extended Data Fig. 8). However, each mutant received a different modulatory effect on the recovery rate by KChIP1 (Fig. 3b–e, Supplementary Fig. 5d, Extended Data Fig. 8). KChIP1 accelerated the recovery rate of the L482A/L485A mutant, but more weakly compared to the wild type (Fig. 3d, Extended Data Fig. 8), whereas it did not affect the recovery rate of the H491A/F493A/V494A mutant (Fig. 3e, Extended Data Fig. 8). KChIP1 accelerated the recovery rate of the F474A/H478A and H480A mutants even more strongly than the wild type, together with an 'overshoot' current[36,37] (Fig. 3b, c,

Extended Data Fig. 8). Altogether, these results indicate that the interaction of the Kv4.2 C-terminal segment with KChIP1 affects the gating modulation of Kv4.2.

## Structures of Kv4.2–DPP6S and Kv4.2–DPP6S–KChIP1

DPP6 and DPP10 are single-pass transmembrane proteins with a large extracellular domain and a short intracellular segment[38,39]. DPPs reportedly accelerate the activation, inactivation and recovery of Kv4s[38,39]. DPPs modulate Kv4s through their single transmembrane helices and short intracellular segments[40,41], suggesting that they have modulatory mechanisms that are distinct from those of KChIPs. To investigate how DPPs modulate the properties of Kv4, we solved the structures of the human Kv4.2–DPP6S binary and Kv4.2–DPP6S–KChIP1 ternary complexes (Fig. 1a, Supplementary Fig. 2c, d, Extended Data Table 1). During 3D classification with *C*1 symmetry, two different classes of structures were obtained, with two or four DPP6S molecules integrated in the complex (Extended Data Figs. 9, 10), which is consistent with the previous stoichiometric analysis of the Kv4–DPP complex[42]. The 3D classes that contained four DPP6S molecules were selected for further 3D refinement with *C*2 symmetry imposed, because two DPP6S dimers were integrated with *C*2 symmetry in the complexes (Extended Data Figs. 9, 10, Supplementary Figs. 6, 7). Owing to the flexible position of the large extracellular domains of DPP6S floating above Kv4.2, the overall resolutions are 4.2 Å and 4.5 Å for the Kv4.2–DPP6S and Kv4.2–DPP6S–KChIP1 complexes, respectively (Extended Data Figs. 9, 10). However, the focused refinement improved the resolutions of the transmembrane and intracellular regions to 3.4 Å and 3.9 Å for the Kv4.2–DPP6S and Kv4.2–DPP6S–KChIP1 complexes, respectively (Extended Data Figs. 9, 10). The dimeric crystal structure of the DPP6S extracellular domain[43] was used as a guide to construct the atomic models of the Kv4.2–DPP6S

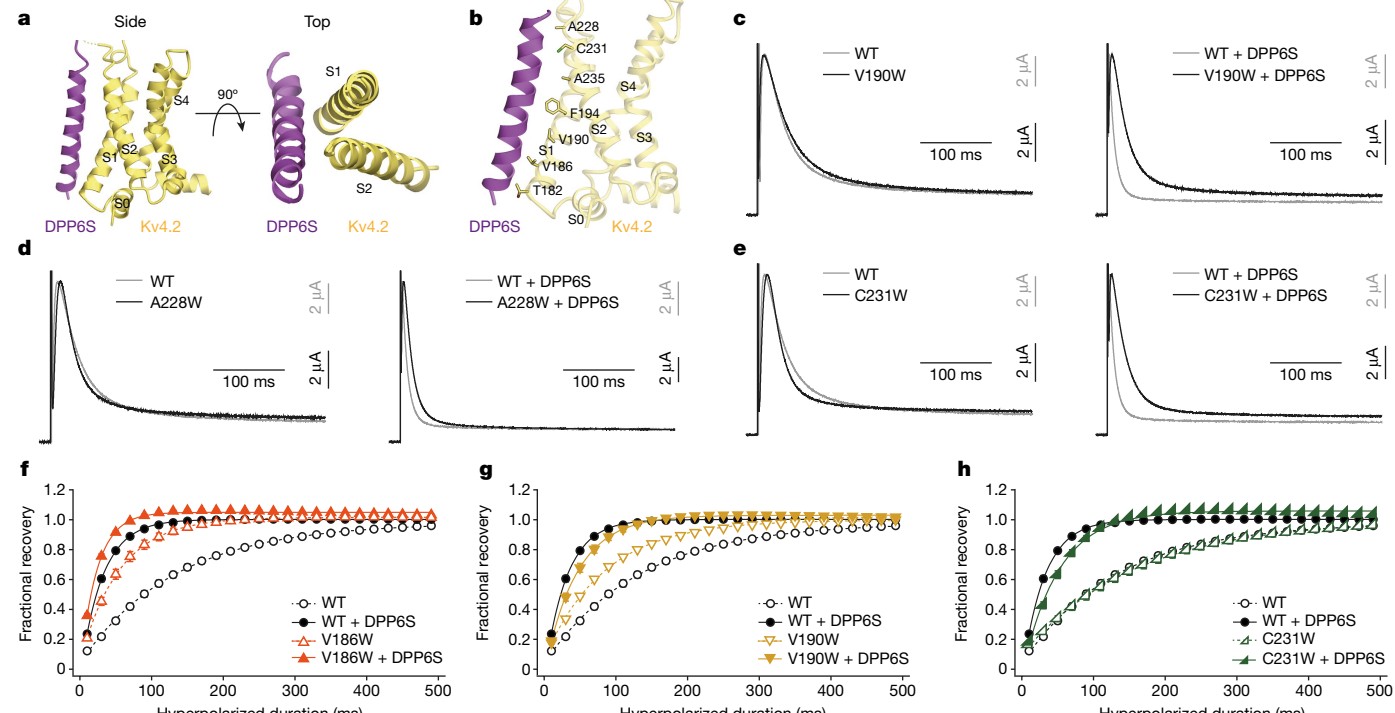

**Fig. 4 | The interaction of Kv4.2–DPP6S and the influence of Kv4.2–DPP6S interface mutations on DPP6S modulation. a**, Side and top views of the Kv4.2–DPP6S complex, focused on a single voltage-sensing domain. DPP6S interacts with S1 and S2 of the Kv4.2 voltage sensor. **b**, Residues in Kv4.2 S1–S2 facing the interface on DPP6S are shown. **c**–**e**, Normalized and superposed current traces of wild-type Kv4.2 (grey) and each mutant Kv4.2 (black) (V190W (**c**), A228W (**d**) and C231W (**e**)) with (right) or without (left) DPP6S elicited by test pulses of 40 mV for the qualitative comparisons of inactivation kinetics

($n$ = 8 independent experiments). **f**–**h**, Comparison of the recovery rate from inactivation in wild-type Kv4.2 with (black circle) or without (white circle) DPP6S, and in each mutant Kv4.2 (V186W (**f**), V190W (**g**) and C231W (**h**)) with (coloured symbol) or without (coloured open symbol) DPP6S, obtained from Supplementary Fig. 11. The fractional recovery at each point was determined by normalizing the peak current amplitude of the test pulse by the amplitude of the prepulse. Symbols and bars represent mean ± s.e.m. ($n$ = 8). Lines represent single-exponential fits.

and Kv4.2–DPP6S–KChIP1 structures (Fig. 1a, Supplementary Fig. 2c, d). The structures revealed that one DPP6S binds to one Kv4.2 in both complexes through their transmembrane domains, forming an octamer for Kv4.2–DPP6S and a dodecamer for Kv4.2–DPP6S–KChIP1 (Fig. 1a, Supplementary Fig. 2c, d). In the extracellular region, two DPP6S dimers float above the channel core. Within the intracellular part, most of the N-terminal intracellular segment of DPP6S (around 30 amino acids) is not resolved, indicating its flexibility.

Structures of the Kv4.2–DPP6S and Kv4.2–DPP6S–KChIP1 complexes adopt the S4 up and S6 open conformation, like those of Kv4.2 alone and Kv4.2–KChIP1 (Fig. 1b, Extended Data Figs. 6a, 11). Structural comparisons of Kv4.2–DPP6S and Kv4.2–DPP6S–KChIP1 as well as that of Kv4.2 alone and Kv4.2–KChIP1 further support the role of KChIP1 as a modulator of the Kv4.2 S6 helix by stabilizing the conformations of the Kv4.2 N and C termini as well as the intracellular S6 helix (Fig. 1b, d, Extended Data Fig. 11a–c).

## Kv4.2–DPP6S interaction

In the structures of the Kv4.2–DPP6S and Kv4.2–DPP6S–KChIP1 complexes, the DPP6S transmembrane helix hydrophobically interacts with the voltage-sensing domain of Kv4.2, specifically at the lower half of S1 and the upper half of S2 (Fig. 4a, b). This is consistent with a previous domain-swapping study, which suggested that DPP10 interacts with S1 and/or S2 of Kv4.3[40]. Recently, two potassium channel structures (Kv7.1 and Slo1) in complex with a modulatory transmembrane β-subunit have been reported[44,45]. In both the Kv7.1–KCNE3 and Slo1–β4 complexes, the β-subunit associates with the transmembrane interface between neighbouring α-subunits (Extended Data Fig. 12). The structure of the Kv4.2–DPP6S complex therefore represents a distinct interaction mode among

the potassium channel complexes reported thus far. The interaction of the Kv4.2–DPP6S complex somewhat resembles that of voltage-gated sodium channels, such as the Nav1.4–β1 and Nav1.7–β1 complexes, in which the β1 transmembrane helix interacts with S0 and S2 of Nav (Extended Data Fig. 12), and therefore their modulation mechanisms could be similar[46,47]. However, the specific involvement of S1 in the Kv4.2–DPP6S interaction suggests the unique modulatory mechanisms of Kv4.

Although the side chains of the DPP6S transmembrane helix could not be easily assigned owing to the lack of characteristic density (Supplementary Figs. 6–8), the Kv4.2–DPP6S structure revealed seven hydrophobic residues in S1 and S2 of Kv4.2 that face and potentially interact with DPP6S (Fig. 4b). To examine the importance of these residues in the modulation of Kv4.2 by DPP6S, we generated a series of Kv4.2–DPP6S interface mutants by substituting each residue in S1–S2 with tryptophan residue to physically interfere with their potential interaction. When expressed alone, the wild type and all Kv4.2 S1–S2 mutants exhibited similar current-time traces and voltage-dependent activation curves. (Fig. 4c–e, Supplementary Fig. 9). As reported previously[38], DPP6S accelerates activation and inactivation and also shifts the voltage-dependent activation curve to more negative membrane potentials (Fig. 4c–e, Extended Data Figs. 8, 13a–g, Supplementary Fig. 9). Although the quite rapid activation mediated by DPP6S made it difficult to evaluate the effects of the mutations on the activation kinetics, three mutants (V190W in S1; and A228W and C231W in S2) were inactivated more slowly than the wild type in the presence of DPP6S (Fig. 4c–e, Extended Data Fig. 8, Supplementary Fig. 9, Supplementary Table 1). In addition, in the presence of DPP6S these three mutants exhibited smaller negative voltage shifts for channel activation, as compared to the wild type (Extended Data Figs. 8, 13c, e, f).

We next assessed the mutational effects on voltage-dependent inactivation (Extended Data Figs. 8, 13h–n, Supplementary Fig. 10). DPP6S shifted the inactivation curves of the wild type to the negative direction with the steeper voltage dependence, indicating relative stabilization of the inactivated state[38] (Extended Data Figs. 8, 13h–n). Five mutants (T182W, V186W, F194W, A228W and C231W) showed a similar negative voltage shift in the presence of DPP6S to that of the wild type (Extended Data Figs. 8, 13h, i, k–m). By contrast, DPP6S shifted the inactivation curves of V190W in S1 and A235W in S2 mutants to the positive direction (Extended Data Figs. 8, 13j, n), suggesting that the S1 and S2 helices of Kv4.2 are important for the modulation of steady-state inactivation by DPP6S.

DPP6S reportedly accelerates the recovery of Kv4.2 from inactivation[38] (Fig. 4f–h, Extended Data Figs. 8, 13o–u, Supplementary Fig. 11). However, the V190W and C231W mutants exhibited slower recovery rates than the wild type in the presence of DPP6S, even though the V190W mutant alone recovered faster than the wild type in the absence of DPP6S (Fig. 4g, h, Extended Data Fig. 8). The V186W mutant alone recovered faster than the wild type, which made it difficult to evaluate the effect of DPP6S on this mutant (Fig. 4f, Extended Data Fig. 8). Together, all these results indicate that DPP6S modulates the activation, inactivation and recovery of Kv4.2 through interactions with the S1 and S2 helices of the Kv4.2 voltage-sensing domain.

## Conclusions

The structures we present here, combined with complementary electrophysiological analyses, suggest that KChIP1 stabilizes the S6 conformation to modulate synchronized and accelerated gating of the S6 helices within the tetramer, preventing N-type inactivation but promoting fast CSI and recovery. On the other hand, DPP6S may accelerate the voltage-dependent movement of the S4 helices by stabilizing the S1–S2 conformation. KChIP1 and DPP6S do not directly interact with each other, and they interact with distinct structures of Kv4.2 to modulate its gating kinetics in different manners. Therefore, our results suggest that these two distinct modes of modulation additively contribute to evoking A-type currents from the native Kv4 macromolecular complex by eliminating OSI, and accelerating CSI and fast recovery from CSI ('Discussion' in Methods).

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

# Methods

## Data reporting

No statistical methods were used to predetermine sample size. The experiments were not randomized and the investigators were not blinded to allocation during experiments and outcome assessment.

## Cloning, expression and purification of Kv4.2–KChIP1, Kv4.2–DPP6S and Kv4.2–KChIP1–DPP6S

The DNAs encoding human Kv4.2, KChIP1 and DPP6S were PCR-amplified from a human brain cDNA library (Zyagen). The C-terminally GFP–8×His-tagged Kv4.2, C-terminally Flag-tagged Kv4.2, N-terminally 8×His–GFP-tagged KChIP1, N-terminally Flag-tagged KChIP1 and C-terminally 8×His-tagged DPP6S were subcloned into the pEG BacMam expression vector. Recombinant baculoviruses of Kv4.2, KChIP1 and DPP6S were generated in *Spodoptera frugiperda* Sf9 cells (American Type Culture Collection, CRL-1711), using the Bac-to-Bac system (Invitrogen). Cultures (3.2 l) of HEK293S GnTI⁻ cells ($1 \times 10^6$–$1.5 \times 10^6$ cells per ml) in Freestyle 293 medium (Gibco) supplemented with 2% FBS were infected with 320 ml of P2 virus mixtures of Kv4.2–GFP–8×His only, Kv4.2–Flag:8×His–GFP–KChIP1 (1:2), Kv4.2–GFP–8×His:DPP6S–8xHis (1:1), or Kv4.2–GFP–8×His:Flag–KChIP1:DPP6S–8×His (1:1:1) for 18–24 h at 37 °C. After adding 10 mM sodium butyrate, the cells were cultured at 30 °C for another 36 h to induce protein expression. The cells expressing Kv4.2–KchIP1, Kv4.2–DPP6S or Kv4.2–DPP6S–KChIP1 complexes were collected and resuspended in 80 ml of buffer consisting of 50 mM Tris, pH 7.4, 150 mM KCl and 2 mM CaCl₂ with protease inhibitor cocktails (Roche), sonicated, and centrifuged at 10,000g for 10 min. The supernatant was further ultracentrifuged at 40,000 rpm for 1 h to precipitate the membrane fraction. The membrane fraction was solubilized by an incubation at 4 °C for 1 h in 50 mM Tris buffer, pH 7.4, containing 150 mM KCl, 2 mM CaCl₂ and 1.5% DDM-0.3% CHS with protease inhibitor cocktails. The cells expressing Kv4.2 alone were collected and directly solubilized by an incubation at 4 °C for 1 h in 50 mM Tris buffer, pH 7.4, containing 150 mM KCl, 2 mM CaCl₂ and 1.5% DDM-0.3% CHS with protease inhibitor cocktails. The cell lysate was cleared by ultracentrifugation at 40,000 rpm for 30 min, and the supernatant was incubated with GFP minimizer nanobody resin for 1 h. The resin was washed with 50 mM Tris buffer, pH 7.4, containing 500 mM KCl, 2 mM CaCl₂ and 0.03% GDN. The GFP tag was cleaved by TEV protease overnight at 4 °C in wash buffer. The Kv4.2 alone and Kv4.2 complexes were further purified by size-exclusion chromatography on a Superose 6 10/300 GL increase column (GE Healthcare) equilibrated with 50 mM Tris buffer, pH 7.4, 150 mM KCl, 2 mM CaCl₂ and 0.03% GDN. Peak fractions were pooled, concentrated to 1.5–2 mg ml⁻¹ using a 100-kDa MWCO centrifugal device (Amicon), and ultracentrifuged at 4,000g for 10 min before grid preparation.

## Grid preparation, data collection and data processing

Quantifoil R1.2/1.3 holey carbon Au grids (Quantifoil) were glow-discharged for 2 min. Afterwards, 3-μl portions of protein samples were applied on the grids, blotted for 4 s with blot force 10 at 100% humidity, and frozen in liquid ethane cooled with liquid nitrogen by using a Vitrobot Mark IV (FEI). Grids were first subjected to Talos Arctica (FEI) with a K2 direct electron detector (Gatan) to screen good ones for data collection using EPU (v.1.19) (FEI). Then, grids were subjected to Titan Krios (FEI) microscopy with a K3 direct electron detector (Gatan). Datasets of Kv4.2–KChIP1, Kv4.2–DPP6S and Kv4.2–DPP6S–KChIP1 complexes were collected with a total dose of around 50 electrons per Å² per 48 frames by the standard mode and datasets of Kv4.2 alone were collected with a total dose of around 50 electrons per Å² per 64 frames by the CDS mode, using SerialEM (v.3.7.10)[48] in the counting mode with a pixel size of 0.83 Å and defocus range of 0.8 to 1.6 μm. Data were processed and structures were determined with RELION v.3.0 or 3.1. For data processing details, see Extended Data Figs. 3, 4, 9, 10.

## Model building

Models were built with Coot[49]. Models for Kv4.2 and KChIP1 were manually built with reference to the crystal structures of the Kv1.2-2.1 chimera (PDB code: 2R9R) and KChIP1 (2I2R). Owing to its flexibility and low-resolution map, modelling for DPP6S was performed by fitting to the crystal structure of DPPX (1XFD), using MOLREP (v.11.7). The structural models were refined with phenix.real_space_refine[50]. The pore radius was calculated with HOLE (v.2.2.004)[51]. Graphics were prepared using UCSF Chimera (v.1.14) and CueMol2 (v.2.2.3.443) (http://www.cuemol.org/).

## Protein expression in *Xenopus laevis* oocytes

The human Kv4.2 (NP_036413.1; wild type and mutants), human KChIP1 (NP_055407.1; wild type), and human DPP6S (NP_001927.3; wild type) genes were cloned into the pGEMHE expression vector[52]. The cRNAs were transcribed using a mMESSAGE mMACHINE T7 Transcription Kit (Thermo Fisher Scientific). Oocytes were surgically taken from female *Xenopus laevis* anaesthetized in water containing 0.15% tricaine (Sigma-Aldrich, E10521) for 15–30 min. They were treated with collagenase (Sigma-Aldrich, C0130) for 6–7 h at room temperature to remove the follicular cell layer. Defolliculated oocytes of a similar size at stage V or VI were selected and microinjected with 50 nl of cRNA solution. They were then incubated for 1–2 days at 18 °C in MBSH buffer, containing 88 mM NaCl, 1 mM KCl, 2.4 mM NaHCO₃, 10 mM HEPES, 0.3 mM Ca(NO₃)₂, 0.41 mM CaCl₂, and 0.82 mM MgSO₄, pH 7.6, supplemented with 0.1% penicillin–streptomycin solution (Sigma-Aldrich, P4333)[25,42]. All experiments were approved by the Animal Care Committee of Jichi Medical University and were performed following the institutional guidelines.

## Two-electrode voltage clamp recordings

Ionic currents were recorded under two-electrode voltage clamp with an OC-725C amplifier (Warner Instruments) at room temperature. The bath chamber was perfused with ND-96 buffer, containing 96 mM NaCl, 2 mM KCl, 1.8 mM CaCl₂, 1 mM MgCl₂ and 5 mM HEPES, pH 7.5. The microelectrodes were drawn from borosilicate glass capillaries (Harvard Apparatus, GC150TF-10) using a P-1000 micropipette puller (Sutter Instrument) to a resistance of 0.2–0.5 MΩ and filled with 3 M KCl. Generation of voltage-clamp protocols and data acquisition were performed using a Digidata 1550 interface (Molecular Devices) controlled by the pClampex 10.7 software (Molecular Devices). Data were sampled at 10 kHz and filtered at 1 kHz by the pClampfit 10.7 software (Molecular Devices).

## Data analysis

**For the voltage-dependent activation.** The holding potential was −80 mV. After 500 ms of hyperpolarization at −110 mV to remove inactivation, currents were elicited by 400-ms test pulses to membrane potentials from −80 to 40 mV with 10-mV increments. Conductance values were calculated from peak current amplitudes by normalizing to the maximum current amplitude obtained in the experiment, assuming a linear open channel current–voltage relationship and a reversal potential of −98 mV (normalized chord conductance). Normalized peak conductance was plotted versus voltage and fitted with single Boltzmann functions to estimate the half-activation voltage ($V_{1/2,act}$) and the effective charge ($z_{act}$) in Extended Data Fig. 8.

**Recovery from inactivation.** The currents were elicited by a two-pulse protocol using the prepulse (500 ms) and the test pulses (100 ms) at 40 mV with an interpulse interval of the duration from 10 to 490 ms at −100 mV. The fractional recovery at each point was determined by normalizing the peak current amplitude of the test pulse by the amplitude of the prepulse and fitted with single exponential functions to estimate the recovery time constant ($\tau_{rec}$) in Extended Data Fig. 8. For

the Kv4.2 (F474A/H478A) with KChIP1 and Kv4.2 (H480A) with KChIP1 conditions, only data obtained using prepulses from 10 ms to 90 ms were used for single-exponential fits owing to reduced fractional recovery at longer prepulses.

**Voltage-dependent prepulse inactivation.** The holding potential was −100 mV. After 5 s of prepulses from −120 mV to 0 mV with 10-mV increments, currents were elicited by 250-ms test pulses at 60 mV. The fractional recovery at each point was determined by normalizing the peak current amplitude of the test pulse by the test pulse after the prepulse of −120 mV and fitted with single Boltzmann functions to estimate the half-inactivation voltage ($V_{1/2,inact}$) and the effective charge ($z_{inact}$) in Extended Data Fig. 8.

## Statistical analysis

The electrophysiological data were expressed as mean ± s.e.m. ($n$ = 8). Differences between wild type and mutants, between wild type with KChIP1 and mutants with KChIP1, and between wild type with DPP6S and mutants with DPP6S were evaluated by Dunnett's test with EZR software[53].

## Discussion

**Modulation by KChIP1.** KChIPs reportedly prevent OSI and accelerate CSI and recovery from inactivation[11,24,35] (Fig. 3, Extended Data Fig. 7a, Supplementary Fig. 5). The structural comparison between Kv4.2 alone and the Kv4.2–KChIP1 complex provides insight into how KChIPs modulate the gating of Kv4s. In the Kv4.2–KChIP1 complex, KChIP1s bind and sequester the both N-terminal inactivation ball and the C terminus (amino acids 472–495) of Kv4.2, which would therefore result in preventing N-type inactivation. Moreover, while S6 gating helices adopt a more flexible conformation with weaker interaction with T1–S1 linkers in the structure of Kv4.2 alone, KChIP1 stabilizes these structures and enhances their interactions in the structure of Kv4.2–KChIP1. These structural changes mediated by KChIPs, together with the following three observations and reports, might explain how KChIPs accelerate the S6 gating of Kv4s, including CSI and recovery from inactivation. First, one KChIP1 stabilizes the S6 conformation as well as the N terminus from the neighbouring subunit of Kv4.2. Second, one KChIP1 also interacts with two T1 domains from neighbouring subunits[26,27] (Fig. 2b). Third, previous functional studies have suggested that the T1–S1 linker of Kv4 dodecameric channels undergoes major conformational shifts tightly coupled to movements of the S6 tail[54,55], although we do not know what the T1 conformational change is. Together, these structural features mediated by KChIP1 may allow synchronized and accelerated S6 gating to enable fast CSI and recovery (Extended Data Fig. 14a).

**Modulation by DPP6.** DPP6S reportedly accelerates the activation, inactivation, and recovery of K4 channels[38]. In the Kv4.2–DPP6S complexes, the single-spanning transmembrane helix of DPP6S apparently stabilizes the structure of S1 and S2 helices because it simultaneously interacts with the lower half of S1 and the upper half of S2 (Fig. 4a, b). DPP6S reportedly accelerates both the outward and the inward movements of the Kv4.2 gating charge after depolarization and repolarization, respectively[28]. Among the hypotheses to explain the voltage dependency in voltage-gated channels, the hypothesis that S4 slides on the surface formed by S1 and S2 depending on the membrane potential might be most likely[13]. Therefore, the stabilization of the S1–S2 conformation may facilitate the movement of the S4 helices upon depolarization and repolarization, which could explain the fast kinetics of activation and recovery from the closed inactivated state (Extended Data Fig. 14b).

Previous studies suggest that DPP6S accelerates both OSI and CSI of Kv4s[39,56] (Extended Data Fig. 14b). The acceleration of OSI by DPP6S could involve the N-terminal intracellular domain of DPP6S and the N terminus of Kv4s[39]; however, both regions are disordered in the

structure of Kv4.2–DPP6S and further investigations are required. Previous studies suggest that the dynamic interaction of the S4–S5 linker and the S6 gate is the molecular basis of CSI[12,22]. Therefore, the acceleration of CSI by DPP6S could be, at least in part, attributed to the accelerated conformational change of S4 as discussed above (Extended Data Fig. 14b).

**Modulation in the Kv4 macromolecular ternary complex.** Native Kv4s form macromolecular ternary complex with KChIPs and DPPs. The structure of the Kv4.2–DPP6S–KChIP1 dodecameric complex (Fig. 1a) supports the additive contribution of KChIPs and DPPs to the modulation of Kv4s in the ternary complex. KChIP1 and DPP6S interact with distinct structures of Kv4.2 to modulate its gating kinetics in different manners (Figs. 1a, 2, 4a). In addition, KChIP1 and DPP6S do not interact with each other. Overall, the modulatory mechanisms of Kv4.2 by KChIP1 and DPP6S are different, and therefore, native Kv4s form ternary macromolecular complexes with both KChIPs and DPPs to exhibit eliminated OSI, accelerated CSI and fast recovery rate from CSI[5] (Extended Data Fig. 14c). Structurally mechanistic elucidations of CSI will further clarify the mechanisms of modulation by KChIPs and DPPs.

**Insight into closed-state inactivation of Kv4.2.** The structural correlates of Kv4 in closed-state inactivation (CSI) remain unknown. Previous studies have proposed that the interaction between the S4–S5 linker and S6 in Kv4s, which couples the S4 movement to S6 gating in Kv1, might be lost following the upshifted movement of S4 during depolarization[12,21,22] (Extended Data Fig. 1b). Indeed, the amino acid sequences of Kv4 around the S4–S5 linker and S6 on the intracellular side are unique among the Shaker-related Kv subfamilies (Kv1–Kv4) (Supplementary Fig. 12a), and mutations of these regions affect the CSI kinetics of Kv4[21,22] (Supplementary Fig. 12b). In addition, the open conformation of Kv4.2 complexes revealed several Kv4-specific residues involved in the intra-subunit interactions between the S4–S5 linker and S6, as well as the inter-subunit interactions between the S4–S5 linker and S5 (Supplementary Fig. 12a, c, d). Further study of this 'pre-closing' conformation may lead to elucidating the mechanism of CSI. Together, future structural studies of the resting and closed inactivated states will provide more mechanistic insights into Kv4 channel gating, CSI and modulation by auxiliary subunits.

## Reporting summary

Further information on research design is available in the Nature Research Reporting Summary linked to this paper.

## Data availability

The cryo-EM density maps and atomic coordinates have been deposited in the Electron Microscopy Data Bank (EMDB). The accession codes for the maps are EMD-31433 (Kv4.2–KChIP1-whole (map A)), EMD-31009, (Kv4.2–KChIP1-whole (map A)), EMD-31005 (Kv4.2–KChIP1-TM (map B)), EMD-31013 (Kv4.2–DPP6S-whole (map E)), EMD-31011 (Kv4.2–DPP6S-TM and cyto (map F)), EMD-31012 (Kv4.2–DPP6S-TM and EC (map G)), EMD-31019 (Kv4.2–DPP6S–KChIP1-whole (map H)), EMD-31016 (Kv4.2–DPP6S–KChIP1 (TM and cyto (map I)), EMD-31018 (Kv4.2–DPP6S–KChIP1-TM and EC (map J)) and EMD-31399 (Kv4.2 alone (map X)). The PDB accession codes for the coordinates are 7F3F7E84 (Kv4.2–KChIP1-whole), 7E83 (Kv4.2–KChIP1-cyto), 7E7Z (Kv4.2–KChIP1-TM), 7E8B (Kv4.2–DPP6S-whole), 7E87 (Kv4.2–DPP6S-TM and cyto), 7E89 (Kv4.2–DPP6S-EC), 7E8H (Kv4.2–DPP6S–KChIP1-whole), 7E8E (Kv4.2–DPP6S–KChIP1-TM and cyto), 7E8G (Kv4.2–DPP6S–KChIP1-EC) and 7F0J (Kv4.2 alone). For detail, see also Extended Data Table 1, Extended Data Figs. 3, 4, 9, 10.

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

**Acknowledgements** We thank the members of the O.N. laboratory and the staff scientists at the University of Tokyo's cryo-EM facility, especially Y. Sakamaki, H. Yanagisawa, A. Tsutsumi, M. Kikkawa and R. Danev. This work was supported by a MEXT Grant-in-Aid for Specially Promoted Research (grant number 16H06294) and JST Core Research for Evolutional Science and Technology (CREST) (grant number 20344981) to O.N.; by KAKENHI (grant numbers18K06207 to Y.K. and 20H03200 to G.K.); and by the Platform Project for Supporting Drug Discovery and Life Science Research (Basis for Supporting Innovative Drug Discovery and Life Science Research) from AMED under grant number JP19am01011115 (support number 1110). Y.K. was also supported by Uchang Cho Institute of Science.

**Author contributions** Y.K. designed the whole study. Y.K. performed the cryo-EM analyses, with sample preparation assistance from H.H.O. and D.Y. Y.K., K.K., T.K. and T.N. performed cryo-EM data collection and processing. Y.K., G.K. and K.N. designed and G.K performed the electrophysiological experiments, and G.K. and K.N. analysed the data. Y.K. performed model building and model refinement with assistance from T.N. and T.K. Y.K. wrote the initial manuscript. Y.K., G.K., K.N. and O.N. edited the manuscript with help from all of the other authors. Y.K., G.K. and O.N. supervised all of the research.

**Competing interests** The authors declare no competing interests.

**Additional information**
**Correspondence and requests for materials** should be addressed to Yoshiaki Kise, Go Kasuya or Osamu Nureki.

**a**

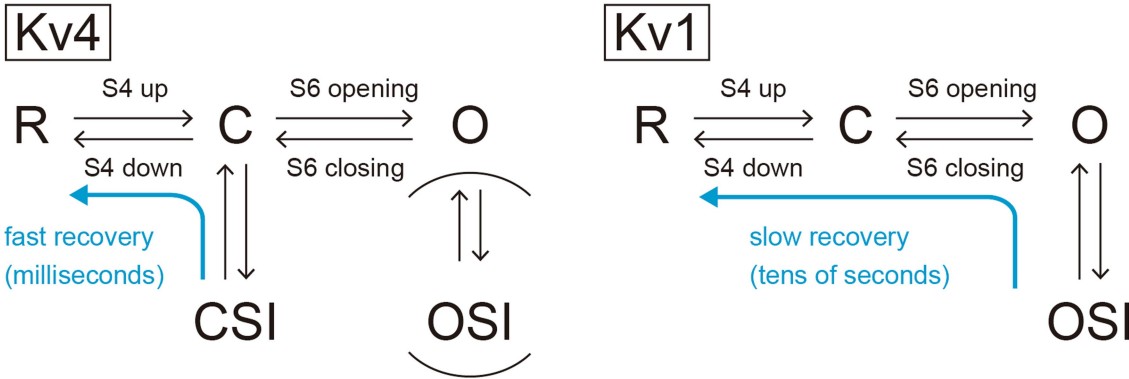

**b**

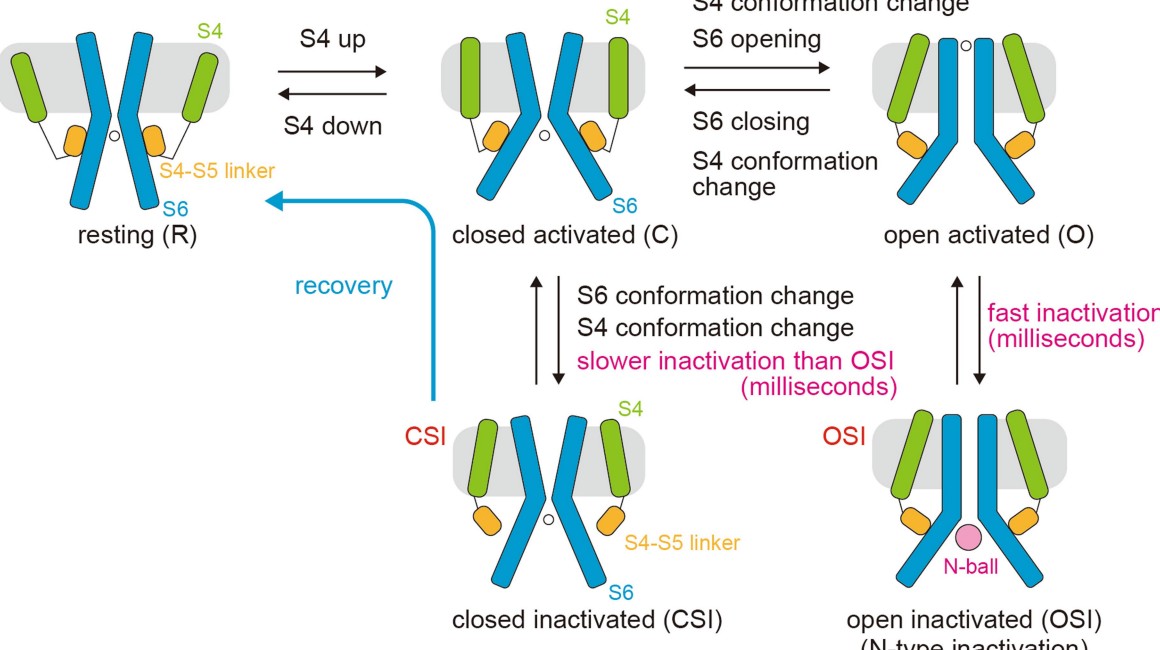

**Extended Data Fig. 1 | Kv4-specific gating mechanism.** Comparison of the inactivation mechanisms between Kv4 and Kv1. R: resting state; C: closed activated state; O: open activated state; OSI: open inactivated state (open state inactivation); CSI: closed inactivated state (closed state inactivation). Upon depolarization, Kv4 adopts CSI to become inactivated. CSI involves the closure of the S6 gate. OSI plays a minor role in Kv4 inactivation, although it is the main pathway to become inactivated in Kv1. Upon repolarization, Kv4 returns to the resting state (R) from CSI with the milliseconds order of the fast recovery rate whereas Kv1 returns to the resting state from OSI with the tens of seconds of the slow recovery rate. For a detailed schematic explanation, please see (**b**) below. **a**. Gating model of Kv4 without auxiliary subunits. Upon depolarization, the S4 (green) adopts the "up" conformation (closed activated: C), and then the S6 gate opens via the interaction with the S4-S5 linker (orange) to form the open

activated conformation (O). After activation, Kv4 takes two distinct inactivation pathways. Open activated Kv4 (O) goes to an open inactivated state (OSI) through the occlusion of the pore by its own N-terminus (N-ball), which is characterized by fast inactivation kinetics and called N-type inactivation or open state inactivation (OSI). However, the open inactivated state of Kv4 (OSI) is not stable, and Kv4 reverts to a closed activated state (C) and then goes to a closed inactivated state (CSI). This process is characterized by slower inactivation kinetics than OSI and referred to as closed state inactivation (CSI) through the S6 closing and S4 conformational change. It should be noted that CSI is still a fast millisecond-order process. As a result, during depolarization Kv4 accumulates in a closed inactivated state (CSI). Upon repolarization, Kv4 recovers from CSI to the resting state (R) through the sliding down of S4 and the conformational change of S6.

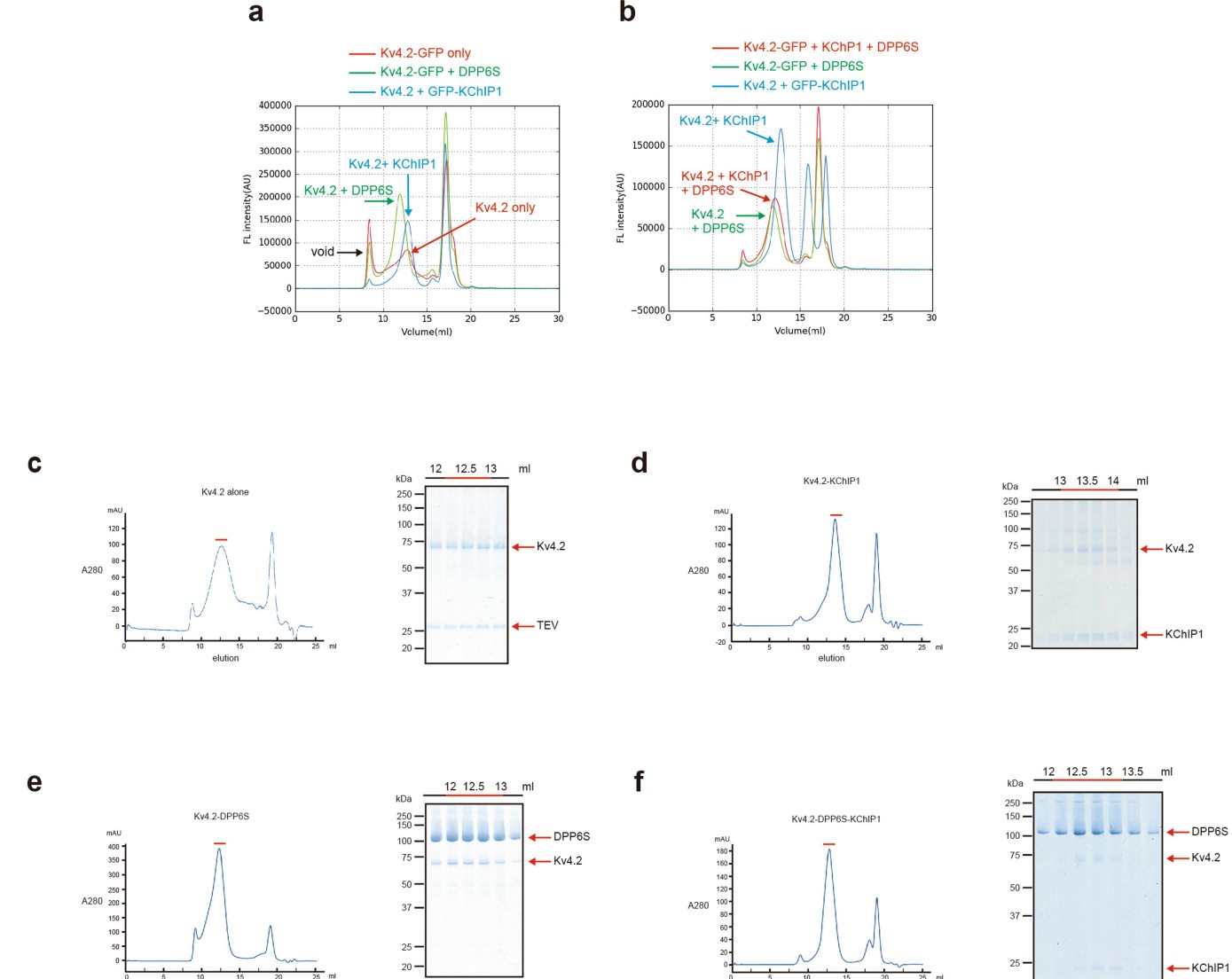

**Extended Data Fig. 2 | Expression and purification of Kv4.2 alone and the Kv4.2–KChIP1, Kv4.2–DPP6S and Kv4.2–DPP6S–KChIP1 complexes.**
**a**, Fluorescence-detection size exclusion chromatography (FSEC) analyses of the Kv4.2 α-subunit alone or in complex with KChIP1 or DPP6S. The human full-length Kv4.2 alone showed a relatively broad but still monodisperse peak. When co-expressed with KChIP1 or DPP6S, Kv4.2 showed a monodisperse and sharp peak with high expression. **b**. FSEC analyses of the Kv4.2–KChIP1, Kv4.2–DPP6S, and Kv4.2–DPP6S–KChIP1 complexes. **c**. Representative size-exclusion chromatography (SEC) profile of the Kv4.2 alone (left) and SDS-PAGE of the SEC

peak fractions stained by Coomassie Brilliant Blue (CBB) (right). Fractions indicated by red bars were pooled for cryo-EM grid preparation.
**d**. Representative SEC profile of the Kv4.2–KChIP1 complex (left) and SDS-PAGE of the SEC peak fractions stained by CBB (right). Fractions indicated by red bars were pooled for cryo-EM grid preparation. **e**. Representative SEC profile of the Kv4.2–DPP6S complex (left) and SDS-PAGE of the SEC peak fractions stained by CBB (right). **f**. Representative SEC profile of Kv4.2–DPP6S–KChIP1 complex (left) and SDS-PAGE of the SEC peak fractions stained by CBB (right).

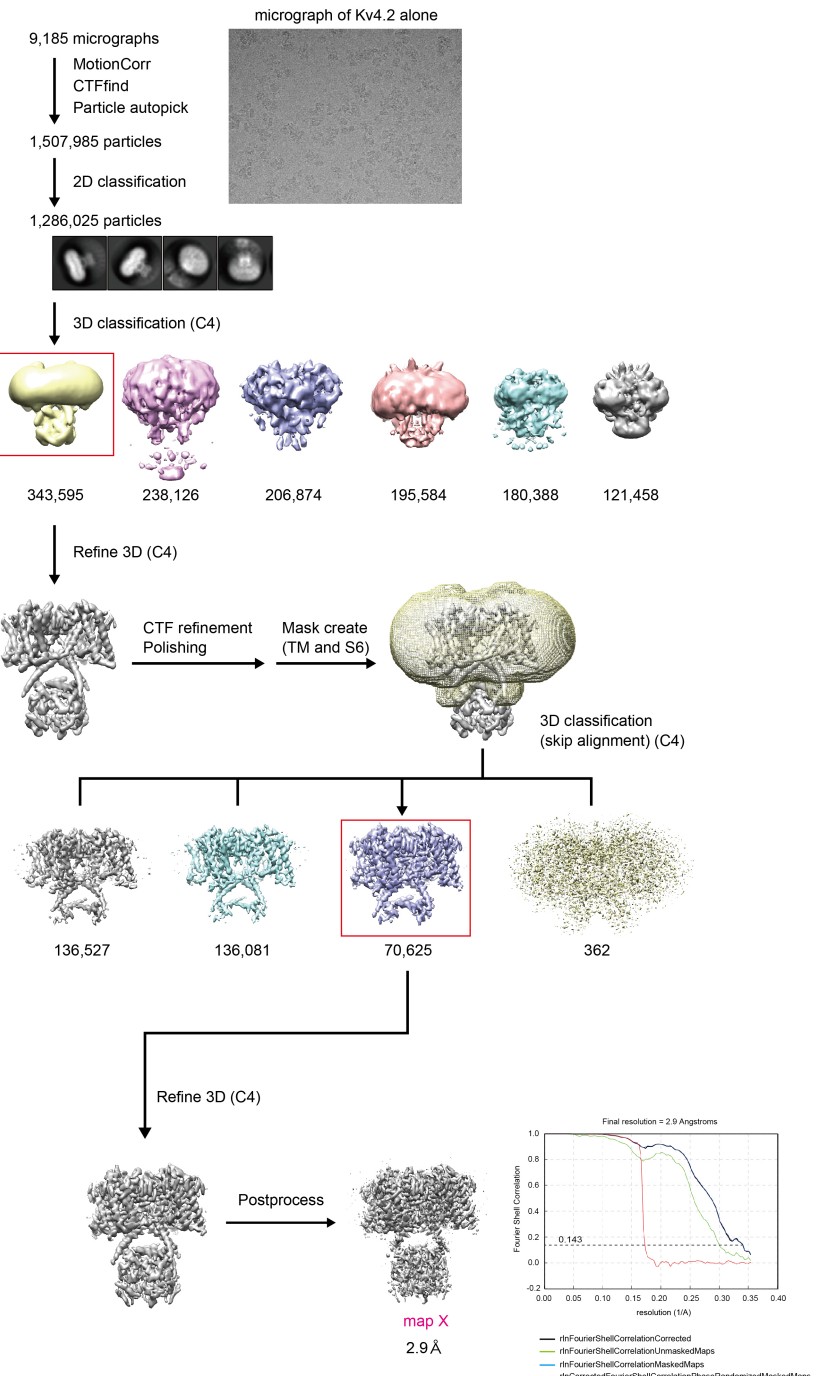

**Extended Data Fig. 3 | Cryo-EM micrograph, data processing and electron microscopy map of Kv4.2 alone.** Each step of data processing leading to the final structure of Kv4.2 alone and representative images of cryo-EM micrograph and 2D classes are shown.

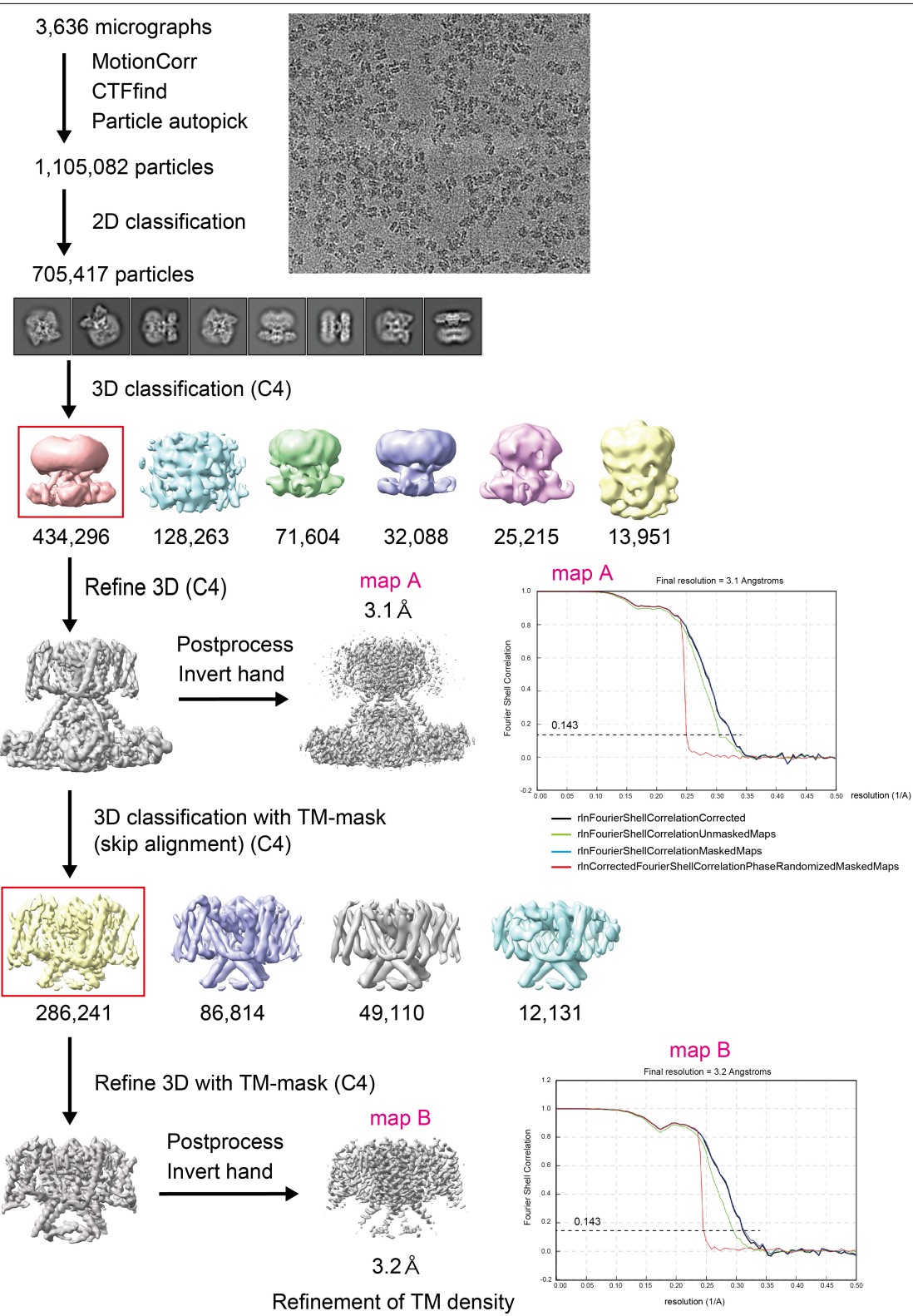

**Extended Data Fig. 4 | Cryo-EM micrograph, data processing and electron microscopy map of the Kv4.2–KChIP1 complex.** Each step of data processing leading to the final structure of Kv4.2-KChIP1 complex and representative images of cryo-EM micrograph and 2D classes are shown.

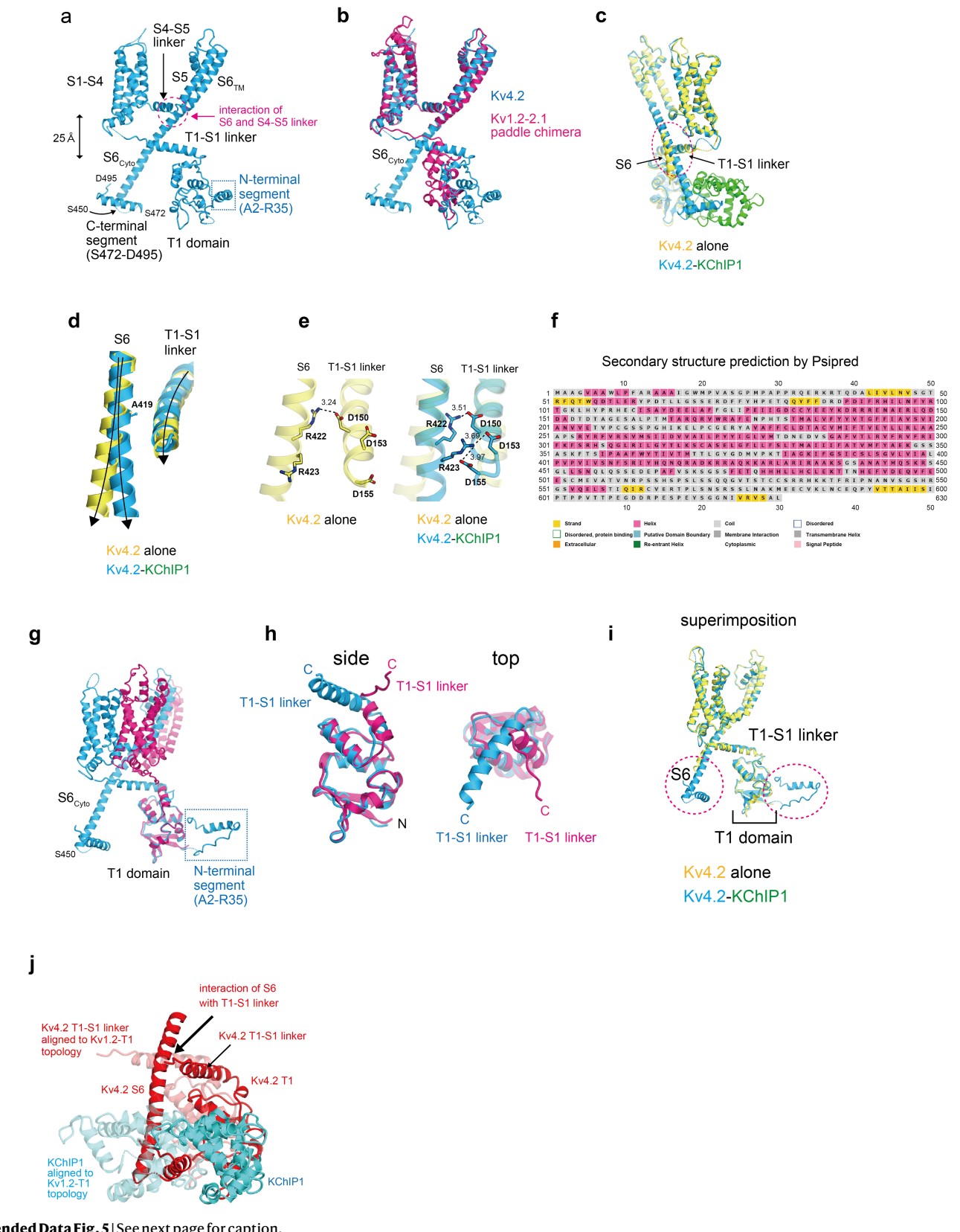

**Extended Data Fig. 5 |** See next page for caption.

**Extended Data Fig. 5 | Structures of the Kv4.2 α-subunit protomer in the presence and absence of KChIP1 and comparison with the Kv1.2-2.1 chimera. a**. Structure of the Kv4.2 α-subunit protomer in the Kv4.2–KChIP1 complex. The T1 domain is located beneath the transmembrane domain at a distance of 25 Å. Note that the transmembrane $S6_{TM}$ helix further extends toward the intracellular region, as indicated by the $S6_{cyto}$ (to S450) and C-terminal segment (S472-D495). Residues 496-630 are disordered and most of this region is predicted to lack secondary structure (**f**). **b**. Structural comparison of Kv4.2–KChIP1 with the Kv1.2-2.1 chimera, superimposed by transmembrane domains. The overall structure of the transmembrane domain of Kv4.2 is similar to the structures of the Kv1.2-2.1 paddle chimera. Note that the two T1 domains do not superimpose on each other. The Kv1.2-2.1 chimera does not have an intracellular S6 helix. **c**. The intracellular S6 helix of Kv4.2 alone bends at the interface on the T1-S1 linker (dashed ellipse) and is subsequently disordered. In contrast, the S6 helix of Kv4.2–KChIP1 complex extends straight toward KChIP1. **d**. Close-up view of the superimposed image in the dashed ellipse in (**c**). The intracellular S6 of Kv4.2 starts bending from A419 and extend away from the T1-S1 linker in the Kv4.2 alone. However, it keeps a close distance to T1-S1 linker without bending in the Kv4.2–KChIP1 complex. **e**. In the Kv4.2–KChIP1 complex, the intracellular S6 and T1-S1 linker interact via electrostatic interactions (right). In the Kv4.2 alone, the intracellular S6 largely dissociates from the T1-S1 linker (left). **f**. Prediction of the secondary structure of Kv4.2 by PSIPRED. Most of the region consisting of residues 496-630 is predicted to lack secondary structure. **g**. Structural comparison of Kv4.2–KChIP1 with the Kv1.2-2.1 chimera, superimposed by the T1 domains. The two T1 domains fit very well, but the transmembrane domains do not. **h**. Different directions of the C-terminal part of T1 domains, resulting in distinct topologies between Kv4.2 and the Kv1.2-2.1 chimera. Side (left) and top (right) views of the T1 domains are shown. **i**. Superimposition of the protomers of Kv4.2 alone and the Kv4.2-KChIP1 complex shows that the T1 domains of Kv4.2 overlap and retain the same topology in the presence and absence of KChIP1. **j**. When the Kv4.2-T1 domain is aligned with the Kv1.2-T1 topology (shown by translucent structure), the Kv4.2 S6 helix clashes with KChIP1 and does not interact with a T1-S1-linker.

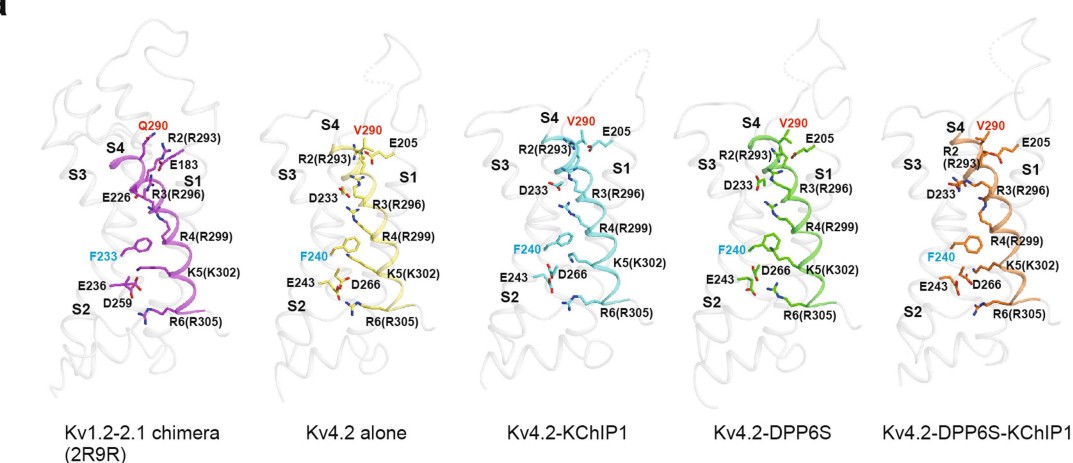

**a**

Kv1.2-2.1 chimera (2R9R)

Kv4.2 alone

Kv4.2-KChIP1

Kv4.2-DPP6S

Kv4.2-DPP6S-KChIP1

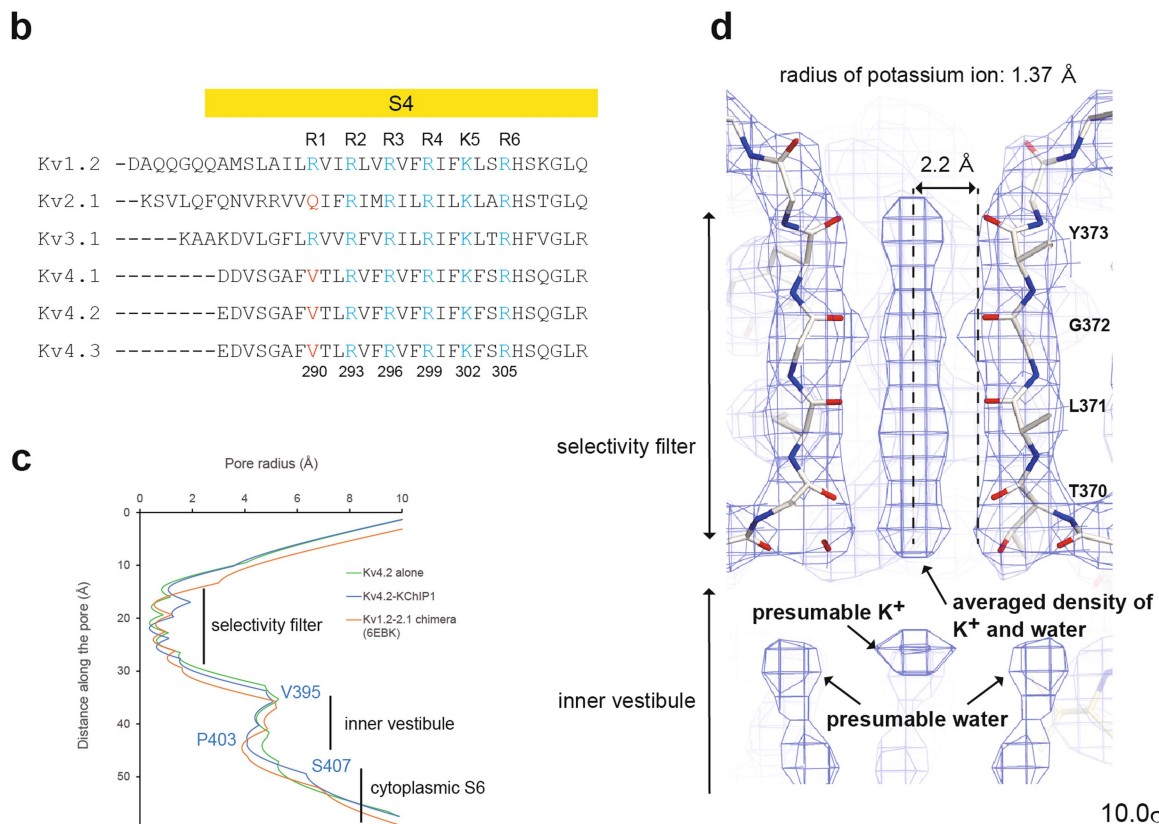

**b**

```
                            S4
                R1  R2  R3  R4  K5  R6
Kv1.2 -DAQQGQQAMSLAILRVIRLVRVFRIFKLSRHSKGLQ

Kv2.1 --KSVLQFQNVRRVVQIFRIMRILRILKLARHSTGLQ

Kv3.1 -----KAAKDVLGFLRVVRFVRILRIFKLTRHFVGLR

Kv4.1 -------DDVSGAFVTLRVFRVFRIFKFSRHSQGLR

Kv4.2 -------EDVSGAFVTLRVFRVFRIFKFSRHSQGLR

Kv4.3 -------EDVSGAFVTLRVFRVFRIFKFSRHSQGLR
                290  293  296  299  302 305
```

**c**

Pore radius (Å)

Distance along the pore (Å)

- Kv4.2 alone
- Kv4.2-KChIP1
- Kv1.2-2.1 chimera (6EBK)

selectivity filter

V395

P403

S407

inner vestibule

cytoplasmic S6

**d**

radius of potassium ion: 1.37 Å

2.2 Å

Y373

G372

L371

T370

selectivity filter

presumable K⁺

averaged density of K⁺ and water

inner vestibule

presumable water

$10.0\sigma$

**Extended Data Fig. 6 | Kv4.2 adopts the S4 up and S6 open conformation.**
**a**. Structures of the voltage sensors (S1-S4) from Kv1.2-2.1, Kv4.2 alone, Kv4.2–KChIP1, Kv4.2–DPP6S, and Kv4.2–DPP6S–KChIP1. S4 helices are coloured. Arg/Lys gating charges as well as other key residues are shown with side chains. The positions of positively charged amino acid residues in the S4 helix relative to a phenylalanine residue in the S2 helix indicates that the present S4 helix of Kv4.2 adopts the depolarized "up" conformation in all of four structures.
**b**. Alignment of S4 amino acid sequences among the closely related Kv1 to Kv4.
**c**. Radii of the pores of Kv4.2 alone, Kv4.2–KchIP1, and the Kv1.2-2.1 chimera, calculated using the HOLE program. **d**. The density map of the Kv4.2–KChIP1 complex at the selectivity filter shows the averaged densities of potassium ions and water. The S6 helix forming the pore adopts an open conformation, with the selectivity filter occupied by dehydrated K⁺ ions and water molecules, through the close interaction with the S4-S5 linker, as observed in the Kv1.2 structure[30] (Extended Data Figs. 5a, b). The previous electrophysiological studies reported that upon depolarization, Kv4s adopt the closed conformation (i.e. CSI) at all physiologically relevant membrane potentials within a cell[11–18] (Extended Data Fig. 1). This discrepancy could be attributed to the micelle which is likely to facilitate the open conformation. Similar inconsistent example was observed in the cryo-EM structure of the HCN channel in a hyperpolarized conformation in which the pore is closed while it is open within a cell[57].

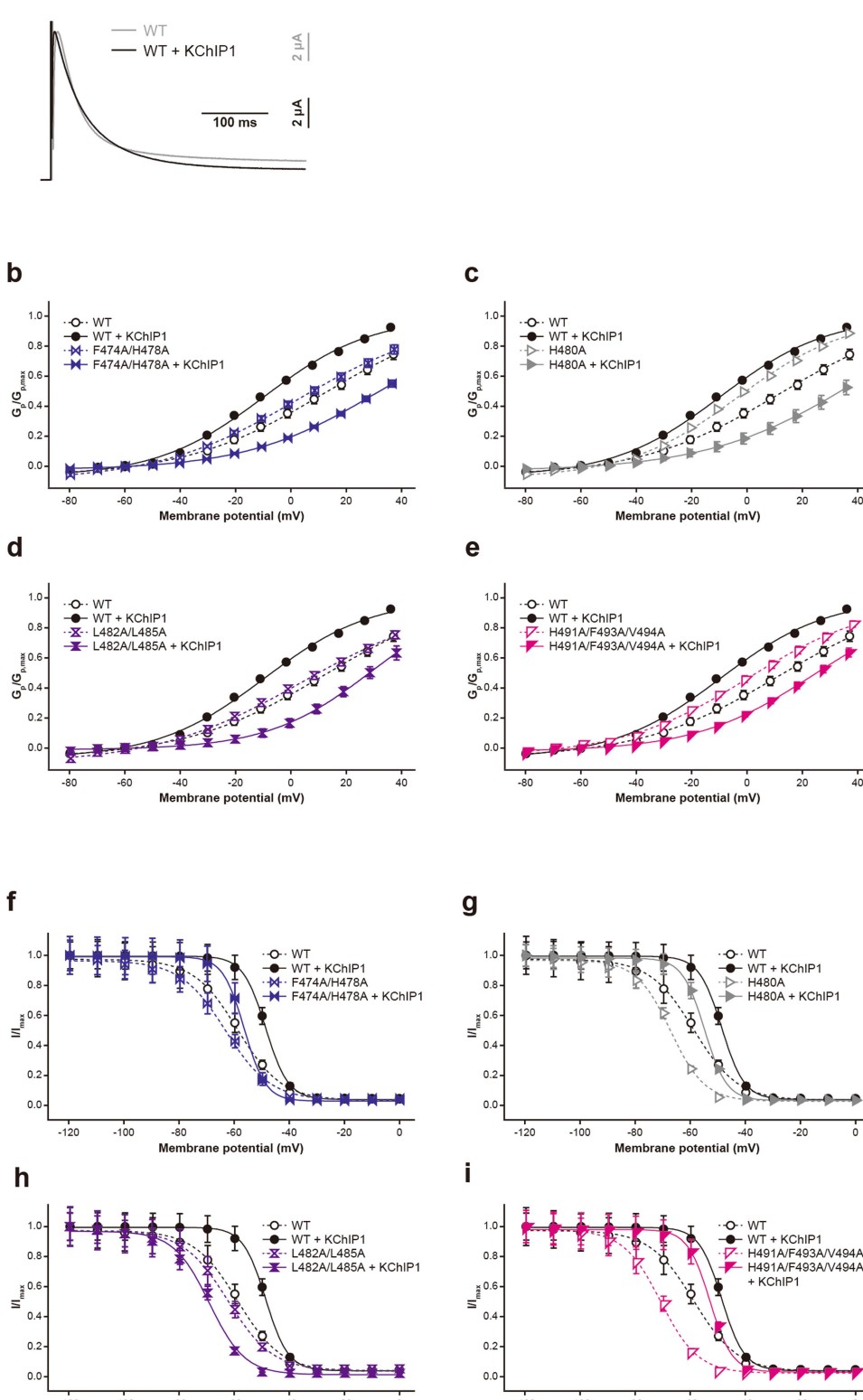

**Extended Data Fig. 7 | Influence of Kv4.2–KChIP1 interface mutations on KChIP1 modulation of activation and inactivation. a**. Normalized and superposed current traces of WT with (black) or without (gray) KChIP1 elicited by test pulses of 40 mV for the qualitative comparisons of inactivation kinetics ($n$ = 8 independent experiments). **b**–**e**. Peak conductance-Voltage (Gp-V) relationships of WT with (black circle) or without (white circle) KChIP1, and each mutant with (coloured symbol) or without (coloured open symbol) KChIP1 obtained from Supplementary Fig. 5a. Symbols and bars represent means ± s.e.m. ($n$ = 8). **f**–**i**. Comparison of the voltage-dependent prepulse inactivation for WT with (black circle) or without (white circle) KChIP1, and each mutant with (coloured symbol) or without (coloured open symbol) KChIP1 obtained from Supplementary Fig. 5c.

**a**

| Condition | $V_{1/2,act}$ (mV) | $z_{act}$ | $V_{1/2,inact}$ (mV) | $z_{inact}$ | $\tau_{rec}$ (ms) |
|---|---|---|---|---|---|
| **α subunit alone** | | | | | |
| Kv4.2 WT | 12.5 ± 2.3 | 1.1 ± 0.1 | -58.5 ± 0.3 | -3.2 ± 0.0 | 133.5 ± 7.6 |
| Kv4.2 T182W | 13.3 ± 2.0 | 1.4 ± 0.0* | -65.1 ± 0.5*** | -5.0 ± 0.2*** | 138.9 ± 5.9 |
| Kv4.2 V186W | 14.0 ± 1.7 | 1.3 ± 0.1 | -60.0 ± 0.5 | -5.0 ± 0.1*** | 55.6 ± 5.7*** |
| Kv4.2 V190W | 6.2 ± 2.5 | 1.3 ± 0.0 | -59.7 ± 0.5 | -4.1 ± 0.2*** | 84.8 ± 1.8*** |
| Kv4.2 F194W | 15.2 ± 2.0 | 1.2 ± 0.1 | -60.3 ± 0.9 | -3.3 ± 0.1 | 133.6 ± 6.7 |
| Kv4.2 A228W | 16.5 ± 1.6 | 1.1 ± 0.1 | -62.6 ± 0.7*** | -4.1 ± 0.1*** | 132.1 ± 10.0 |
| Kv4.2 C231W | 13.9 ± 1.6 | 1.4 ± 0.1** | -58.4 ± 0.5 | -5.4 ± 0.1*** | 161.9 ± 12.9 |
| Kv4.2 A235W | 5.9 ± 2.1 | 1.0 ± 0.1 | -70.1 ± 0.6*** | -3.4 ± 0.1 | 146.9 ± 10.9 |
| Kv4.2 F474A/H478A | 4.0 ± 2.4* | 1.0 ± 0.1 | -63.1 ± 0.6*** | -3.1 ± 0.1 | 149.2 ± 4.0 |
| Kv4.2 H480A | -3.1 ± 0.6*** | 1.3 ± 0.0 | -67.0 ± 0.5*** | -4.1 ± 0.0*** | 143.7 ± 8.2 |
| Kv4.2 L482A/L485A | 5.3 ± 2.0* | 1.0 ± 0.0 | -62.0 ± 0.7*** | -3.1 ± 0.1 | 156.0 ± 4.7 |
| Kv4.2 H491A/F493A/V494A | 1.8 ± 1.8** | 1.1 ± 0.1 | -70.7 ± 0.5*** | -3.8 ± 0.1*** | 155.3 ± 8.9 |
| | | | | | |
| **α subunit + KChIP1** | | | | | |
| Kv4.2 WT + KChIP1 WT | -9.0 ± 1.5 | 1.3 ± 0.0 | -48.5 ± 0.5 | -6.3 ± 0.2 | 42.0 ± 1.7 |
| Kv4.2 F474A/H478A + KChIP1 WT | 31.7 ± 1.6*** | 1.1 ± 0.1 | -56.6 ± 0.4*** | -6.8 ± 0.3 | 14.1 ± 0.9***,# |
| Kv4.2 H480A + KChIP1 WT | 32.7 ± 1.6*** | 1.1 ± 0.1 | -54.8 ± 0.2*** | -6.3 ± 0.1 | 19.4 ± 2.6***,# |
| Kv4.2 L482A/L485A + KChIP1 WT | 26.2 ± 3.1*** | 1.4 ± 0.1 | -68.8 ± 0.3*** | -3.9 ± 0.1*** | 98.4 ± 3.3*** |
| Kv4.2 H491A/F493A/V494A + KChIP1 WT | 24.9 ± 2.3*** | 1.2 ± 0.1 | -52.5 ± 0.6*** | -6.4 ± 0.2 | 44.0 ± 1.8 |
| | | | | | |
| **α subunit + DPP6S** | | | | | |
| Kv4.2 WT + DPP6S WT | -19.7 ± 2.1 | 1.3 ± 0.1 | -62.5 ± 0.3 | -6.3 ± 0.1 | 31.7 ± 1.1 |
| Kv4.2 T182W + DPP6S WT | -18.9 ± 1.4 | 1.4 ± 0.1 | -67.6 ± 0.5*** | -7.2 ± 0.1** | 25.3 ± 0.5 |
| Kv4.2 V186W + DPP6S WT | -20.3 ± 2.0 | 1.3 ± 0.1 | -70.0 ± 0.4*** | -5.8 ± 0.2 | 23.2 ± 1.1** |
| Kv4.2 V190W + DPP6S WT | -9.8 ± 1.6** | 1.3 ± 0.0 | -57.1 ± 0.5*** | -6.9 ± 0.2 | 44.8 ± 3.5*** |
| Kv4.2 F194W + DPP6S WT | -15.6 ± 1.5 | 1.3 ± 0.0 | -64.0 ± 0.7 | -5.8 ± 0.1 | 28.4 ± 0.8 |
| Kv4.2 A228W + DPP6S WT | -5.4 ± 1.9*** | 1.2 ± 0.0 | -67.1 ± 0.4*** | -6.4 ± 0.1 | 31.9 ± 0.8 |
| Kv4.2 C231W + DPP6S WT | -1.5 ± 2.0*** | 1.1 ± 0.0* | -69.7 ± 0.8*** | -7.1 ± 0.2** | 48.8 ± 2.6*** |
| Kv4.2 A235W + DPP6S WT | -20.8 ± 1.2 | 1.3 ± 0.0 | -67.9 ± 0.5*** | -7.2 ± 0.2*** | 30.6 ± 1.2 |

**b**

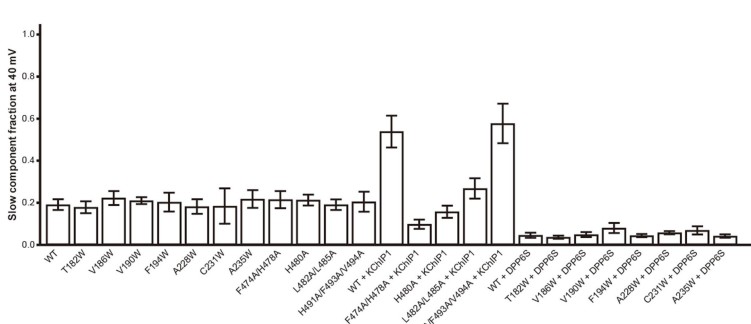

**c**

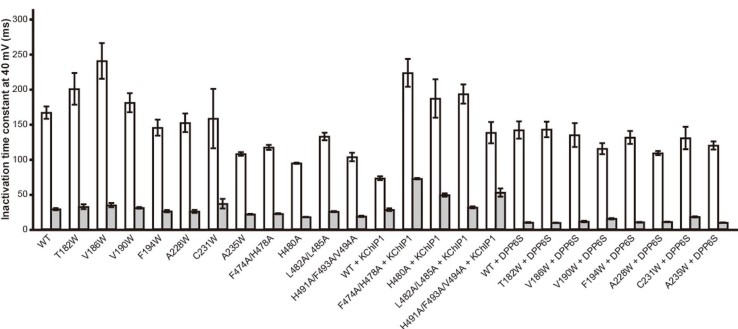

**Extended Data Fig. 8** | See next page for caption.

**Extended Data Fig. 8 | Summary of parameters for wild-type Kv4.2 and Kv4.2 mutants and inactivation kinetics obtained at 40 mV. a**. The number of the half-activation voltage ($V_{1/2,act}$) and effective charge ($z_{act}$) of the voltage-dependent activation experiments obtained by single Boltzmann fittings, the inactivation voltage ($V_{1/2,inact}$) and effective charge ($z_{inact}$) of the recovery from inactivation experiments obtained by single Boltzmann fittings, and the recovery time constant ($\tau_{rec}$) of the recovery from inactivation experiments obtained by single exponential fittings are listed as average ± s.e.m. Statistical significance was determined by Dunnett's test. *, **, and *** denote $P < 0.05$, $P < 0.01$, and $P < 0.001$ for each mutant compared to WT, for each mutant with KChIP1 compared to WT with KChIP1, and each mutant with DPP6S compared to WT with DPP6S. For Kv4.2 F474A/H478A with KChIP1 and Kv4.2 H480A with KChIP1 conditions, only data obtained using prepulses from 10 ms to 90 ms were used for single-exponential fits to calculate the recovery time constant owing to reduced fractional recovery at longer prepulses (marked as [#]). **b**. Fractional contribution of the slow inactivation component ($A_{slow}/(A_{slow} + A_{fast})$) at 40 mV. **c.** The slow ($\tau_{slow}$; white) and fast ($\tau_{fast}$; gray) inactivation time constants at 40 mV. Bars represent means ± s.e.m. ($n = 8$). Inactivation time constants ($\tau_{slow}$ and $\tau_{fast}$) and the corresponding amplitude ($A_{slow}$ and $A_{fast}$) were obtained by fitting the inactivation time course to a sum of two exponentials.

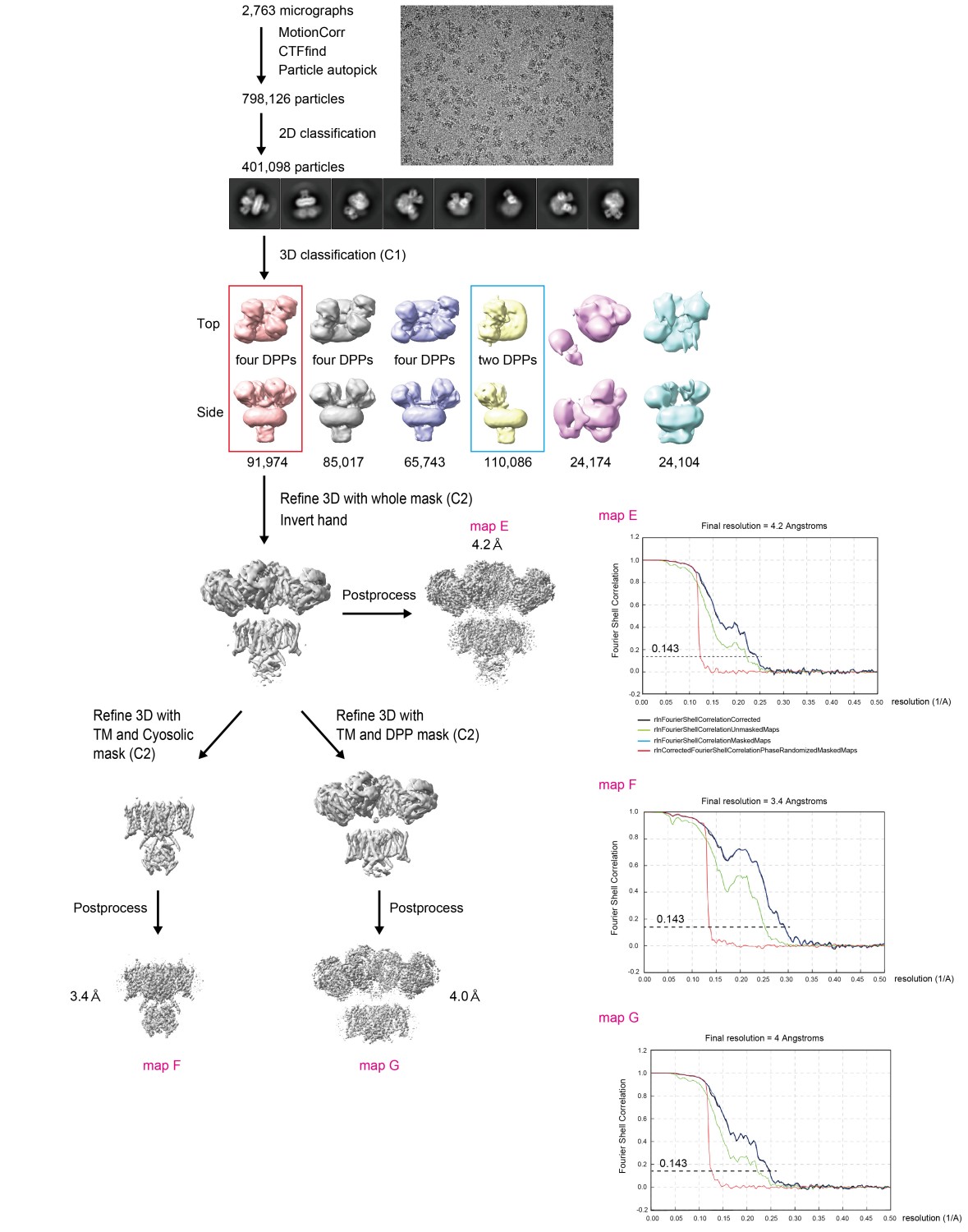

**Extended Data Fig. 9 | Cryo-EM micrograph, data processing and electron microscopy map of the Kv4.2–DPP6S complex.** Focused refinement at TM-intracellular part and TM-extracellular part was applied to improve the resolution of each part. The local resolutions of each density map and model building at TM region are shown in Supplementary Fig. 6.

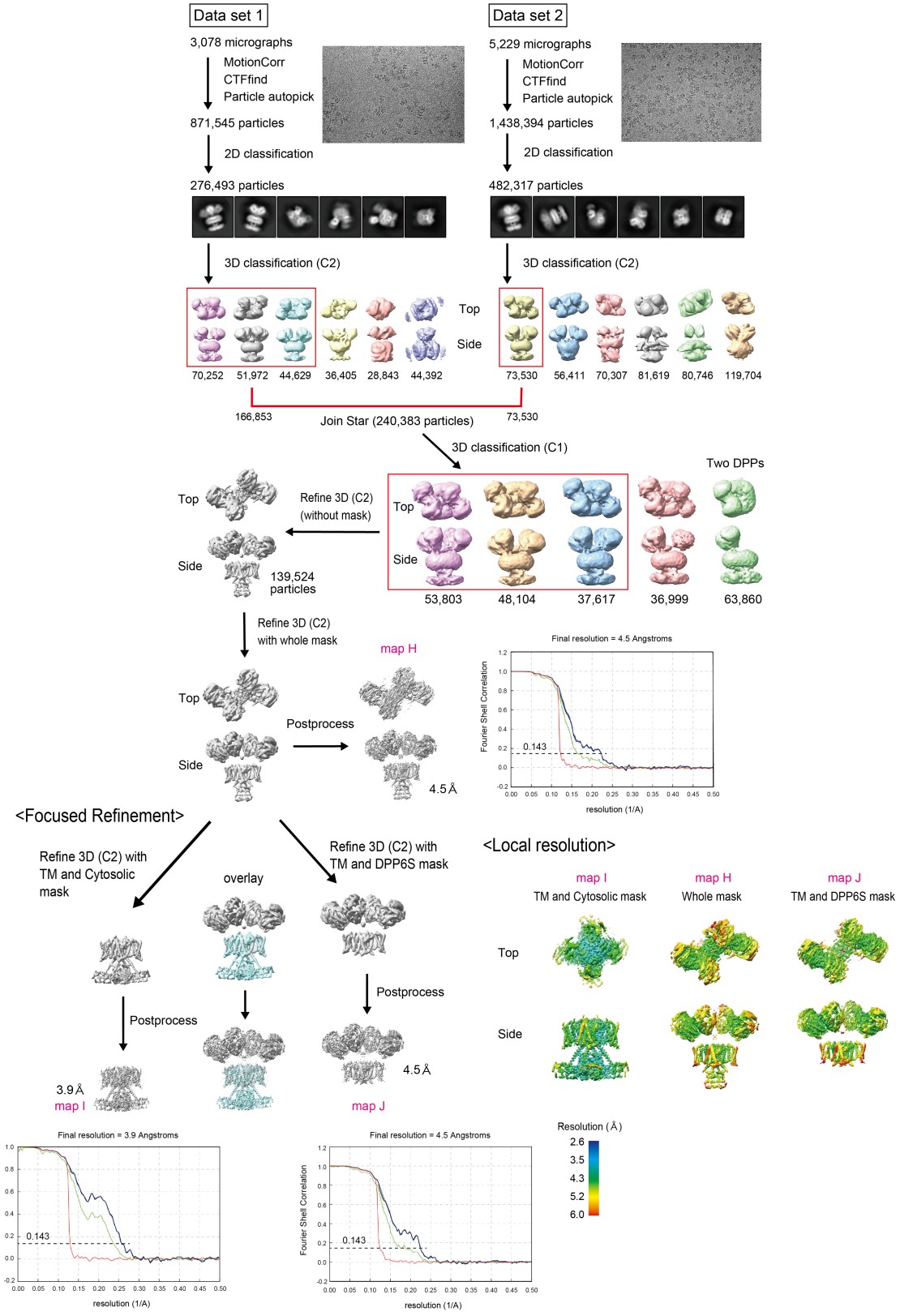

**Extended Data Fig. 10 | Cryo-EM micrograph, data processing and electron microscopy map of the Kv4.2–DPP6S–KChIP1 complex.** Focused refinement at TM-intracellular part and TM-extracellular part was applied to improve the resolution of each part. Model building at TM region is shown in Supplementary Fig. 7.

**a**

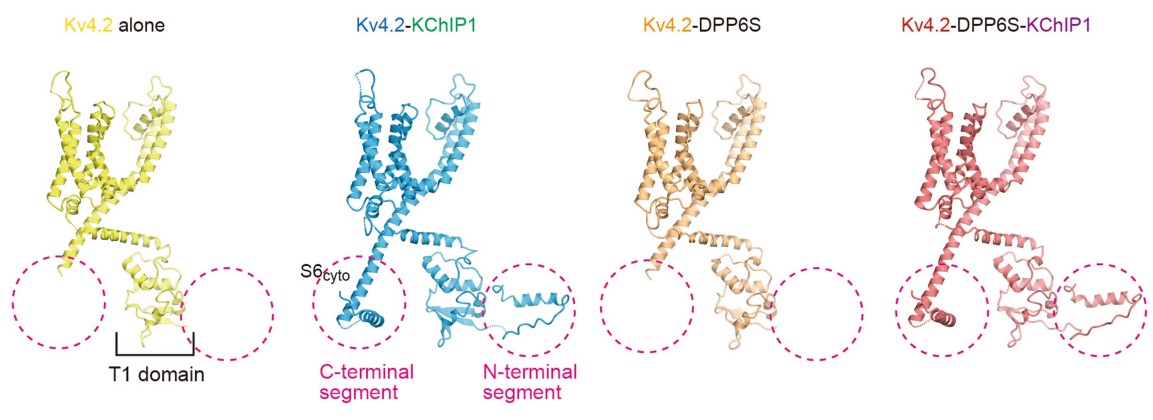

Kv4.2 alone Kv4.2-KChIP1 Kv4.2-DPP6S Kv4.2-DPP6S-KChIP1

S6$_{Cyto}$

T1 domain C-terminal segment N-terminal segment

**b**

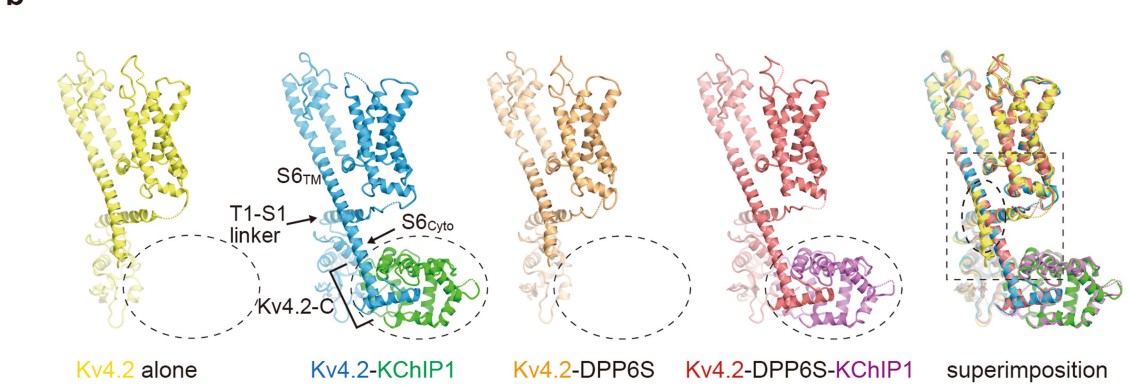

S6$_{TM}$

T1-S1 linker S6$_{Cyto}$

Kv4.2-C

Kv4.2 alone Kv4.2-KChIP1 Kv4.2-DPP6S Kv4.2-DPP6S-KChIP1 superimposition

**c**

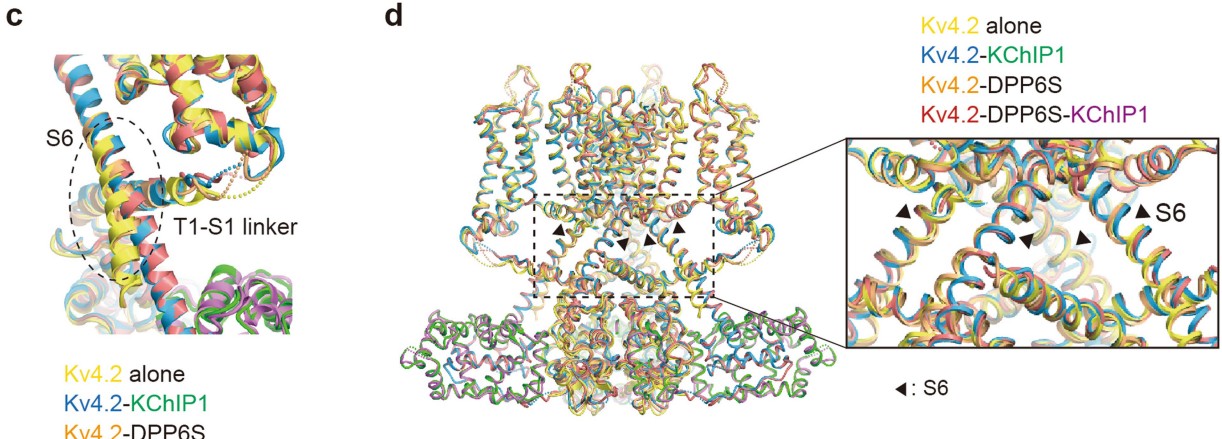

S6

T1-S1 linker

Kv4.2 alone
Kv4.2-KChIP1
Kv4.2-DPP6S
Kv4.2-DPP6S-KChIP1

**d**

Kv4.2 alone
Kv4.2-KChIP1
Kv4.2-DPP6S
Kv4.2-DPP6S-KChIP1

S6

◄ : S6

**Extended Data Fig. 11 | Structural comparison of N- and C-terminal conformations in the presence and absence of KChIP1. a.** Structural comparison of the Kv4.2 protomers from Kv4.2 alone, Kv4.2–KChIP1, Kv4.2–DPP6S, and Kv4.2–DPP6S–KChIP1, showing that both the N- and C-terminal regions are disordered in the absence of KChIP1 as observed in the structure of Kv4.2–DPP6S and Kv4.2 alone. Both terminal regions are resolved in the structure of Kv4.2–DPP6S–KChIP1 and Kv4.2–KChIP1. **b.** Comparison of the Kv4.2 S6 conformations. The intracellular S6 helices of Kv4.2–DPP6S and Kv4.2 alone bend at the interface on the T1-S1 linker (dashed ellipse in the superimposed image) and is subsequently disordered. By contrast, the S6 helices of Kv4.2–DPP6S–KChIP1 and Kv4.2–KChIP1 complexes extend straight toward KChIP1. **c.** Close-up view of the superimposed image in the dashed box in (b). The intracellular S6 of Kv4.2 bend and extend away from the T1-S1 linker in the Kv4.2–DPP6S complex and Kv4.2 alone. However, it keeps a close distance to T1-S1 linker without bending in the Kv4.2–DPP6S–KChIP1 and Kv4.2–KChIP1 complexes. **d.** Superimposition of the four Kv4.2 structures reveals that the S6 helices adopt an open conformation in all structures.

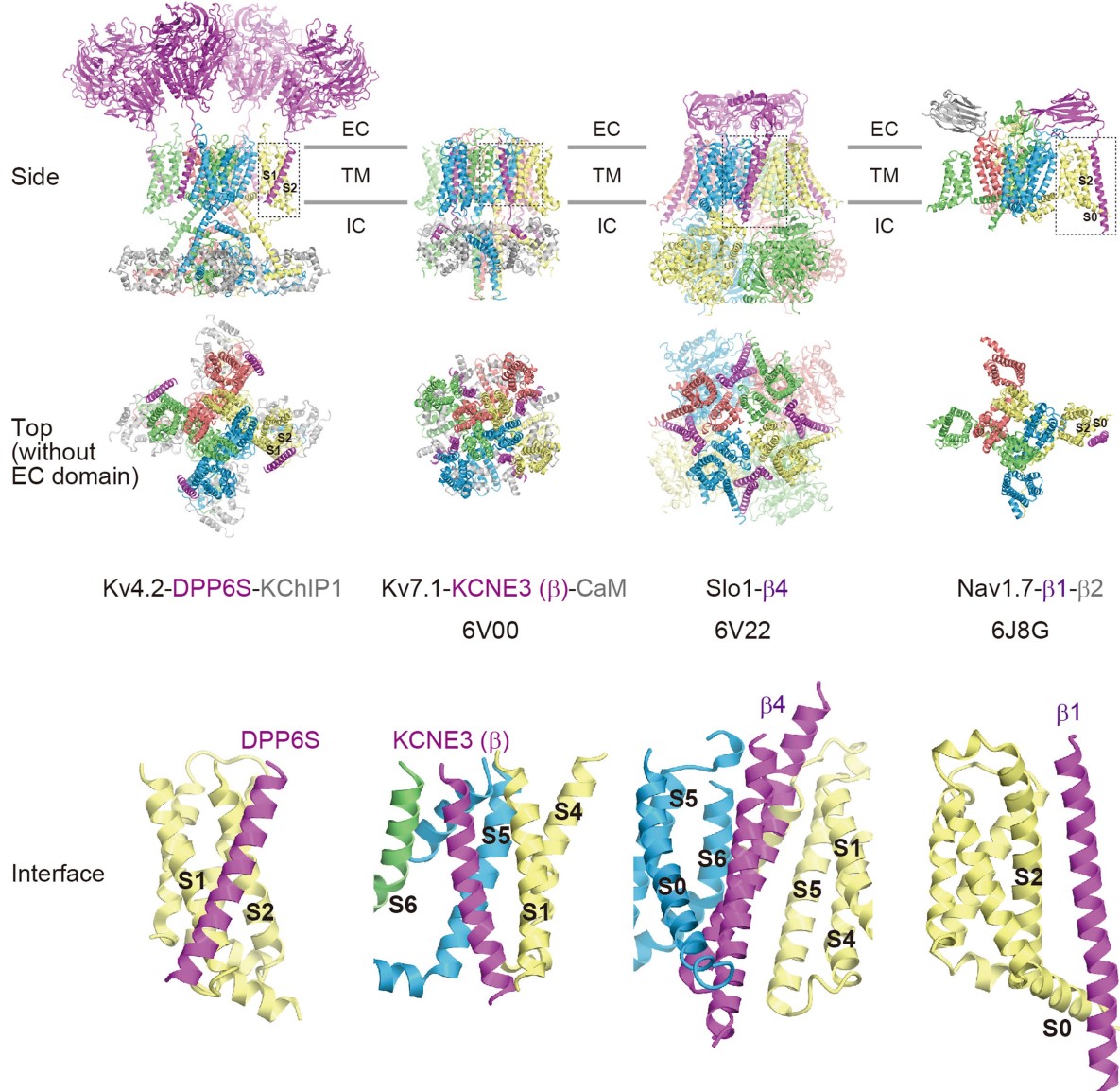

Side

EC
TM
IC

S1
S2

EC
TM
IC

EC
TM
IC

EC
TM
IC

S2
S0

Top
(without
EC domain)

S2
S1

S2 S0

Kv4.2-DPP6S-KChIP1

Kv7.1-KCNE3 (β)-CaM
6V00

Slo1-β4
6V22

Nav1.7-β1-β2
6J8G

Interface

DPP6S

S1

S2

KCNE3 (β)

S6

S5

S1

β4

S5

S6

S0

S4

S1

S5

S4

β1

S2

S0

**Extended Data Fig. 12 | Kv4-specific interaction with the transmembrane β-subunit revealed by structural comparisons with other potassium and sodium channel complexes.** Side and top views (without extracellular domain) of the Kv4.2–DPP6S–KChIP1, Kv7.1-KCNE3-CaM, Slo1-β4, and Nav1.7-β1-β2 complexes are shown from left to right. EC: extracellular region; TM: transmembrane region; IC: intracellular region. Dotted boxes in the side views highlight the interface of the channel α subunits and β subunits, and close-up views are shown (bottom). Note that a single DPP6S interacts with S1-S2 of a single voltage-sensing domain (VSD), whereas KCNE3 and β4 interact with the interface between two neighbouring α subunits in the Kv7.1 and Slo1 complexes, respectively. The interaction of Nav1.7 and β1 is rather similar to that of Kv4.2–DPP6S, in that a single β subunit interacts with a single VSD. However, the interaction of Kv4.2 and DPP6S is unique, because S1 of Nav1.7 is not involved in the interaction with β1.

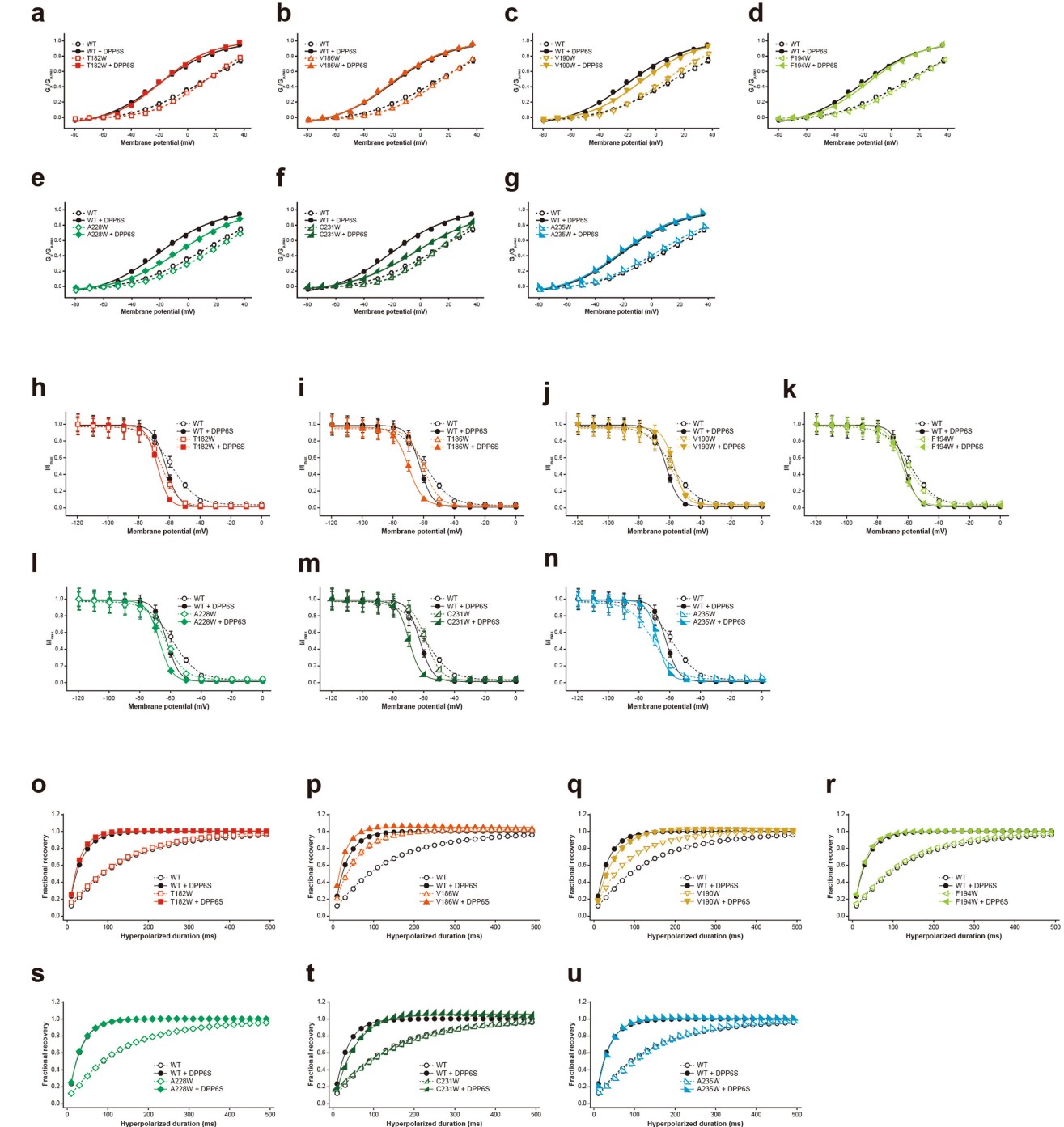

**Extended Data Fig. 13 | Influence of Kv4.2–DPP6S interface mutations on DPP6S modulation. a-g**. Peak conductance-Voltage (Gp-V) relationships of WT with (black circle) or without (white circle) DPP6S, and each mutant with (coloured symbol) or without (coloured open symbol) DPP6S obtained from Supplementary Fig. 9a. Symbols and bars represent means ± s.e.m. ($n$ = 8). **h-n**. Comparison of the voltage-dependent prepulse inactivation for WT with (black circle) or without (white circle) DPP6S, and each mutant with (coloured symbol) or without (coloured open symbol) DPP6S obtained from Supplementary Fig. 10. The fractional recovery at each point was determined

by normalizing the peak current amplitude of the test pulse by the test pulse after the prepulse of -120 mV and fitted with single Boltzmann functions. Symbols and bars represent means ± s.e.m. ($n$ = 8). **o-u**. Comparison of the recovery rate from inactivation among WT with (black circle) or without (white circle) DPP6S and each mutant with (coloured symbol) or without (coloured open symbol) DPP6S, obtained from Supplementary Fig. 11. The fractional recovery at each point was determined by normalizing the peak current amplitude of the test pulse by the amplitude of the prepulse. Symbols and bars represent means ± s.e.m. ($n$ = 8). Lines represent single-exponential fits.

**a**

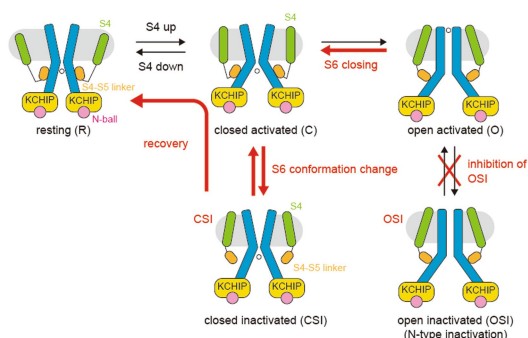

**b**

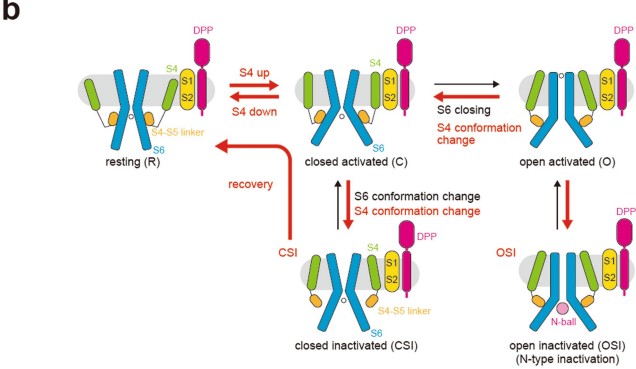

**c**

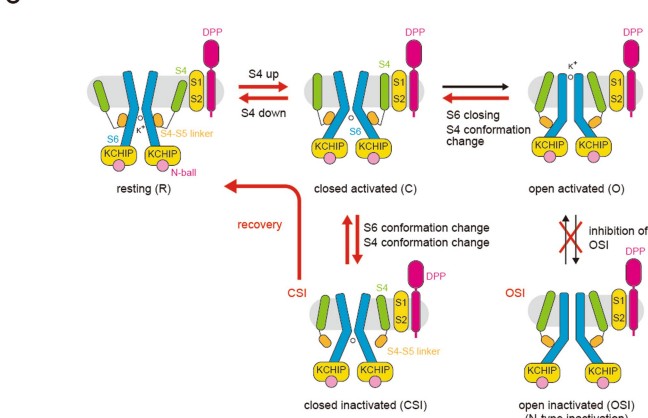

**Extended Data Fig. 14 | Model for Kv4 modulation by KChIP and DPP.**
**a**. Model for Kv4 modulation by KChIP. KChIPs capture the Kv4 N- and C-termini, thereby inhibiting open state inactivation (OSI). On the other hand, KChIPs stabilize the S6 conformation and might enable synchronized movement of the four S6 gating helices, thereby accelerating closed state inactivation and recovery. **b**. Model for Kv4 modulation by DPP. DPPs might stabilize the conformation of Kv4 S1-S2 and facilitate S4 conformational change, including S4 sliding up and down, thereby accelerating activation, inactivation, and recovery. **c**. Model for Kv4 modulation in the Kv4–DPP–KChIP ternary complex. KChIPs capture the Kv4 N- and C-termini of two adjacent subunits, thereby prevent open state inactivation (OSI). As a result, Kv4 ternary complex preferentially inactivates from a closed activated state (CSI). In addition, KChIPs stabilize the S6 conformation and accelerates S6 gating. DPPs stabilize the conformation of Kv4 S1-S2 and accelerates S4 conformation change including S4 movement upon membrane voltage shift. All together additive modulations by KChIPs and DPPs confer A-type current characterized as fast activation, fast closed state inactivation, and fast recovery.

## Extended Data Table 1 | Cryo-EM data collection, refinement and validation statistics

| | hKv4.2-KChIP1 (EMD-31433) (PDB 7F3F) | hKv4.2-KChIP1 (EMD-31009) (PDB 7E83) | hKv4.2-KChIP1 (EMD-31005) (PDB 7E7Z) |
|---|---|---|---|
| **Data collection and processing** | | | |
| Magnification | 105,000 | 105,000 | 105,000 |
| Voltage (kV) | 300 | 300 | 300 |
| Electron exposure (e–/Å²) | 50 | 50 | 50 |
| Defocus range (μm) | -0.8 to -1.6 | -0.8 to -1.6 | -0.8 to -1.6 |
| Pixel size (Å) | 0.83 | 0.83 | 0.83 |
| Symmetry imposed | C4 | C4 | C4 |
| Initial particle images (no.) | 1,105,082 | 1,105,082 | 1,105,082 |
| Final particle images (no.) | 434,296 | 434,296 | 286,241 |
| Map resolution (Å) | 3.1 | 3.1 | 3.2 |
| FSC threshold | 0.143 | 0.143 | 0.143 |
| Map resolution range (Å) | 2.9-8.8 | 2.9-8.8 | 3.0-8.3 |
| | | | |
| **Refinement** | | | |
| Initial model used (PDB code) | | | |
| Model resolution (Å) | 3.4 | 3.5 | 5.9 |
| FSC threshold | 0.5 | 0.5 | 0.5 |
| Model resolution range (Å) | | | |
| Map sharpening $B$ factor (Å²) | -179.0 | -179.0 | -177.3 |
| Model composition | | | |
| Non-hydrogen atoms | 20,580 | 12,704 | 7,808 |
| Protein residues | 2,484 | 1,496 | 976 |
| Ligands | | | |
| $B$ factors (Å²) | | | |
| Protein | 22.3 | 35.7 | 34.9 |
| Ligand | | | |
| R.m.s. deviations | | | |
| Bond lengths (Å) | 0.002 | 0.003 | 0.002 |
| Bond angles (°) | 0.535 | 0.593 | 0.495 |
| Validation | | | |
| MolProbity score | 1.86 | 1.92 | 1.90 |
| Clashscore | 7.04 | 7.05 | 9.80 |
| Poor rotamers (%) | 0.14 | 0.00 | 0.00 |
| Ramachandran plot | | | |
| Favored (%) | 92.34 | 90.71 | 94.36 |

| | hKv4.2-DPP6S (EMD-31013) (PDB 7E8B) | hKv4.2-DPP6S (EMD-31011) (PDB 7E87) | hKv4.2-DPP6S (EMD-31012) (PDB 7E89) |
|---|---|---|---|
| **Data collection and processing** | | | |
| Magnification | 105,000 | 105,000 | 105,000 |
| Voltage (kV) | 300 | 300 | 300 |
| Electron exposure (e–/Å²) | 50 | 50 | 50 |
| Defocus range (μm) | -0.8 to -1.6 | -0.8 to -1.6 | -0.8 to -1.6 |
| Pixel size (Å) | 0.83 | 0.83 | 0.83 |
| Symmetry imposed | C2 | C2 | C2 |
| Initial particle images (no.) | 798,126 | 798,126 | 798,126 |
| Final particle images (no.) | 91,974 | 91,974 | 91,974 |
| Map resolution (Å) | 4.2 | 3.4 | 4.0 |
| FSC threshold | 0.143 | 0.143 | 0.143 |
| Map resolution range (Å) | 3.7-12.5 | 3.1-12.5 | 3.7-12.6 |
| | | | |
| **Refinement** | | | |
| Initial model used (PDB code) | 1XFD | 1XFD | 1XFD |
| Model resolution (Å) | 6.98 | 5.95 | 4.45 |
| FSC threshold | 0.5 | 0.5 | 0.5 |
| Model resolution range (Å) | | | |
| Map sharpening $B$ factor (Å²) | -202.6 | -116.4 | -178.9 |
| Model composition | | | |
| Non-hydrogen atoms | 36,832 | 13,304 | 23,528 |
| Protein residues | 4,548 | 1,632 | 2,908 |
| Ligands | | | |
| $B$ factors (Å²) | | | |
| Protein | 80.9 | 97.9 | 113 |
| Ligand | | | |
| R.m.s. deviations | | | |
| Bond lengths (Å) | 0.003 | 0.003 | 0.003 |
| Bond angles (°) | 0.585 | 0.632 | 0.601 |
| Validation | | | |
| MolProbity score | 2.14 | 2.09 | 2.34 |
| Clashscore | 15.01 | 13.44 | 19.30 |
| Poor rotamers (%) | 0.05 | 0.00 | 0.04 |
| Ramachandran plot | | | |
| Favored (%) | 92.70 | 93.01 | 89.75 |

| | hKv4.2-DPP6S-KChIP1 (EMD-31019) (PDB 7E8H) | hKv4.2-DPP6S-KChIP1 (EMD-31016) (PDB 7E8E) | hKv4.2-DPP6S-KChIP1 (EMD-31018) (PDB 7E8G) | hKv4.2 alone (EMD-31399) (PDB 7F0J) |
|---|---|---|---|---|
| **Data collection and processing** | | | | |
| Magnification | 105,000 | 105,000 | 105,000 | 105,000 |
| Voltage (kV) | 300 | 300 | 300 | 300 |
| Electron exposure (e–/Å²) | 50 | 50 | 50 | 50 |
| Defocus range (μm) | -0.8 to -1.6 | -0.8 to -1.6 | -0.8 to -1.6 | -0.8 to -1.6 |
| Pixel size (Å) | 0.83 | 0.83 | 0.83 | 0.83 |
| Symmetry imposed | C2 | C2 | C2 | C4 |
| Initial particle images (no.) | 2,309,939 | 2,309,939 | 2,309,939 | 1,507,985 |
| Final particle images (no.) | 139,524 | 139,524 | 139,524 | 70,625 |
| Map resolution (Å) | 4.5 | 3.9 | 4.5 | 2.9 |
| FSC threshold | 0.143 | 0.143 | 0.143 | 0.143 |
| Map resolution range (Å) | 4.0-13.0 | 3.5-12.5 | 4.0-15.3 | 2.8-6.8 |
| | | | | |
| **Refinement** | | | | |
| Initial model used (PDB code) | 1XFD | 1XFD | 1XFD | |
| Model resolution (Å) | 6.91 | 4.01 | 6.47 | 3.25 |
| FSC threshold | 0.5 | 0.5 | 0.5 | 0.5 |
| Model resolution range (Å) | | | | |
| Map sharpening $B$ factor (Å²) | -225.9 | -189.392 | -221.152 | -94.449 |
| Model composition | | | | |
| Non-hydrogen atoms | 44,894 | 21,380 | 23,528 | 12,424 |
| Protein residues | 5,516 | 2,606 | 2,908 | 1,520 |
| Ligands | | | | |
| $B$ factors (Å²) | | | | |
| Protein | 101 | 69.1 | 149 | 33.5 |
| Ligand | | | | |
| R.m.s. deviations | | | | |
| Bond lengths (Å) | 0.002 | 0.004 | 0.002 | 0.002 |
| Bond angles (°) | 0.602 | 0.660 | 0.534 | 0.531 |
| Validation | | | | |
| MolProbity score | 2.20 | 2.16 | 2.22 | 1.92 |
| Clashscore | 18.10 | 15.86 | 16.82 | 9.61 |
| Poor rotamers (%) | 0.00 | 0.04 | 0.08 | 0.00 |
| Ramachandran plot | | | | |

# Reporting Summary

## Statistics

For all statistical analyses, confirm that the following items are present in the figure legend, table legend, main text, or Methods section.

| n/a | Confirmed | |
|---|---|---|
| ☐ | ☒ | The exact sample size (*n*) for each experimental group/condition, given as a discrete number and unit of measurement |
| ☐ | ☒ | A statement on whether measurements were taken from distinct samples or whether the same sample was measured repeatedly |
| ☐ | ☒ | The statistical test(s) used AND whether they are one- or two-sided<br>*Only common tests should be described solely by name; describe more complex techniques in the Methods section.* |
| ☒ | ☐ | A description of all covariates tested |
| ☒ | ☐ | A description of any assumptions or corrections, such as tests of normality and adjustment for multiple comparisons |
| ☐ | ☒ | A full description of the statistical parameters including central tendency (e.g. means) or other basic estimates (e.g. regression coefficient) AND variation (e.g. standard deviation) or associated estimates of uncertainty (e.g. confidence intervals) |
| ☐ | ☒ | For null hypothesis testing, the test statistic (e.g. *F*, *t*, *r*) with confidence intervals, effect sizes, degrees of freedom and *P* value noted<br>*Give P values as exact values whenever suitable.* |
| ☒ | ☐ | For Bayesian analysis, information on the choice of priors and Markov chain Monte Carlo settings |
| ☒ | ☐ | For hierarchical and complex designs, identification of the appropriate level for tests and full reporting of outcomes |
| ☒ | ☐ | Estimates of effect sizes (e.g. Cohen's *d*, Pearson's *r*), indicating how they were calculated |

*Our web collection on statistics for biologists contains articles on many of the points above.*

## Software and code

Policy information about availability of computer code

| Data collection | EPU (version 1.19), Serial EM (version 3.7.10), Clampex (version 10.7) |
|---|---|
| Data analysis | RELION (version 3.0 and version 3.1), PHENIX (version 1.19), MOLREP (version 11.7), COOT (version 0.8.9.1), UCSF Chimera (version 1.14), CueMol2 (http://www.cuemol.org/ version 2.2.3.443), HOLE (version 2.2.004), Clampfit (version 10.7), EZR (version 1.54) |

For manuscripts utilizing custom algorithms or software that are central to the research but not yet described in published literature, software must be made available to editors and reviewers. We strongly encourage code deposition in a community repository (e.g. GitHub). See the Nature Portfolio guidelines for submitting code & software for further information.

## Data

Policy information about availability of data

All manuscripts must include a data availability statement. This statement should provide the following information, where applicable:

- Accession codes, unique identifiers, or web links for publicly available datasets
- A description of any restrictions on data availability
- For clinical datasets or third party data, please ensure that the statement adheres to our policy

The cryo-EM density maps and atomic coordinates have been deposited in the Electron Microscopy Data Bank. The accession codes for the maps are EMD-31010 (Kv4.2-KChIP1-whole (map A)), EMD-31009, (Kv4.2-KChIP1-whole (map A)), EMD-31005 (Kv4.2-KChIP1-TM (map B)) , EMD-31013 (Kv4.2-DPP6S-whole (map E)), EMD-31011 (Kv4.2-DPP6S-TM and Cyto (map F)), EMD-31012 (Kv4.2-DPP6S-TM and EC (map G)), EMD-31019 (Kv4.2-DPP6S-KChIP1-whole (map H)), EMD-31016 (Kv4.2-DPP6S-KChIP1 (TM and Cyto (map I)), EMD-31018 (Kv4.2-DPP6S-KChIP1-TM and EC (map J)), and EMD-31399 (Kv4.2 alone (map X)). The accession codes for the coordinates are 7E84 (Kv4.2-KChIP1-whole), 7E83 (Kv4.2-KChIP1-Cyto), 7E7Z (Kv4.2-KChIP1-TM), 7E8B (Kv4.2-DPP6S-whole), 7E87 (Kv4.2-DPP6S-TM and Cyto),

March 2021

7E89 (Kv4.2-DPP6S-EC), 7E8H (Kv4.2-DPP6S-KChIP1-whole), 7E8E (Kv4.2-DPP6S-KChIP1-TM and Cyto), 7E8G (Kv4.2-DPP6S-KChIP1-EC), and 7F0J (Kv4.2 alone). For detail, see also Extended Data Table 1, Extended Data Fig. 3, 4, 8, 9.

# Field-specific reporting

Please select the one below that is the best fit for your research. If you are not sure, read the appropriate sections before making your selection.

☒ Life sciences ☐ Behavioural & social sciences ☐ Ecological, evolutionary & environmental sciences

For a reference copy of the document with all sections, see nature.com/documents/nr-reporting-summary-flat.pdf

# Life sciences study design

All studies must disclose on these points even when the disclosure is negative.

| | |
|---|---|
| Sample size | No statistical method was used to determine the sample size. For cryo-EM analyses, sample sizes were determined by the availability of microscope time and the number of particles on electron microscopy grids enough to obtain a structure at the reported resolution. For electrophysiological analyses, sample sizes were determined based on the previous reports of this type of study and the reproducibility of results across independent experiments. |
| Data exclusions | For cryo-EM analyses, particles that did not contribute to improving map quality were excluded following the standard classification procedures in RELION. This is standard practice for structure determination by cryo-EM. For electrophysiological analyses, recordings that contain leak or endogenous currents were excluded. This is standard practice in electrophysiology. |
| Replication | For cryo-EM analyses, related experiments including FSEC, purification, and SDS-PAGE were reproduced at least two times and structure determination was completed once. For electrophysiological analyses, all data sets were pooled from at least two independent oocyte batches. |
| Randomization | For cryo-EM analyses, particles were randomly assigned to half-maps for resolution determination following the standard procedures in RELION. For electrophysiological analyses, randomization was not performed since samples were not divided into two or more groups. |
| Blinding | For cryo-EM analyses, blinding was not applicable since this type of studies does not use group allocation. For electrophysiological analyses, blinding was not applied since it was not technically or practically feasible to do so. |

# Reporting for specific materials, systems and methods

We require information from authors about some types of materials, experimental systems and methods used in many studies. Here, indicate whether each material, system or method listed is relevant to your study. If you are not sure if a list item applies to your research, read the appropriate section before selecting a response.

## Materials & experimental systems

| n/a | Involved in the study |
|---|---|
| ☒ ☐ | Antibodies |
| ☐ ☒ | Eukaryotic cell lines |
| ☒ ☐ | Palaeontology and archaeology |
| ☐ ☒ | Animals and other organisms |
| ☒ ☐ | Human research participants |
| ☒ ☐ | Clinical data |
| ☒ ☐ | Dual use research of concern |

## Methods

| n/a | Involved in the study |
|---|---|
| ☒ ☐ | ChIP-seq |
| ☒ ☐ | Flow cytometry |
| ☒ ☐ | MRI-based neuroimaging |

# Eukaryotic cell lines

Policy information about cell lines

| | |
|---|---|
| Cell line source(s) | HEK293S GnTI- (ATCC, Cat.#CRL-3022), Sf9 (ATCC, Cat.#CRL-1711) |
| Authentication | The cell lines listed above were purchased from ATCC Cell lines and no further authentication was performed. |
| Mycoplasma contamination | Not performed |
| Commonly misidentified lines (See ICLAC register) | HEK cells are listed in the register but it does not specify which type of HEK strains. Our secondary HEL293S GnTI- cell lines was purchased by from ATCC, where they validated. |

## Animals and other organisms

Policy information about studies involving animals; ARRIVE guidelines recommended for reporting animal research

| | |
|---|---|
| Laboratory animals | Adult female Xenopus laevis were used to obtain oocytes. |
| Wild animals | The study did not involve wild animals. |
| Field-collected samples | The study did not involve samples collected from the field. |
| Ethics oversight | All electrophysiological experiments were approved by the Animal Care Committee of Jichi Medical University (Japan) under the protocol no. 18027-03 and were performed following the institutional guidelines. |

Note that full information on the approval of the study protocol must also be provided in the manuscript.

