## [Peer Review File · Nature]

Manuscript Title: Structural basis of gating modulation of Kv4 channel complexes

Editorial Notes:

Redactions – Third Party Material

Reviewer Comments & Author Rebuttals

Reviewer Reports on the Initial Version:

Referees' comments:

Referee #1 (Remarks to the Author):

The manuscript by Kise et al. reports an extraordinary achievement. Using cryo-EM, these authors solved the novel dodecameric structure of a voltage-gated K (Kv) channel at the atomic level. This macromolecular complex is one of the most elaborate K channel structures solved to date. It includes the Kv4.2 pore forming alpha subunit, and two ancillary beta subunits, KChIP1 and DPP6S. Moreover, to infer modulatory mechanisms they also solved octameric structures of the Kv4.2-KChIP1 complex and the Kv4.2-DPP6S complex. From the biological and physiological perspectives, this discovery is highly relevant because the Kv4.2-KChIP1-DPP6S dodecameric complex and other complexes made of similar subunits underlie the subthreshold-activating fast-inactivating Kv currents in the nervous system. These Kv currents are conserved across the animal kingdom (from jelly fish and nematodes, to mice and humans) and play key roles regulating latency to the first spike, slow repetitive spiking, action potential backpropagation, etc.

What is perhaps most remarkable from this study is that it begins to illuminate (literally!) for the first time how these interesting and important Kv channels work. Current evidence strongly suggests that the Kv4.2 dodecameric complex and similar complexes do not inactivate by the classical N-type and C-type mechanisms first reported for Shaker-type Kv1 channels that undergo open-state inactivation. Consequently, work published in the past 25 years by a handful of groups generated working hypotheses that attempt to explain the mechanisms of inactivation gating of Kv4 channels in various configurations. Collectively, previous work provided sound evidence to suggest that Kv4 channels undergo closed-state inactivation involving the S4S5 linker and the distal S6 regions that are canonically associated with activation gating. However, without direct visualization, these ideas and other intriguing features of Kv4 gating remained as hypothetical models. Fortunately, the beautiful structures solved by Kise et al. provide at last direct atomic-level visualization of unique Kv4 channel features previously hypothesized by others:

- 1) The proximal N- and C-termini interact and are sequestered by KChIP1
- 2) Kv4 channel gating involves novel interactions between the intracellular activation gate and the T1-S1 linker
- 3) Native Kv4 channels exist as dodecameric complexes including KChIPs and DPPs
- 4) The voltage sensing domain is involved in closed-state inactivation and acts as the docking site for DPPs
- 5) Potentially weaker interactions between the S4S5 linker and the distal segment of S6 are responsible for the activation gate collapse that underlies closed-state inactivation

In brief, this study represents a major breakthrough that many researcher in the ion channel field where hoping for. The quality of the structural work is outstanding and the manuscript is generally well written. Also, the authors provide enough information to assess the robustness of the results and eventual replication of the experiments. I have some specific comments and suggestions that would improve the rigor of the electrophysiological analyses and the accuracy of the cited work from others.

Specific Comments

1) In my view, it would be more appropriate to use this title: "Structural basis of gating modulation of the Kv4 macromolecular channel complex".

2) The performance of the electrophysiological experiments to characterize mutations is sound. However, the description and interpretation of the results is entirely qualitative, and some analyses may be misleading.

For instance, to assess the voltage dependence of activation, the authors only display the current/voltage relations and arbitrarily normalize to the current amplitude at +40 mV. This normalization is misleading because changes may occur not only in terms of the voltage range over which the current operates; changes in the apparent sensitivity to voltage can also occur. These changes cannot be unambiguously determined from I/V curves. Authors should at a minimum consider chord conductance/voltage curves and nth-order Boltzmann fits, which is an accepted standard in the field.

Steady-state (pre-pulse) inactivation is a highly significant property of Kv4 channels, which the authors did not assess, but can provide significant information about the impact of the investigated mutations on voltage dependence and kinetics of inactivation. Also, the authors qualitatively refer to changes in inactivation kinetics, but provide no measurements to characterize these changes (e.g., weighted time constants of inactivation and recovery from inactivation).

Please note that apparent changes in inactivation kinetics at a given voltage that is not sufficient to reach G_{max} may be secondary to the shifts in the voltage-dependence of activation.

I understand that a detailed biophysical characterization of the reported mutations is outside the scope of this study. What I am suggesting provides the most basic semiquantitative characterization that would help interested readers better understand the reported effects, their interpretations and structural implications. In my view, the remarkable and exciting structural discoveries of this study deserve a minimum of rigor when assessing the electrophysiological correlates. Given the capabilities of the authors, the recommended analyses may not represent a major endeavor. They may, however, improve the study in a significant way.

3) There are a few places where important citations are misquoted or incomplete. Specific instances by line # are listed below.

Lines 69-72; Lines 482-490: The most conclusive evidence for CSI in tetrameric, octameric and dodecameric Kv4 channels was provided in these reports:

- Fineberg, J. D., Szanto, T. G., Panyi, G. & Covarrubias, M. Closed-state inactivation involving an internal gate in Kv4.1 channels modulates pore blockade by intracellular quaternary ammonium ions. *Sci Rep* 6, 31131, doi:10.1038/srep31131 (2016).
- Fineberg, J. D., Ritter, D. M. & Covarrubias, M. Modeling-independent elucidation of inactivation pathways in recombinant and native A-type Kv channels. *J.Gen.Physiol* 140, 513-527 (2012).
- Dougherty, K., Santiago-Castillo, J. A. & Covarrubias, M. Gating charge immobilization in Kv4.2 channels: the basis of closed-state inactivation. *J.Gen.Physiol* 131, 257-273 (2008).
- Kaulin, Y. A., Santiago-Castillo, J. A., Rocha, C. A. & Covarrubias, M. Mechanism of the modulation of Kv4:KChIP-1 channels by external K⁺. *Biophysical Journal* 94, 1241-1251 (2008).

Lines 173-189: Previous studies listed below demonstrated that the T1-S1 linker of Kv4 dodecameric channels undergoes major conformational shifts tightly coupled to movements of the S6 tail (i.e., the activation gate). These rearrangements dramatically change the accessibility to three Cys residues in the T1 Zn binding site. It is surprising that the authors made no reference to these observations and do not describe important structural changes that must have occurred (or not?) in this region of the solved structure. Was the site occupied by Zn? The site is expected to be Zn free because, otherwise, tightly bound Zn would protect against Cys modification.

- Wang, G. & Covarrubias, M. Voltage-dependent gating rearrangements in the intracellular T1-T1 interface of a K⁺ channel. *J.Gen.Physiol* 127, 391-400(2006).
- Wang, G. et al. Functionally active t1-t1 interfaces revealed by the accessibility of intracellular thiolate groups in kv4 channels. *J.Gen.Physiol* 126, 55-69 (2005).

Lines 300-306; Lines 482-490; Lines 492-493: Amarillo et al. (2008) demonstrated that expression of precisely the dodecameric complex solved by authors closely recapitulates the A-type current endogenously expressed in cerebellar granule cells. Also, this study characterized the biophysical properties of octameric complexes solved by the authors and postulated gating models that explain gating of all configurations – Kv4.2 alone, Kv4.2+KChIP1, Kv4.2+DPP6S and Kv4.2+KChIP1+DPP6S. Furthermore, it should be noted that DPP6S also determines the unitary conductance of the native A-type K channel in cerebellar granule neurons, and that this is electrostatically dictated by the cytoplasmic N-terminal region of DPP6S (Kaulin et al., 2009).

- Amarillo, Y. et al. Ternary Kv4.2 channels recapitulate voltage-dependent inactivation kinetics of A-type K⁺ channels in cerebellar granule neurons. *J.Physiol* 586, 2093-2106 (2008).
- Kaulin, Y. A. et al. The dipeptidyl-peptidase-like protein DPP6 determines the unitary conductance of neuronal Kv4.2 channels. *The Journal of Neuroscience* 29, 3242-3251, doi:10.1523/JNEUROSCI.4767-08.2009 (2009).

4) I wonder whether the unidentified density (Extended Data Fig. 10) may reflect the structural instability of the pore cavity rather than phospholipids or hydrated K ions. The single channel properties of Kv4 channels demonstrate a highly unstable opening state with frequent sojourns to partially and fully closed states before the channel finally enters an absorbing inactivated state (see Kaulin et al., 2009, cited above).

5) A cartoon that shows how dodecameric Kv4 channels may gate is not shown, and should be included in the main document, since this is the physiologically relevant complex. The cartoons of octameric Kv4 channels, currently included in the main document, may be presented as Extended Data, if there is limited space in the main document.

6) Authors tend to overuse “In contrast...”, “Intriguingly,”, and “Importantly, ...” in ways that don’t always make sense in terms of what is being reported/discussed.

In conclusion, this is a terrific study that reveals the long-awaited structural properties of the Kv4.2 dodecameric complex. Improving the rigor of the electrophysiological analysis and the accuracy and/or completeness of the cited work would, however, be necessary to avoid misleading readers and facilitate understanding of the mechanistic and physiological implications.

Manuel Covarrubias

Referee #2 (Remarks to the Author):

The manuscript by Kise et al reports the cryo-EM structures of 3 different macromolecular complexes of Kv4.2 together with 1) KChIP1, 2) dpp6S, and 3) both KChip1 and dpp6s. This is an important area because these complexes assemble in vivo in brain and heart cells to yield the rapidly activating and inactivating A-type current which prevents backpropagation of the action potential. The different structures reveal novel interactions between the accessory subunits and the pore-forming subunit not previously reported/observed. Mutagenesis directed at these observed interfaces between the accessory subunits and pore-forming subunits revealed changes in the function of the complex rather than the pore-forming subunit alone (as measured with electrophysiology), suggesting that these interactions are important in providing the in vivo

phenotype of these channels. The structures provided are of high quality, they provide novel insights and also provide a handle towards dissecting the mechanism of rapid activation and inactivation of these complexes. In addition, the functional analysis of the mutants provides some insights into the modulation of Kv4.2 by the accessory subunits KCHIP and dpp6s.

On the other hand, the elephant in the room is the absence of a Kv4.2 structure without either of the subunits bound. It would be informative to see what the structural differences are in the pore-forming subunit upon co-assembly with the accessory subunits. Related to this, I find it rather interesting that the Kv4.2 main subunit is in quasi-identical conformation in the 3 different oligomeric complexes. The only difference is observed in the complex lacking KChips, where a large part of the intracellular domain becomes disordered, suggesting that KChips indeed stabilize this region into one conformation. In general, the authors are overly speculative both in the results and in the discussion, drawing mechanistic conclusions that are decidedly not supported by their data. Figure 6 is a good summary of the massive speculation present in this manuscript, where two 5-state mechanisms are proposed, but only one structure available for each model (not clear what states the structures correspond to either). In this context, it is difficult to see how the data in this manuscript provides insights into the “structural basis of kinetic modulation...”, as the title states.

Major concerns

1. The author should provide a better introduction about how much of the in vivo phenotype is captured by expression of Kv4.2 alone, compared to that in the presence of the other twosubunits and what is specifically observed for each subunit as well as for both coexpressed together with Kv4.2. The existing text is vague (87-102) and it almost sounds as if coexpression with dpp6s alone may be sufficient to recapitulate the in vivo phenotype, and KChips are not needed. Or does dpp6s speed inactivation more than required and Kchips are necessary to slow it back down?
2. Expressions such as “expedites the movement of the voltage sensor” or “stabilizes the S6 conformation” or “representing the open and closed conformations of Kv channels” and others, are highly speculative and not justified by either the existing literature data or presented data. They are used quite a lot throughout the manuscript (115, 130, 185-187, and soon).
3. The suggestion that phospholipids bind inside the pore of Kv4.2 is quite fantastic (215-221). I would expect to see perhaps phospholipid densities binding in areas where they are more likely to bind, such as in between voltage sensors crevices, etc, where they were seen in Kv2.1. It is highly unlikely that phospholipids bind in the pore. Are there favorable interaction sites? I recommend the authors discuss more likely scenarios, such as waters, or an inactivation domain?
4. The authors refer to the N-terminus as the “inactivation ball” but they do not appear to think that this domain actually leads to fast inactivation at all. A more focused paragraph on the evidence that fast inactivation is NOT induced by this domain is needed. Importantly, the fact that KCHIPs bind to the N-domain and keep it from reaching the pore, while in its absence, presumably this domain is free to go and bind in the pore, suggests that KCHIPs may prevent ball and chain inactivation. The structures immediately predict such mechanism. Have the authors tested it?
5. Importantly, the authors speculate heavily in the discussion that KChips “eliminate open state inactivation and accelerate closed state inactivation, particularly by accelerating the channel closing from the open state.....” (433-438). The authors have not presented any data to justify such statements. Further down (442-446) they speculate than in the absence of KChip the C and N termini are freely mobile and proceed “to act as fast inactivation gate from the open state by occluding the pore”. Again, they do not have a structure of Kv4.2 alone, so it is unclear what allows them to state this. Furthermore, they cite a reference in the intro where they claim that truncating the N terminus has no effect on inactivation (78-82). How do they reconcile these?
6. The authors need to show what the effect of KCHIP1 is on Kv4.2 in fig 3A. A direct overlay is needed. From staring at the two traces, it does not look like KCHIPs have an effect on inactivation of the pore-forming subunit. What does it mean then that the mutations at the interface slow it down? Is this expected? How is this interpreted mechanistically other than the vague “interaction with KCHIP affects channel gating” (293). In addition, Fig 3A is too small, the labels are not readable, the lines are too thin and colors are almost indistinguishable.

7. Dpp6s helix binding near the S1-S2 helices without affecting the structure of the VSD is not an indication that “dpp6s modulates or synchronizes the dynamics of the S1 S2 helices” (364-365). It does not “explain the different modulation mechanism” either (372). Such statements have to be removed from a results section. The entire paragraph is overly speculative based on little data (363-377). Perhaps a heavily toned-down speculation of this kind could be included in discussion.
8. Figure 5A, although better than Fig 3, also needs thicker lines, overlays between WT and dpp6s, and maybe not yellow.
9. Massive speculations in the discussion. The conclusions do not seem to be warranted. For example “Energetically, the synchronized S6 gating by KCHIP1 would reduce the energy barriers of the transitions between the closed, open, and closed inactivated states”. I did not see any evidence of synchronized S6 gating by KCHIP1 (no structure of Kv4.2 alone) and it is unclear what T1 conformational change is referred to here since the structures did not indicate such changes (449-450). In addition, it is unclear why “synchronized gating” would reduce any energy barriers.
10. Also in discussion, the authors take proposals from literature and describe them as facts such as “S1 and S2 form an interacting surface on which S4 slides up and down” (470-471). Building on these “facts”, they speculate that because the dpp6s helix binds in the vicinity of the S1 and S2 helices, this would reduce the energy barrier between the up and down conformation of S4, assertion which has no support from their data. These areas need to be toned down.
11. In the discussion, the authors make an observation (which should have been a result, if this is the case) that dpp6s accelerates both the early phase of fast inactivation and the late phase of slower inactivation. This is not clear from their data in fig 5A. More data and better analysis is required.
12. The authors do not discuss at all the phenotype expected from complexes containing both KCHIPs and dpp6s. This should be included.
13. The authors want to conclude that their structures provided insights on the mechanism of closed-state inactivation (516-517) but this is not the case. I would also expect that at least one of their structures should be in such state but there is no insight on why any of them should be closed-state inactivated (no constriction, no recognizable density in the pore) and there is no active or resting state available to compare with. Overall, more clarity is required in this manuscript about what states are predicted to be captured in the conditions of 0 voltage, and with each of the subunits bound.

Author Rebuttals to Initial Comments:

Responses to reviewers' comments:

“Referee #1:

The manuscript by Kise et al. reports an extraordinary achievement. Using cryo-EM, these authors solved the novel dodecameric structure of a voltage-gated K (Kv) channel at the atomic level. This macromolecular complex is one of the most elaborate K channel structures solved to date. It includes the Kv4.2 pore forming alpha subunit, and two ancillary beta subunits, KChIP1 and DPP6S. Moreover, to infer modulatory mechanisms they also solved octameric structures of the Kv4.2-KChIP1 complex and the Kv4.2-DPP6S complex. From the biological and physiological perspectives, this discovery is highly relevant because the Kv4.2-KChIP1-DPP6S dodecameric complex and other complexes made of similar subunits underlie the subthreshold-activating fast-inactivating Kv currents in the nervous system. These Kv currents are conserved across the animal kingdom (from jelly fish and nematodes, to mice and humans) and play key roles regulating latency to the first spike, slow repetitive spiking, action potential backpropagation, etc.

What is perhaps most remarkable from this study is that it begins to illuminate (literally!) for the first time how these interesting and important Kv channels work. Current evidence strongly suggests that the Kv4.2 dodecameric complex and similar complexes do not inactivate by the classical N-type and C-type mechanisms first reported for Shaker-type Kv1 channels that undergo open-state inactivation. Consequently, work published in the past 25 years by a handful of groups generated working hypotheses that attempt to explain the mechanisms of inactivation gating of Kv4 channels in various configurations. Collectively, previous work provided sound evidence to suggest that Kv4 channels undergo closed-state inactivation involving the S4S5 linker and the distal S6 regions that are canonically associated with activation gating. However, without direct visualization, these ideas and other intriguing features of Kv4 gating remained as hypothetical models. Fortunately, the beautiful structures solved by Kise et al. provide at last direct atomic-level visualization of unique Kv4 channel features previously hypothesized by others:

- 1) The proximal N- and C-termini interact and are sequestered by KChIP1*
- 2) Kv4 channel gating involves novel interactions between the intracellular activation gate and the T1-S1 linker*
- 3) Native Kv4 channels exist as dodecameric complexes including KChIPs and DPPs*
- 4) The voltage sensing domain is involved in closed-state inactivation and acts as the docking*

site for DPPs

5) Potentially weaker interactions between the S4S5 linker and the distal segment of S6 are responsible for the activation gate collapse that underlies closed-state inactivation

In brief, this study represents a major breakthrough that many researcher in the ion channel field where hoping for. The quality of the structural work is outstanding and the manuscript is generally well written. Also, the authors provide enough information to assess the robustness of the results and eventual replication of the experiments. I have some specific comments and suggestions that would improve the rigor of the electrophysiological analyses and the accuracy of the cited work from others.”

We really appreciate Referee #1's quite positive comments and helpful suggestions, which clearly improved the quality of our work. In the revised manuscript, according to your recommendations, we re-analyzed our electrophysiological data. We applied chord conductance to obtain activation curves and estimate the half-activation voltage and effective charge, and calculated the recovery time constant of the recovery from inactivation experiments. Also, we performed electrophysiological analyses to investigate the steady-state inactivation properties and estimate the half-inactivation voltage and effective charge. In addition, we created a cartoon model of the modulation in the Kv4.2-DPP6S-KChIP1 ternary complex.

“Specific Comments

1) It my view, it would be more appropriate to use this title: “Structural basis of gating modulation of the Kv4 macromolecular channel complex”.

We appreciate this comment. We changed the title according to your suggestion.

“2) The performance of the electrophysiological experiments to characterize mutations is sound. However, the description and interpretation of the results is entirely qualitative, and some analyses may be misleading.

For instance, to assess the voltage dependence of activation, the authors only display the current/voltage relations and arbitrarily normalize to the current amplitude at +40 mV. This normalization is misleading because changes may occur not only in terms of the voltage range over which the current operates; changes in the apparent sensitivity to voltage can also occur. These changes cannot be unambiguously determined from I/V curves. Authors should at a

minimum consider chord conductance/voltage curves and nth-order Boltzmann fits, which is an accepted standard in the field.”

Thank you for your constructive comment. We apologize for the misleading analyses. Based on your suggestion, we re-analyzed our data and applied chord conductance to obtain activation curves, and fitted them with single Boltzmann functions to estimate the half-activation voltage ($V_{1/2,act}$) and effective charge (z_{act}). We have revised Fig. 3C-F and Fig5.C-I, and $V_{1/2,act}$ and the estimated parameters are listed in Extended Data Table 2.

“Steady-state (pre-pulse) inactivation is a highly significant property of Kv4 channels, which the authors did not assess, but can provide significant information about the impact of the investigated mutations on voltage dependence and kinetics of inactivation. Also, the authors qualitatively refer to changes in inactivation kinetics, but provide no measurements to characterize these changes (e.g., weighted time constants of inactivation and recovery from inactivation).”

Thank you for your constructive comment. Based on your suggestion, we performed electrophysiological analyses to investigate the steady-state inactivation properties. These results are presented in Fig. 3G-J and Fig5. J-P, and the estimated half-inactivation voltage ($V_{1/2,inact}$) and effective charge (z_{inact}) are included in Extended Data Table 2. We also estimated the recovery time constant (τ_{rec}) of the recovery from inactivation experiments. These parameters are listed in Extended Data Table 2. We did not include the time constants of inactivation since WT and some mutants seemed not to fit the same nth-order curve, which makes us difficult to describe quantitatively. Therefore, we described them qualitatively in the text.

“Please note that apparent changes in inactivation kinetics at a given voltage that is not sufficient to reach Gmax may be secondary to the shifts in the voltage-dependence of activation.”

Thank you for this comment. We understand your concerns that both faster inactivation and slower activation affect the G-V curve by apparently shifting to the depolarized side, which may prevent us from evaluating the precise effect of each mutant. Therefore, in the manuscript, we toned down this description to avoid the totally quantitative evaluation of each mutant.

“I understand that a detailed biophysical characterization of the reported mutations is outside the scope of this study. What I am suggesting provides the most basic semiquantitative

characterization that would help interested readers better understand the reported effects, their interpretations and structural implications. In my view, the remarkable and exciting structural discoveries of this study deserve a minimum of rigor when assessing the electrophysiological correlates. Given the capabilities of the authors, the recommended analyses may not represent a major endeavor. They may, however, improve the study in a significant way.”

Thank you for your critical and constructive comments. Based on your suggestion, we have performed additional experiments to investigate the inactivation properties (Fig. 3G-J and Fig5. J-P), and calculated the parameters of the voltage-dependent activation experiments (Extended Data Table 2) and the recovery from inactivation (Fig. 3K-N and Fig. 5Q-W, Extended Data Table 2), which indeed improved our study.

“3) There are a few places where important citations are misquoted or incomplete. Specific instances by line # are listed below.

Lines 69-72; Lines 482-490: The most conclusive evidence for CSI in tetrameric, octameric and dodecameric Kv4 channels was provided in these reports:

- Fineberg, J. D., Szanto, T. G., Panyi, G. & Covarrubias, M. Closed-state inactivation involving an internal gate in Kv4.1 channels modulates pore blockade by intracellular quaternary ammonium ions. Sci Rep 6, 31131, doi:10.1038/srep31131 (2016).*
- Fineberg, J. D., Ritter, D. M. & Covarrubias, M. Modeling-independent elucidation of inactivation pathways in recombinant and native A-type Kv channels. J.Gen.Physiol 140, 513-527 (2012).*
- Dougherty, K., Santiago-Castillo, J. A. & Covarrubias, M. Gating charge immobilization in Kv4.2 channels: the basis of closed-state inactivation. J.Gen.Physiol 131, 257-273 (2008).*
- Kaulin, Y. A., Santiago-Castillo, J. A., Rocha, C. A. & Covarrubias, M. Mechanism of the modulation of Kv4:KChIP-1 channels by external K⁺. Biophysical Journal 94, 1241-1251 (2008).”*

Thank you for this comment. We have added these references.

“Lines 173-189: Previous studies listed below demonstrated that the T1-S1 linker of Kv4 dodecameric channels undergoes major conformational shifts tightly coupled to movements of the S6 tail (i.e., the activation gate). These rearrangements dramatically change the accessibility to three Cys residues in the T1 Zn binding site. It is surprising that the authors

made no reference to these observations and do not describe important structural changes that must have occurred (or not?) in this region of the solved structure. Was the site occupied by Zn? The site is expected to be Zn free because, otherwise, tightly bound Zn would protect against Cys modification.

- Wang, G. & Covarrubias, M. Voltage-dependent gating rearrangements in the intracellular T1-T1 interface of a K⁺ channel. *J.Gen.Physiol* 127, 391-400 (2006).
- Wang, G. et al. Functionally active t1-t1 interfaces revealed by the accessibility of intracellular thiolate groups in kv4 channels. *J.Gen.Physiol* 126, 55-69 (2005).”

Thank you for this comment. Indeed, we could observe the EM density corresponding to Zn²⁺ in-between neighboring T1 domains in all four Kv4.2 structures (see the attached figure below), and the T1 domain does not undergo conformational changes, as in the previous crystal structure of the T1 domain with KChIP. Therefore, we believe that the coupling movement of S6 and the T1-S1 linker occurs regardless of Zn²⁺ binding. We already cited these references in another discussion.

Zinc site in the T1 domain of Kv4.2 channels

“Lines 300-306; Lines 482-490; Lines 492-493: Amarillo et al. (2008) demonstrated that expression of precisely the dodecameric complex solved by authors closely recapitulates the A-type current endogenously expressed in cerebellar granule cells. Also, this study characterized the biophysical properties of octameric complexes solved by the authors and postulated gating models that explain gating of all configurations – Kv4.2 alone, Kv4.2+KChIP1, Kv4.2+DPP6S and Kv4.2+KChIP1+DPP6S. Furthermore, it should be noted that DPP6S also determines the unitary conductance of the native A-type K channel in cerebellar granule neurons, and that this is electrostatically dictated by the cytoplasmic N-terminal region of DPP6S (Kaulin et al., 2009).”

- *Amarillo, Y. et al. Ternary Kv4.2 channels recapitulate voltage-dependent inactivation kinetics of A-type K⁺ channels in cerebellar granule neurons. J.Physiol 586, 2093-2106 (2008).*
- *Kaulin, Y. A. et al. The dipeptidyl-peptidase-like protein DPP6 determines the unitary conductance of neuronal Kv4.2 channels. The Journal of Neuroscience 29, 3242-3251, doi:10.1523/JNEUROSCI.4767-08.2009 (2009).”*

Thank you for this comment. We have added these references.

“4) I wonder whether the unidentified density (Extended Data Fig. 10) may reflect the structural instability of the pore cavity rather than phospholipids or hydrated K ions. The single channel properties of Kv4 channels demonstrate a highly unstable opening state with frequent sojourns to partially and fully closed states before the channel finally enters an absorbing inactivated state (see Kaulin et al., 2009, cited above).”

We appreciate your productive comment. In the inner pore, we also found four similar unidentified densities in the crystal (2R9R) and cryo-EM (6EBK) structures of Kv1.2-Kv2.1 chimera channels (see the attached figure below). Although we do not know what they are, it seems that these densities are common to Kv channels. In addition, these densities are observed in all four structures of different complexes, including Kv4.2 alone (this revised study). Therefore, the densities probably do not reflect the structural instability of the pore cavity. One possibility is that these densities could be an averaged density of the detergent (GDN) used in the purification of the Kv4 complexes, as the inner pore is a hydrophobic environment. If this is the case, then GDN would preclude the closed conformation and hold the S6 gate open, although the gate is expected to be closed in a cell after depolarization at 0 mV. This case was reported for the eukaryotic Nav1.4 channels from eel and human, in which GDN occupies the pore and holds the gate open; otherwise, it is expected to close (Pan et al., Science, 2018; Yan et

Densities in the inner pore of Kv channels

al., Cell, 2017). Alternatively, as we described in our previous manuscript, the open conformation might be stable within a micelle, in which the densities represent hydrated water. A similar inconsistency was observed in the cryo-EM structure of the HCN channel in a hyperpolarized conformation, in which the pore is closed although it is open within a cell (Lee and MacKinnon, Cell, 2019).

References:

Lee, C. H. and MacKinnon, R.

Voltage Sensor Movements during Hyperpolarization in the HCN Channel.

Cell 179, 1582-1589 (2019).

Pan, X., Li, Z., Zhou, Q., Shen, H., Wu, K., Huang, X., Chen, J., Zhang, J., Zhu, X., Lei, J., Xiong, W., Gong, H., Xiao, B., and Yan, N.

Structure of the human voltage-gated sodium channel Nav1.4 in complex with β 1.
Science 362, eaau2486 (2018).

Yan, Z., Zhou, Q., Wang, L., Wu, J., Zhao, Y., Huang, G., Peng, W., Shen, H., Lei, J., and Yan, N.
Structure of the Nav1.4- β 1 Complex from Electric Eel.
Cell 170, 470-482 (2017).

“5) A cartoon that shows how dodecameric Kv4 channels may gate is not shown, and should be included in the main document, since this is the physiologically relevant complex. The cartoons of octameric Kv4 channels, currently included in the main document, may be presented as Extended Data, if there is limited space in the main document.”

According to your suggestion, we have revised Fig. 6 to include a cartoon of the dodecameric Kv4 channel. The cartoons of octameric Kv4 channels were moved to Extended Data Fig. 17.

“6) Authors tend to overuse “In contrast...”, “Intriguingly,”, and “Importantly, ...” in ways that don’t always make sense in terms of what is being reported/discussed.”

We have removed the descriptions as much as possible in the revised manuscript.

“In conclusion, this is a terrific study that reveals the long-awaited structural properties of the Kv4.2 dodecameric complex. Improving the rigor of the electrophysiological analysis and the accuracy and/or completeness of the cited work would, however, be necessary to avoid misleading readers and facilitate understanding of the mechanistic and physiological implications.

Manuel Covarrubias”

“Referee #2:

The manuscript by Kise et al reports the cryo-EM structures of 3 different macromolecular complexes of Kv4.2 together with 1) KChIP1, 2) dpp6S, and 3) both KChip1 and dpp6s. This is an important area because these complexes assemble in vivo in brain and heart cells to yield the rapidly activating and inactivating A-type current which prevents backpropagation of the action potential. The different structures reveal novel interactions between the accessory subunits and the pore-forming subunit not previously reported/observed. Mutagenesis directed at these observed interfaces between the accessory subunits and pore-forming subunits revealed changes in the function of the complex rather than the pore-forming subunit alone (as measured with electrophysiology), suggesting that these interactions are important in providing the in vivo phenotype of these channels. The structures provided are of high quality, they provide novel insights and also provide a handle towards dissecting the mechanism of rapid activation and inactivation of these complexes. In addition, the functional analysis of the mutants provides some insights into the modulation of Kv4.2 by the accessory subunits KChIP and dpp6s.

On the other hand, the elephant in the room is the absence of a Kv4.2 structure without either of the subunits bound. It would be informative to see what the structural differences are in the pore-forming subunit upon co-assembly with the accessory subunits. Related to this, I find it rather interesting that the Kv4.2 main subunit is in quasi-identical conformation in the 3 different oligomeric complexes. The only difference is observed in the complex lacking KChips, where a large part of the intracellular domain becomes disordered, suggesting that KChips indeed stabilize this region into one conformation. In general, the authors are overly speculative both in the results and in the discussion, drawing mechanistic conclusions that are decidedly not supported by their data. Figure 6 is a good summary of the massive speculation present in this manuscript, where two 5-state mechanisms are proposed, but only one structure available for each model (not clear what states the structures correspond to either). In this context, it is difficult to see how the data in this manuscript provides insights into the “structural basis of kinetic modulation...”, as the title states.”

We really appreciate Referee #2's comments and suggestions, particularly regarding the importance of the structure of Kv4.2 alone. In this revision, we successfully solved the structure of Kv4.2 alone. We included the structure of Kv4.2 alone and compared it with that of Kv4.2-KChIP1 complex in new Figure 1. In addition, we compared the structures of Kv4.2 alone, Kv4.2-DPP6S, and Kv4.2-DPP6S-KChIP1 complexes in new Figure 4. We also revised

new Extended Data Fig, 3, 4, 5, 8, and 12 which include the data of sample preparation, structure determination, and structural analyses of Kv4.2 alone. Based on it, we have proposed a plausible model of the Kv4.2 gating mechanism modulated by auxiliary β subunits, by direct structural comparisons between the four structures. Furthermore, by addressing Referee #2's point-by-point comments, we have improved our manuscript by toning down the overly speculative descriptions. Concerning Figure 6, although the structures of the Kv4 complexes solved in our manuscript represent a single open state, this study has addressed many important questions raised during the past 25 years. Therefore, we believe that our present study will become a milestone in further analyses of the mechanisms of Kv4 modulation as well as those of CSI in Kv4s. Solving the structures of the other states in each complex is planned for future research.

“Major concerns

1. The author should provide a better introduction about how much of the in vivo phenotype is captured by expression of Kv4.2 alone, compared to that in the presence of the other two subunits and what is specifically observed for each subunit as well as for both coexpressed together with Kv4.2. The existing text is vague (87-102) and it almost sounds as if coexpression with dpp6s alone may be sufficient to recapitulate the in vivo phenotype, and KChips are not needed. Or does dpp6s speed inactivation more than required and Kchips are necessary to slow it back down?”

We apologize for not providing a clear introduction about the specific roles of KChIPs and DPPs in the gating modulation of the ternary Kv4 complex *in vivo*. KChIPs are indeed necessary for explaining the inactivation mechanism of the native A-type potassium current. The most striking difference in the gating properties among Kv4.2 alone, Kv4.2-KChIP1 binary complex, Kv4.2-DPP6S binary complex, and Kv4.2-KChIP1-DPP6S ternary complex is the voltage-dependent inactivation kinetics (Amarillo et al., 2008) (also see figures below). Furthermore, in a heterologous expression system, only the Kv4.2-KChIP1-DPP6S ternary complex (but not the Kv4.2-DPP6S binary complex) recapitulates the voltage-dependent inactivation kinetics of the A-type current from cerebellar granule cells, strongly suggesting that the native Kv4 complex includes both KChIP and DPP for its physiological function (Amarillo et al., 2008) (also see figures below). The inactivation kinetics of the native A-type current and Kv4.2 ternary complex in the heterologous expression system shows the unique property of voltage-dependence. The rate of inactivation slows down with increasing depolarization (0 to 60 mV), which is not observed in other Kv subfamilies (Amarillo et al., 2008). This unique property is caused by KChIPs and consistent with the loss of N-type open state inactivation,

where stronger depolarization results in a higher probability of channel opening and a lower probability of staying in the pre-open closed state.

In addition, DPP6S accelerates the inactivation rate of Kv4.2 at the negative voltage (-60 mV to 0) to recapitulate the native A-type current (Amarillo et al., 2008) (also see figures below). This effect is caused by a hyperpolarizing shift in inactivation (Nadal et al., 2003) and an accelerating closed state inactivation (Barghaan et al., 2008). Barghaan et al. have shown that DPP6 accelerates the deactivation of Kv4.2 ($\Delta 2-40$), which lacks the N-terminal inactivation ball for open state inactivation (Barghaan et al., 2008), indicating that DPP6 accelerates channel closure. All of these studies strongly suggest that the Kv4-KChIP-DPP ternary complex is the native Kv4 complex.

Therefore, we have revised the introduction section accordingly.

Redacted: Amarillo, Y., De Santiago-Castillo, J. A., Dougherty, K., Maffie, J., Kwon, E., Covarrubias, M., and Rudy, B. Ternary Kv4.2 channels recapitulate voltage-dependent inactivation kinetics of A-type K⁺ channels in cerebellar granule neurons. *J. Physiol.* 586, 2093-2106 (2008).

References:

Amarillo, Y., De Santiago-Castillo, J. A., Dougherty, K., Maffie, J., Kwon, E., Covarrubias, M., and Rudy, B.

Ternary Kv4.2 channels recapitulate voltage-dependent inactivation kinetics of A-type K⁺ channels in cerebellar granule neurons.

J. Physiol. 586, 2093-2106 (2008).

Nadal, M. S., Ozaita, A., Amarillo, Y., Vega-Saenz de Miera, E., Ma, Y., Mo, W., Goldberg, E.

M., Misumi, Y., Ikehara, Y., Neubert, T. A., and Rudy, B.

The CD26-related dipeptidyl aminopeptidase-like protein DPPX is a critical component of neuronal A-type K⁺ channels.

Neuron 37, 449-61 (2003).

Barghaan, J., Tozakidou, M., Ehmke, H., and Bähring, R.

Role of N-terminal domain and accessory subunits in controlling deactivation-inactivation coupling of Kv4.2 channels.

Biophys. J. 94, 1276-94 (2008).

“2. Expressions such as “expedites the movement of the voltage sensor” or “stabilizes the S6 conformation” or “representing the open and closed conformations of Kv channels” and others, are highly speculative and not justified by either the existing literature data or presented data. They are used quite a lot throughout the manuscript (115, 130, 185-187, and so on).”

Thank you for this comment. We apologize for our poor explanations. However, our additional Kv4.2 alone structure, as well as its comparisons with the other 3 complex structures (new Figure 1 and 4), still supports some of our previous statements and discussions. Accordingly, in the revised manuscript, we deleted or toned down the highly speculative statements mentioned by Referee #2, and some of them have been moved to the Discussion, as follows.

Especially regarding lines 113-115 in the Introduction:

“In terms of Kv4 modulation by DPPs, DPP6 reportedly accelerates the “gating charge” movement of Kv4.2, indicating that DPP6 expedites the movement of the S4 voltage-sensing helix (Dougherty and Covarrubias, J Gen Physiol, 2006).”

A previous electrophysiology study showed that the gating charge movement of Kv4 is accelerated by DPP6 (Dougherty et al., J. Gen. Physiol., 2006). Generally, in the case of voltage-gated ion channels, the gating charge is almost equivalent to that of the positively charged residues (Arg/Lys) on the S4 voltage sensor. Therefore, the movement of the gating charge is almost equivalent to the movement of S4. To tone-down our statement, we now use “suggesting” instead of “indicating” in this sentence.

Regarding line 130 in the Introduction: *KChIP1 “stabilizes the S6 conformation”*

We have now obtained the structure of Kv4.2 alone, which confirmed that KChIP1 stabilizes

the S6 conformation.

Regarding lines 185-187 in the Results:

“These interactions would enable the concerted conformational changes of T1, S6, and the voltage sensing domain (S1-S4) as a rigid body during the gating process, which is required for the kinetic modulation of Kv4.2 by KCHIP1.”

The structure of Kv4.2 alone supports the speculation that KCHIP1 connects the cytoplasmic S6 to the T1 domain and T1-S1 linker (indirect voltage sensor) (new Figure 1). As the previous study suggested that conformational changes of the T1 domain and T1-S1 linker play a role in channel gating (Wang et al., 2005; Wang et al., 2006), it is feasible to speculate that the concerted conformational changes of T1, S6, and T1-S1 (possibly voltage sensor as well) occur during gating. We have moved this statement to the Discussion.

References:

Wang, G. & Covarrubias, M.

Voltage-dependent gating rearrangements in the intracellular T1-T1 interface of a K⁺ channel. *J.Gen.Physiol* 127, 391-400 (2006).

Wang, G. et al.

Functionally active t1-t1 interfaces revealed by the accessibility of intracellular thiolate groups in kv4 channels.

J.Gen.Physiol 126, 55-69 (2005).

Regarding lines 195-196 in the Results:

“representing the open and closed conformations of Kv channels”

Referee #2's comment is correct and we cannot determine the open and closed conformations of the channels from the S4 conformation, particularly in Kv4 channels. Therefore, we deleted this sentence.

“3. The suggestion that phospholipids bind inside the pore of Kv4.2 is quite fantastic (215-221). I would expect to see perhaps phospholipid densities binding in areas where they are more likely to bind, such as in between voltage sensors crevices, etc, where they were seen in Kv2.1. It is highly unlikely that phospholipids bind in the pore. Are there favorable interaction sites? I

recommend the authors discuss more likely scenarios, such as waters, or an inactivation domain?"

We appreciate your kind suggestion. As you expected, lipid densities are indeed observed in-between the voltage sensors and the crevices from the neighboring α subunits of Kv4.2 (see the attached figure below). In the inner pore, we also found four similar unidentified densities in the crystal structure (2R9R) and the cryo-EM structure (6EBK) of Kv1.2-Kv2.1 chimera channels (as in the attached figure below). Therefore, these densities are apparently common among Kv channels, although we do not know what they are. In addition, these densities are observed in all four structures of different complexes, including Kv4.2 alone (this revised study). Therefore, it is unlikely that the densities are inactivation domains. One possibility is that these densities could be an averaged density of the detergent (GDN) used in the purification of the Kv4 complexes, as the inner pore is a hydrophobic environment. If this is the case, then GDN precludes the closed conformation and holds the S6 gate open, although in a cell the gate is expected to be closed after depolarization at 0 mV. This case has been reported for the eukaryotic Nav1.4 channels from eel and human, where GDN occupies the pore and holds the gate open; otherwise, it is expected to close (Pan et al., *Science*, 2018; Yan et al., *Cell*, 2017). Alternatively, as we had described in our previous manuscript, the open conformation might be stable within a micelle, in which the densities represent hydrated water. A similar inconsistent example was observed in the cryo-EM structure of the HCN channel in a hyperpolarized conformation in which the pore is closed, although it is open within a cell (Lee and MacKinnon, *Cell*, 2019).

In all four of the present Kv4.2 structures, for whatever reason, S4 is up and the S6 gate is open. Therefore, we described that the structures adopt an open conformation in the text.

References:

Lee, C. H. and MacKinnon, R.

Voltage Sensor Movements during Hyperpolarization in the HCN Channel.

Cell 179, 1582-1589 (2019).

Pan, X., Li, Z., Zhou, Q., Shen, H., Wu, K., Huang, X., Chen, J., Zhang, J., Zhu, X., Lei, J., Xiong, W., Gong, H., Xiao, B., and Yan, N.

Structure of the human voltage-gated sodium channel Nav1.4 in complex with β 1.

Science 362, eaau2486 (2018).

Yan, Z., Zhou, Q., Wang, L., Wu, J., Zhao, Y., Huang, G., Peng, W., Shen, H., Lei, J., and Yan, N. Structure of the Nav1.4- β 1 Complex from Electric Eel. *Cell* 170, 470-482 (2017).

Densities in the inner pore of Kv channels

Redacted: Amarillo, Y., De Santiago- Castillo, J. A., Dougherty, K., Maffie, J., Kwon, E., Covarrubias, M., and Rudy, B. Ternary Kv4.2 channels recapitulate voltage-dependent inactivation kinetics of A-type K⁺ channels in cerebellar granule neurons. *J. Physiol.* 586, 2093-2106

(2008).

“4. The authors refer to the N-terminus as the “inactivation ball” but they do not appear to think that this domain actually leads to fast inactivation at all. A more focused paragraph on the evidence that fast inactivation is NOT induced by this domain is needed.”

Closed-state inactivation of Kv4s is still a fast process with millisecond-order kinetics, even though it is slower than N-type inactivation (Barghaan et al., 2008). Therefore, regardless of the presence of KCHIP1, the deletion of the N-terminal ball of Kv4s causes only modest slowing of their inactivation (Barghaan et al., 2008). More importantly, Kv4s end up in a closed-inactivated state irrespective of the magnitude of depolarization, and the N-type inactivation that occurs at the strong depolarization is a transient step for their inactivation (Bähring and Covarrubias, 2011). Furthermore, Kv4s form ternary complexes with KChIPs and DPPs *in vivo*. Therefore, N-type inactivation does not contribute to the inactivation of native Kv4 ternary complexes *in vivo*, and we have included this description in the introduction.

References:

Barghaan, J., Tozakidou, M., Ehmke, H., and Bähring, R.

Role of N-terminal domain and accessory subunits in controlling deactivation-inactivation coupling of Kv4.2 channels.

Biophys. J. 94, 1276-94 (2008).

Bähring, R. and Covarrubias, M.

Mechanisms of closed-state inactivation in voltage-gated ion channels.

J. Physiol. 589, 461-79 (2011).

“Importantly, the fact that KCHIPs bind to the N-domain and keep it from reaching the pore, while in its absence, presumably this domain is free to go and bind in the pore, suggests that KCHIPs may prevent ball and chain inactivation. The structures immediately predict such mechanism. Have the authors tested it?”

In our structures of Kv4.2 alone and the Kv4.2-DPP6S complex, the N-terminus is structurally

disordered, suggesting its flexibility (new Figure 4 and Extended Data Fig. 12). Nevertheless, the N-terminus of Kv4s indeed acts as an “inactivation ball” for the fast N-type inactivation in the absence of KChIPs. The evidence for this is provided by the domain-swapping experiment, where the cytoplasmic N-terminus of human Kv2.1 (amino acids 1-176) was replaced by the corresponding cytoplasmic N-terminus of human Kv4.2 (amino acids 1-180), which includes both the N-terminal 40 amino acids (inactivation ball) and subsequent T1 domain [Kv2.1(4.2NT)] (Gebauer et al., 2004). While wild type Kv2.1 does not have an inactivation ball for the N-type inactivation and is inactivated from the closed state with much slower kinetics ($\tau \sim 3$ sec) than Kv4, Kv2.1(Kv4.2NT) displays fast inactivation with millisecond-order kinetics ($\tau_1 \sim 120$ ms). The deletion of the N-terminus (amino acids 1-40) from Kv2.1(Kv4.2NT), as well as the coexpression of KChIP2.1 with Kv2.1(4.2NT), abolishes the fast inactivation, indicating that the N-terminus of Kv4.2 (amino acids 1-40) acts as inactivation ball and KChIP prevents N-type inactivation through interacting with and sequestering the N-terminus of Kv4.2 (Gebauer et al., 2004). The role of KChIPs in preventing N-type inactivation is also structurally supported by the previous structural studies of the Kv4-N-terminus (inactivation ball and T1 domain) in complex with KChIP (Pioletti et al., 2006; Wang et al., 2007), and of course further supported by our current four different full length structures of Kv4s.

References:

Gebauer, M., Isbrandt, D., Sauter, K., Callsen, B., Nolting, A., Pongs, O., and Bähring, R.

N-type inactivation features of Kv4.2 channel gating.

Biophys. J. 86, 210-23 (2004).

Pioletti, M., Findeisen, F., Hura, G. L., and Minor, D. L. Jr.

Three-dimensional structure of the KChIP1–Kv4.3 T1 complex reveals a cross-shaped octamer.

Nat. Struct. Mol. Biol. 13, 987-995 (2006).

Wang, H., Yan, Y., Liu, Q., Huang, Y., Shen, Y., Chen, L., Chen, Y., Yang, Q., Hao, Q., Wang, K., and Chai, J.

Structural basis for modulation of Kv4 K⁺ channels by auxiliary KChIP subunits.

Nat. Neurosci. 10, 32-39 (2007).

“5. Importantly, the authors speculate heavily in the discussion that KChIPs “eliminate open state inactivation and accelerate closed state inactivation, particularly by accelerating the channel closing from the open state.....” (433-438). The authors have not presented any data to justify such statements. Further down (442-446) they speculate than in the absence of KChIP the C and N termini are freely mobile and proceed “to act as fast inactivation gate from the open state by occluding the pore”. Again, they do not have a structure of Kv4.2 alone, so it is unclear what allows them to state this. Furthermore, they cite a reference in the intro where they claim

that truncating the N terminus has no effect on inactivation (78-82). How do they reconcile these?”

Regarding lines 442-446 in the discussion section, in this revision, we have solved the structure of Kv4.2 alone at 2.9 Å resolution, which shows that both the N-terminus (amino acids 1-40) and C-terminus (amino acids ~435-603) are disordered in the absence of KChIP1, as also observed in the Kv4.2-DPP6S binary complex (new Figure 1A, B). Therefore, it is feasible to speculate that both the N- and C- termini are freely mobile in the absence of KChIP1. What is still uncertain is whether the N- (1-35) and C- (472-495) termini maintain their interactions in the absence of KChIP1. Previous electrophysiological studies suggested that the N- and C-termini interact with and occlude the pore in the early phase of fast inactivation (i.e., N-type inactivation), as discussed below (Jerng and Covarrubias, 1997). In contrast, in the complex with KChIPs, KChIPs indeed bind and sequester the N-terminal inactivation ball of Kv4.2 to prevent N-type inactivation (i.e., open state inactivation). Moreover, in the structure of Kv4.2 alone, the S6 gating helices adopt a more flexible conformation with weaker interactions with

the T1-S1 linkers, and KChIPs stabilize these structures to enhance their interactions for channel closing (new Figure 1B-E). Therefore, together with previous studies below, we can speculate that KChIPs eliminate the open state inactivation and accelerate the closed state inactivation.

Previous electrophysiological studies have also shown that the deactivation of Kv4s is faster in the presence of KChIPs than Kv4s alone, which indicates that KChIPs accelerate the Kv4 channel closing from the open state (Beck et al., 2002; Gebauer et al., 2004). Furthermore, Beck et al. have shown that KChIP1 accelerates Kv4 inactivation from the preopen closed state as well (Beck et al., 2002). Thus, the electrophysiological studies published over the past 20 years strongly suggest that KChIPs accelerate both the channel closing after opening and inactivation from the pre-open closed state (i.e., closed state inactivation). However, the effect of KChIPs on the activation rate of Kv4s needs to be considered more carefully because the accelerated activation by KChIPs might be a result of slowed rapid open state inactivation, as discussed previously (Beck et al., 2002). Therefore, we removed the statements regarding the effect of the activation rate by KChIPs (lines 427, 436-438 in our previous manuscript).

In comparison with the extensively studied shaker B channels, previous functional studies have shown that, when Kv4 is expressed alone, the deletion of the C-terminus causes the elimination of the early phase of fast inactivation (N-type inactivation), as observed in the deletion of the N-terminus (Jerng & Covarrubias, 1997). In contrast, the deletion of the C-terminus does not significantly affect the inactivation of shaker B channels (Hoshi et al., 1991). Both the N- and C-terminal deletion mutants of Kv4 are inactivated almost completely at 1 sec of prolonged depolarization, as observed in wild type Kv4, suggesting that the closed state inactivation is almost intact in both mutants (Jerng & Covarrubias, 1997; Bähring et al., 2001). In addition, the

positively charged amino acid (R13) of Kv4 is not important for N-type inactivation, whereas the N-terminal Arg/Lys cluster of shaker is important for classical N-type inactivation, suggesting distinct mechanisms of N-type inactivation between Kv4 and shaker (Jerng & Covarrubias, 1997). These functional observations are consistent with the speculation that the N- and C-termini of Kv4 maintain their interaction in the absence of KChIPs and form the functional unit to act as a fast inactivation gate from the open state by occluding the pore for the early phase of fast inactivation (i.e., N-type inactivation).

Regarding lines 78-82 in the introduction section, Kv4s inactivate preferentially in a closed inactivated state at all relevant membrane potentials, as discussed above (comment 4). This means that the open inactivated state immediately shifts to the closed inactivated state. In Kv4s,

CSI is a fast inactivation step with millisecond-order kinetics, while OSI in the absence of KChIPs is a bit faster. Therefore, the truncation of the N-terminus does not significantly affect the inactivation kinetics of Kv4s.

References:

Beck, E. J., Bowlby, M., An, W. F., Rhodes, K. J., and Covarrubias, M.

Remodelling inactivation gating of Kv4 channels by KCHIP1, a small-molecular-weight calcium-binding protein.

J. Physiol. 538, 691-706 (2002).

Gebauer, M., Isbrandt, D., Sauter, K., Callsen, B., Nolting, A., Pongs, O., and Bähring, R.

N-type inactivation features of Kv4.2 channel gating.

Biophys. J. 86, 210-23 (2004).

Jerng, H. H. and Covarrubias, M.

K⁺ channel inactivation mediated by the concerted action of the cytoplasmic N- and C-terminal domains.

Biophys. J. 72, 163-174 (1997).

Hoshi, T., Zagotta, W. N., and Aldrich, R. W.

Two types of inactivation in Shaker K⁺ channels: effects of alterations in the carboxy-terminal region.

Neuron, 7, 547-556 (1991).

Bähring, R., Boland, L. M., Varghese, A., Gebauer, M., and Pongs, O.

Kinetic analysis of open- and closed-state inactivation transitions in human Kv4.2 A-type potassium channels.

J. Physiol. 535, 65-81 (2001).

“6. The authors need to show what the effect of KCHIP1 is on Kv4.2 in fig 3A. A direct overlay is needed. From staring at the two traces, it does not look like KCHIPs have an effect on inactivation of the pore-forming subunit. What does it mean then that the mutations at the interface slow it down? Is this expected? How is this interpreted mechanistically other than the

vague “interaction with KCHIP affects channel gating” (293). In addition, Fig 3A is too small, the labels are not readable, the lines are too thin and colors are almost indistinguishable.”

Thank you for this comment. Accordingly, we superimposed Kv4.2 WT or each mutant with or without KCHIP1, and used thicker lines and changed the coloring to clarify the differences in Figure 3A and Extended Data Fig. 9B. The direct overlay of the current traces of wild type Kv4.2 in the presence and absence of KCHIP1 showed that KCHIP1 affects the inactivation kinetics of wild type Kv4.2. Specifically, KCHIP1 decelerates inactivation at the early phase of depolarization, but accelerates inactivation during the late phase, which finally results in the faster inactivation in the presence of KCHIP1. We do not know how the mutations at the Kv4.2-KCHIP1 interface slow the gating channel mechanistically. However, the slowing effects are not surprising when considering that KCHIPs accelerate channel closure and closed state inactivation of wild type Kv4s, as reported in the previous functional studies (Beck et al., 2002; Gebauer et al., 2004). Even a subtle change in the interaction of KCHIP with Kv4.2 C-terminal mutants could affect the gating properties of the channel in any direction. Another possible reason could be that, in addition to the Kv4.2-KCHIP1 interactions, these mutations change the Kv4.2 N- and C-terminus interactions since KCHIP1 sandwiches the Kv4.2 N- and C-termini, which results in complex effects. Our point is that our data indicate that the interaction of KCHIP1 with the C-terminus of Kv4.2 (novel interaction that we identified from the structures) actually modulates the gating of Kv4.2 regardless of positive or negative effects on the activation, inactivation, and recovery of the C-terminal mutants of Kv4.2.

References:

Beck, E. J., Bowlby, M., An, W. F., Rhodes, K. J., and Covarrubias, M.

Remodelling inactivation gating of Kv4 channels by KCHIP1, a small-molecular-weight calcium-binding protein.

J. Physiol. 538, 691-706 (2002).

Gebauer, M., Isbrandt, D., Sauter, K., Callsen, B., Nolting, A., Pongs, O., and Bähring, R.

N-type inactivation features of Kv4.2 channel gating.

Biophys. J. 86, 210-23 (2004).

7. *Dpp6s* helix binding near the S1-S2 helices without affecting the structure of the VSD is not

an indication that “dpp6s modulates or synchronizes the dynamics of the S1 S2 helices” (364-365). It does not “explain the different modulation mechanism” either (372). Such statements have to be removed from a results section. The entire paragraph is overly speculative based on little data (363-377). Perhaps a heavily toned-down speculation of this kind could be included in discussion.

All four structures, regardless of the absence or presence of DPP6S, adopt the depolarized (S4 up) conformation and their VSDs almost completely overlap (i.e., without affecting the structure of VSD) (new Extended Data Fig. 8A, 12). This is not surprising because their VSDs are most likely stabilized in the depolarized conformation within a micelle. However, our mutagenesis and electrophysiological experiments indicated that DPP6S modulates Kv4.2 through the S1-S2 helices.

We did not intend to “conclude” that “dpp6s modulates or synchronizes the dynamics of the S1 S2 helices” (lines 364-365). That’s why we said “DPP6S appears to modulate” and added the next sentence “The role of these interactions in Kv4 modulation will be discussed later in this paper”. We just wanted to mention that we would discuss the role of the DPP’s interaction with the S1-S2 helices later in the Discussion. We also did not intend to conclude that the structure of the Kv4.2-DPP6S “explains the different modulation mechanism” (line 372). This is our speculation and that’s why we said the structure “could explain”. However, the speculation is feasible because the interaction mode is novel among the interactions between any voltage-gated channels and auxiliary subunits reported thus far. Nevertheless, we agree with the referee’s concerns and we toned-down our statements and moved them to the Discussion.

“8. Figure 5A, although better than Fig 3, also needs thicker lines, overlays between WT and dpp6s, and maybe not yellow.”

According to the comment, we superimposed Kv4.2 WT or each mutant with and without DPP6S (Fig. 5A, Extended Data Fig. 15C). Furthermore, we used thicker lines and changed the coloring to clarify the differences.

“9. Massive speculations in the discussion. The conclusions do not seem to be warranted. For example “Energetically, the synchronized S6 gating by KCHIP1 would reduce the energy barriers of the transitions between the closed, open, and closed inactivated states”. I did not see any evidence of synchronized S6 gating by KCHIPS (no structure of Kv4.2 alone) and it is unclear what T1 conformational change is referred to here since the structures did not indicate

such changes (449-450). In addition, it is unclear why “synchronized gating” would reduce any energy barriers.”

We agree that there are no data to discuss the energetics of S6 gating, and deleted the sentences regarding the gating energetics. However, in this revision, we have solved the structure of Kv4.2 alone and it is quite possible that S6 gating is synchronized by KChIPs, for the following reasons. First, it is obvious that one KChIP1 stabilizes the S6 conformation as well as the N-terminus from the neighboring subunit of Kv4.2, from a structural comparison between Kv4.2 alone and the Kv4.2-KChIP1 complex. Second, one KChIP1 also interacts with two T1 domains from neighboring subunits (our structures) (Pioletti et al., 2006; Wang et al., 2007). Third, previous functional studies have suggested that the T1-S1 linker of Kv4 dodecameric channels undergoes major conformational shifts tightly coupled to movements of the S6 tail upon binding with KChIP1, although the T1 conformation change is still unknown. Fourth, KChIP1 facilitates the interaction between the T1-S1 linker and S6 gate, as shown by a structural comparison between Kv4.2 alone and the Kv4.2-KChIP1 complex (new Figure 1). Together, KChIP1 binds and stabilizes the S6 helix and T1 domains from neighboring subunits in the tetramer, which may induce synchronized S6 gating. We included these descriptions in the discussion.

References:

Pioletti, M., Findeisen, F., Hura, G. L., and Minor, D. L. Jr.

Three-dimensional structure of the KChIP1–Kv4.3 T1 complex reveals a cross-shaped octamer. *Nat. Struct. Mol. Biol.* 13, 987-995 (2006).

Wang, H., Yan, Y., Liu, Q., Huang, Y., Shen, Y., Chen, L., Chen, Y., Yang, Q., Hao, Q., Wang, K., and Chai, J.

Structural basis for modulation of Kv4 K⁺ channels by auxiliary KChIP subunits. *Nat. Neurosci.* 10, 32-39 (2007).

“10. Also in discussion, the authors take proposals from literature and describe them as facts such as “S1 and S2 form an interacting surface on which S4 slides up and down” (470-471). Building on these “facts”, they speculate that because the dpp6s helix binds in the vicinity of the S1 and S2 helices, this would reduce the energy barrier between the up and down conformation of S4, assertion which has no support from their data. These areas need to be toned down.”

Among the hypotheses to explain the voltage dependency in voltage-gated channels, the hypothesis that S4 slides on the surface formed by S1 and S2, depending on the membrane potential, might be most likely. In our Kv4.2 complexes, the single-spanning transmembrane helix of DPP6S binds and apparently stabilizes the structures of the S1 and S2 helices. Therefore, we suggested that DPP6S enhances the voltage sensitivity of Kv4.2. Nevertheless, according to the referee's comment, we toned-down these descriptions.

“11. In the discussion, the authors make an observation (which should have been a result, if this is the case) that dpp6s accelerates both the early phase of fast inactivation and the late phase of slower inactivation. This is not clear from their data in fig 5A. More data and better analysis is required.”

Previous studies of the heterologous expression of Kv4 and DPP6S, as we had mentioned (line 484), showed that DPP6S accelerates both fast and slower (although still fast) inactivation, although the degree of acceleration varies (Barghaan et al., 2008; Jerng et al., 2004). The acceleration of fast inactivation is canceled by the further addition of KChIP (Barghaan et al., 2008). In addition, DPP6S accelerates both the inactivation and deactivation of even N-terminally truncated Kv4 (i.e., loss of N-type inactivation) (Barghaan et al., 2008). These studies strongly suggest that DPP6S accelerates both the N-type open inactivation and closed state inactivation. Nevertheless, as the suggestions are derived from the previous functional studies, we toned down the description.

References:

Barghaan, J., Tozakidou, M., Ehmke, H., and Bähring, R.

Role of N-terminal domain and accessory subunits in controlling deactivation-inactivation coupling of Kv4.2 channels.

Biophys. J. 94, 1276-94 (2008).

Jerng, H. H., Qian, Y., and Pfaffinger, P. J.

Modulation of Kv4.2 channel expression and gating by Dipeptidyl Peptidase 10 (DPP10).

Biophys. J. 87, 2380-2396 (2004).

12. The authors do not discuss at all the phenotype expected from complexes containing both KCHIPs and dpp6s. This should be included.

We have already described that as KCHIP1 and DPP6S interact with distinct structures of Kv4.2 to modulate its kinetics in different manners, their effects are additive. We have included this description in the discussion.

“13. The authors want to conclude that their structures provided insights on the mechanism of closed-state inactivation (516-517) but this is not the case. I would also expect that at least one of their structures should be in such state but there is no insight on why any of them should be closed-state inactivated (no constriction, no recognizable density in the pore) and there is no active or resting state available to compare with. Overall, more clarity is required in this manuscript about what states are predicted to be captured in the conditions of 0 voltage, and with each of the subunits bound.”

We did not intend to conclude that our structures provide all insights into the mechanisms of closed state inactivation. As you pointed out, we do not have the structures corresponding to the closed inactivated conformation. As we described above (the answer to comment 4), our structures of both Kv4.2 alone and in the other three complexes capture the open conformation under the depolarized condition (0 mV), while they are predicted to adopt a closed conformation after depolarization within a cell. This discrepancy could be attributed to the micelles in the cryo-EM structures or other unknown factors. A similar inconsistent example has recently been observed in the cryo-EM structure of the HCN channel in a hyperpolarized conformation, in which the pore is closed while it is open within a cell (Lee and MacKinnon, Cell, 2019). Another possibility is that the uncharacterized density in the inner pore is a detergent such as GDN, which precludes the closed conformation and holds the S6 gate open. This situation has also been reported for the eukaryotic Nav1.4 channels from eel and human, in which GDN occupies the pore and holds the gate open, while otherwise it is expected to close (Pan et al., Science, 2018; Yan et al., Cell, 2017).

We agree that our statement is speculative, and thus we toned-down our description. However, our structures of the open conformation still provide a good starting point toward revealing the structural basis of closed state inactivation. This is because previous studies strongly suggested that the dynamic interaction of the S4-S5 linker and S6 gate is the molecular basis of closed state inactivation (Bähring and Covarrubias, 2011; Wollberg and Bähring, 2016). Therefore, we

could consider the present open conformation as a pre-closing conformation. While our statement in this section would be too speculative, we believe that our discussion paves the way for elucidating the mechanism of closed state inactivation of Kv4, which is clearly distinct from the classical N-type/C-type inactivations of other voltage-gated channels, such as shaker and Kv1.

References:

Bähring, R. and Covarrubias, M.

Mechanisms of closed-state inactivation in voltage-gated ion channels.

J. Physiol. 589, 461-79 (2011).

Wollberg, J. and Bähring, R.

Intra- and Intersubunit Dynamic Binding in Kv4.2 Channel Closed-State Inactivation.

Biophys. J. 110, 157-175 (2016).

Lee, C. H. and MacKinnon, R.

Voltage Sensor Movements during Hyperpolarization in the HCN Channel.

Cell 179, 1582-1589 (2019).

Pan, X., Li, Z., Zhou, Q., Shen, H., Wu, K., Huang, X., Chen, J., Zhang, J., Zhu, X., Lei, J., Xiong, W., Gong, H., Xiao, B., and Yan, N.

Structure of the human voltage-gated sodium channel Nav1.4 in complex with β 1.

Science 362, eaau2486 (2018).

Yan, Z., Zhou, Q., Wang, L., Wu, J., Zhao, Y., Huang, G., Peng, W., Shen, H., Lei, J., and Yan, N.

Structure of the Nav1.4- β 1 Complex from Electric Eel.

Cell 170, 470-482 (2017).

Reviewer Reports on the First Revision:

Referees' comments:

Referee #1 (Remarks to the Author):

The revised manuscript by Kise et al. has been significantly improved in many ways by including a more quantitative analysis of electrophysiological properties (i.e., G-V curves, pre-pulse inactivation curves, and time constant of recovery from inactivation), and toning down some mechanistic speculation (as suggested by Rev. 2). Furthermore, the authors solved the cryoEM structure of the Kv4.2 alpha subunit alone, allowing a direct assessment of structural changes induced by the interactions with the ancillary subunits KCHIP1 and DPP6s. As a result, my enthusiasm has been increased even further. There is, however, a previously raised issue that the authors did not fully address.

In their rebuttal, the authors indicated that it was not possible to obtain time constants of inactivation because wild-type and mutants did not seem to fit the same nth order curve. This problem is not surprising because the kinetics of macroscopic inactivation in Kv4 channels is non-exponential and particularly complex. This is why I suggested to compute weighted average time constants to evaluate the development of macroscopic inactivation $[\tau_w = (\tau_1 \cdot A_1 + \dots + \tau_n \cdot A_n) / (A_1 + \dots + A_n)]$ at various voltages. Generally, the nth order of the sum of exponential terms increases with depolarization. In my opinion this measurement is important because the study is significantly focused on mechanisms of inactivation. Otherwise the electrophysiological analysis would lack a minimum of quantitative analysis. Moreover, follow up studies would not find a quantitative reference in this study to compare results. If manuscript length is an issue, this analysis could be included as extended data. Please compute weighted average time constants and fraction of sustained current at various voltages for the wild type and the mutants.

Conducting this analysis should be straightforward since the authors already have the recordings and the means to conduct this analysis. Its execution would provide the minimum of information to complete a reasonably quantitative assessment of the observed changes. Some inactivation rate changes over the voltage range of activation gating may be secondary to voltage dependent shifts in the G-V curve because inactivation is generally coupled to activation.

Minor comments

Line 287 – it states: “...indicating preferential effect on the development of recovery from inactivation”. This statement is misleading because -in a simple scenario- a depolarizing shift of the inactivation curve could result from either slowing inactivation or accelerating recovery from inactivation. Assuming steady-state, it is more accurate to describe this change as a “relative destabilization of the inactivated state”.

Line 294 – please state the voltage at which you measured the recovery from inactivation (albeit it is indicated in the Methods, -100 mV).

Line 412 - it states: “...indicating its preferential effect on the development of inactivation”. As mentioned above, this statement is misleading. It is more accurate to refer to the hyperpolarizing shift of the inactivation curve as a “relative stabilization of the inactivated state”.

Referee #2 (Remarks to the Author):

The authors have done a wonderful job revising the manuscript and they addressed the majority of my concerns. The remaining issue is the cartoon in Figure 6. Usually, such cartoons are meant to represent a schematic of a proposed mechanism, from insights shown in the manuscript. The authors presented in this manuscript several amazing structures of: Kv4.2 alone, in complex with KChips, in complex with DPP6 and in complex with both KCHIPS and DPP6 together. However, the channel part, with the voltage sensors, gate and pore, appears identical in all structures: open pore with voltage sensors activated, indicating the same conformation (open-activated?). The scheme presented in Fig. 6 is misleading since it suggests that the structures solved here somehow led to this cartoon, which includes gating between 5 different channel conformations (closed, closed activated, open activated, closed inactivated, open inactivated). Bottom line: in my opinion the scheme is not based on their results, so it does not belong in the manuscript as is.

Author Rebuttals to First Revision:

Referee #1

The revised manuscript by Kise et al. has been significantly improved in many ways by including a more quantitative analysis of electrophysiological properties (i.e., G-V curves, pre-pulse inactivation curves, and time constant of recovery from inactivation), and toning down some mechanistic speculation (as suggested by Rev. 2). Furthermore, the authors solved the cryoEM structure of the Kv4.2 alpha subunit alone, allowing a direct assessment of structural changes induced by the interactions with the ancillary subunits KChIP1 and DPP6s. As a result, my enthusiasm has been increased even further. There is, however, a previously raised issue that the authors did not fully address.

In their rebuttal, the authors indicated that it was not possible to obtain time constants of inactivation because wild-type and mutants did not seem to fit the same nth order curve. This problem is not surprising because the kinetics of macroscopic inactivation in Kv4 channels is non-exponential and particularly complex. This is why I suggested to compute weighted average time constants to evaluate the development of macroscopic inactivation [$\tau_w = (\tau_1 \cdot A_1 + \dots + \tau_n \cdot A_n) / (A_1 + \dots + A_n)$] at various voltages. Generally, the nth order of the sum of exponential terms increases with depolarization. In my opinion this measurement is important because the study is significantly focused on mechanisms of inactivation. Otherwise the electrophysiological analysis would lack a minimum of quantitative analysis. Moreover, follow up studies would not find a quantitative reference in this study to compare results. If manuscript length is an issue, this analysis could be included as

extended data. Please compute weighted average time constants and fraction of sustained current at various voltages for the wild type and the mutants.

Conducting this analysis should be straightforward since the authors already have the recordings and the means to conduct this analysis. Its execution would provide the minimum of information to complete a reasonably quantitative assessment of the observed changes. Some inactivation rate changes over the voltage range of activation gating may be secondary to voltage dependent shifts in the G-V curve because inactivation is generally coupled to activation.

Thank you for your further constructive comment. Based on your guidance, we calculated the inactivation time constants and the corresponding amplitudes. These data were inserted as Extended Data Table 2 and Supplementary Table.

Minor comments

Line 287 – it states: “...indicating preferential effect on the development of recovery from inactivation”. This statement is misleading because -in a simple scenario- a depolarizing shift of the inactivation curve could result from either slowing inactivation or accelerating recovery from

inactivation. Assuming steady-state, it is more accurate to describe this change as a “relative destabilization of the inactivated state”.

We appreciate your comment. We changed the description according to your suggestion.

Line 294 – please state the voltage at which you measured the recovery from inactivation (albeit it is indicated in the Methods, -100 mV).

We included the protocol of this experiment in Figure 3b and Figure legend 3b-e in which the voltage (-100 mV) we applied is described.

Line 412 - it states: “...indicating its preferential effect on the development of inactivation”. As mentioned above, this statement is misleading. It is more accurate to refer to the hyperpolarizing shift of the inactivation curve as a “relative stabilization of the inactivated state”.

We appreciate your comment. We changed the description according to your suggestion.

Referee #2

The authors have done a wonderful job revising the manuscript and they addressed the majority of my concerns. The remaining issue is the cartoon in Figure 6. Usually, such cartoons are meant to represent a schematic of a proposed mechanism, from insights shown in the manuscript. The authors presented in this manuscript several amazing structures of: Kv4.2 alone, in complex with KChips, in complex with DPP6 and in complex with both KCHIPS and DPP6 together. However, the channel part, with the voltage sensors, gate and pore, appears identical in all structures: open pore with voltage sensors activated, indicating the same conformation (open-activated?). The scheme presented in Fig. 6 is misleading since it suggests that the structures solved here somehow led to this cartoon, which includes gating between 5 different channel conformations (closed, closed activated, open activated, closed inactivated, open inactivated). Bottom line: in my opinion the scheme is not based on their results, so it does not belong in the manuscript as is.

We appreciate your comment. We moved Figure 6 to Extended Data Fig. 13 to discuss our model for gating modulation of Kv4 complexes by KChIP and DPP.